# DECOUPLED SGDA FOR GAMES WITH INTERMITTENT STRATEGY COMMUNICATION

## ABSTRACT

We focus on reducing communication overhead in multiplayer games, where frequently exchanging strategies between players is not feasible and players have noisy or outdated strategies of the other players. We propose *Decoupled SGDA*, an extension of Stochastic Gradient Descent Ascent (SGDA), where players perform independent updates using outdated strategies of opponents, with periodic strategy synchronization. For Strongly-Convex-Strongly-Concave (SCSC) games, we demonstrate that Decoupled SGDA achieves near-optimal communication complexity comparable to the best-known GDA rates. For *weakly coupled* games where the interaction between players is lower relative to non-interactive part of the game, Decoupled SGDA significantly reduces communication costs compared to standard SGDA. Our findings extend to multi-player games. To provide insights into the effect of communication frequency and convergence, we extensively study the convergence of Decoupled SGDA for quadratic minimax problems. Lastly, in settings where the noise over the players is imbalanced, Decoupled SGDA significantly outperforms federated minimax methods.

## 1 INTRODUCTION

Several real-world problems in diverse areas, such as economics and computer science, can frequently be described as $N$-player differentiable games (Von Neumann & Morgenstern, 2007). While players may have competing objectives, the aim is to identify an equilibrium, a strategy where no player benefits from deviating unilaterally. Examples of such games in machine learning include Generative Adversarial Networks (GANs, Goodfellow et al., 2014), adversarial robustness (Madry et al., 2017; Shafahi et al., 2019; Robey et al., 2023) and multi-agent reinforcement learning (e.g., Lowe et al., 2017; Li et al., 2019).

Minimax optimization problems are a special case of $N$-player games, where the aim is to find a *saddle point* of an objective $f(\mathbf{u}, \mathbf{v})$ that minimizes $f(\mathbf{u}, \cdot)$ over $\mathbf{u}$ and maximizes $f(\cdot, \mathbf{v})$ over $\mathbf{v}$.

Several gradient-based methods have been proposed for solving the above problem (Korpelevich, 1976; Popov, 1980; Balduzzi et al., 2018; Nouiehed et al., 2019; Chavdarova et al., 2020; Kovalev & Gasnikov, 2022). One of the most widely used is the gradient descent method. In the context of zero-sum minimax games, this approach is referred to as *Gradient Descent Ascent* (GDA), where the minimizing player takes descent steps and the maximizing player takes ascent steps.

In some situations, however, players may not have direct access to their opponents' exact strategies. The $\mathbf{u}$–player might only have a noisy estimate of $\mathbf{v}$ when updating its parameters, and vice versa. In extreme cases, players might operate with outdated strategies from their opponents, with limited opportunities to synchronize. We refer to this scenario as *games with intermittent strategy communication* (ISC-games). Here are a few illustrative examples:

- **Corporate competitors.** Companies frequently adjust their strategies based on individual objectives and the strategies of their competitors. For instance, *Netflix* may need to lower its prices if a competitor like *Max* reduces its subscription rates (Jagadeesan et al., 2022). Corporations may occasionally release (noisy) general information about their strategies, giving each company an imperfect understanding of its competitor's actions. Alternatively, companies might hire experts to estimate competitor strategies using publicly available data, although this process is expensive and infrequent.

- $N$**-agents with restricted communication.** In control theory, applications involving drones or robots are modeled with $N$-player games (see Spica et al., 2020; Laine et al., 2021; Zhou et al., 2021, and references therein). However, due to factors like long distances or limited battery life caused by weight constraints, communication between agents regarding learned strategies is costly and can only occur intermittently.

In summary, this paper focuses on the following questions.

- *Can players learn "locally" when relying on noisy or outdated strategies from their opponents in ISC-games?*
- *How do the convergence rate and communication costs of the proposed optimization method compare to the baseline? Is acceleration achievable?*

To address the former, we propose a simple extension of the gradient descent method where agents perform local updates while using outdated strategies from their opponents. For minimax problems, we refer to this method as *Decoupled SGDA*, and for $N$-player games *Decoupled SGD*. The second question is explored in detail by analyzing the convergence rate of Decoupled SGD(A) and identifying the problem class where acceleration is achieved, which we refer to as *Weakly Coupled Games*.

**Contributions.** Our contributions include:

- We introduce *Decoupled SGD(A)* for games with intermittent strategy communication, where each player performs $K$ updates based on outdated strategies from the other players, followed by a synchronization step to exchange the updated strategies of all players.
- We analyze its convergence in both the strongly-convex strongly-concave (SCSC) setting and in $N$-player games where each player's utility is strongly convex. Additionally, we identify a specific regime, termed *Weakly Coupled Games*, where Decoupled SGD(A) demonstrates communication acceleration compared to the baseline GD(A), by removing the dependency on player conditioning. Moreover, under an additional assumption on the interactive part of the game, our method can outperform the optimal first-order method in terms of communication rounds for solving SCSC games.
- We study the convergence of Decoupled SGDA for quadratic minimax games with bilinear coupling between the players, providing in-depth insights into the algorithm's convergence behavior. We provide a convergence guarantee for Federated Minimax games in the context of our method, matching the state-of-the-art results of Local SGDA.
- Through numerical experiments, we demonstrate the efficacy of Decoupled SGDA (i) in minimax optimization and non-convex GAN settings in weakly coupled games, and (ii) over federated learning in settings where opponents have gradients with imbalanced noise.
- We also propose a heuristic to accelerate the convergence of decoupled SGDA in Appendix G and present numerical evidence demonstrating its practical effectiveness.

To simplify the exposition, the main body of the paper focuses on the minimax setting, while the extension to $N$-player games is presented in Appendix C.

## 1.1 RELATED WORKS

Our work draws from multiple lines of work, and herein, we review these and discuss the difference with federated learning. Appendix E gives additional discussion and lists works on decentralized optimization. The latter are further from our work in that there is no centralized communication, and nodes communicate with neighbors.

**Game optimization.** Nemirovski (2004); Nesterov (2007) achieve a rate of $\mathcal{O}(\frac{1}{T})$ for convex-concave minimax problems. For strongly-convex-strongly-concave games, (i) Thekumparampil et al. (2019) combine Nestrov's Accelerated Gradient and mirror-prox and achieve $\tilde{\mathcal{O}}(\frac{1}{T^2})$ rate of convergence, (ii) Wang & Li (2020) explore ideas from accelerated proximal point and achieve a linear rate, and (iii) Kovalev & Gasnikov (2022) propose a method with $\mathcal{O}(\sqrt{\kappa_u \kappa_v} \log \frac{1}{\epsilon})$ rate of convergence which matches the lower bounds (Zhang et al., 2022b; Ibrahim et al., 2020). Several works focus on accelerating the convergence of GDA (Lee et al., 2024; Zhang et al., 2022a).

Quadratic games with bilinear coupling are studied in (Zhang et al., 2021). Nouiehed et al. (2019) propose a method that performs multiple first-order steps on only one of the parameters to solve minimax problems. Tsaknakis et al. (2021) study a generalized minimax problem with linear constraints coupling the decision variables. Tseng & Yun (2009) study coordinate gradient descent method for minimizing the sum of a smooth and separable convex function. Jain et al. (2018) shows that accelerated stochastic gradient descent can outperform traditional stochastic gradient descent in minimax optimal statistical risk for least squares regression. Yoon & Ryu (2021) present algorithms with accelerated $\mathcal{O}(1/k^2)$ last-iterate rates for smooth minimax optimization—outperforming existing methods—and establish the optimality of this rate through a matching lower bound.

**Federated learning.** Building on the foundational work of McMahan et al. (2017), numerous works have explored distributed minimization, or federated learning, across various settings (e.g., Stich, 2018; Koloskova et al., 2020; Karimireddy et al., 2020; Woodworth et al., 2020a;b). In the context of minimax optimization, Deng & Mahdavi (2021); Sharma et al. (2022) extended the so-called *Local SGD* to the minimax setting, achieving convergence rates for different classes of functions in both heterogeneous and homogeneous regimes. Although both Federated Minimax and Decoupled SGDA are designed to solve minimax optimization problems in a distributed fashion, their approaches to achieving this are fundamentally different. Refer to Section F.1 for more details.

## 2 SETTING AND PRELIMINARIES

We consider the following saddle-point problem over $\mathcal{X} = \mathcal{X}_u \times \mathcal{X}_v$, with $\mathcal{X}_u = \mathbb{R}^{d_u}, \mathcal{X}_v = \mathbb{R}^{d_v}$:

$$\min_{\mathbf{u} \in \mathcal{X}_u} \max_{\mathbf{v} \in \mathcal{X}_v} f(\mathbf{u}, \mathbf{v}), \tag{SP}$$

where $f \colon \mathcal{X} \to \mathbb{R}$ is a differentiable function. Its solution is defined as a point $\mathbf{x}^\star \equiv (\mathbf{u}^\star, \mathbf{v}^\star) \in \mathcal{X}$ satisfying the following variational principle: $f(\mathbf{u}^\star, \mathbf{v}) \leqslant f(\mathbf{u}^\star, \mathbf{v}^\star) \leqslant f(\mathbf{u}, \mathbf{v}^\star)$ for all $(\mathbf{u}, \mathbf{v}) \in \mathcal{X}$. We define the following operator $F \colon \mathcal{X} \to \mathcal{X}$:

$$F(\mathbf{x}) := (\nabla_u f(\mathbf{x}), -\nabla_v f(\mathbf{x})), \qquad \mathbf{x} \in \mathcal{X}. \tag{1}$$

An important property of this operator is that $F(\mathbf{x}^\star) = 0$.

**Notation.** We use bold lower-case letters for vectors and bold capital letters for matrices. We consider unconstrained two-player games where the decision vectors of the players (typically denoted by $\mathbf{u}$ and $\mathbf{v}$) live in the spaces $\mathcal{X}_u = \mathbb{R}^{d_u}$ and $\mathcal{X}_v = \mathbb{R}^{d_v}$, respectively. The corresponding product space $\mathcal{X} = \mathcal{X}_u \times \mathcal{X}_v = \mathbb{R}^d$ (with $d = d_u + d_v$) consists of vectors $\mathbf{x} = (\mathbf{u}, \mathbf{v}) \in \mathbb{R}^d$, where $\mathbf{u} \in \mathcal{X}_u$ and $\mathbf{v} \in \mathcal{X}_v$. For a differentiable function $f \colon \mathcal{X} \to \mathbb{R}$, we denote partial gradients at a point $\mathbf{x} = (\mathbf{u}, \mathbf{v}) \in \mathcal{X}$ w.r.t. the corresponding variables by $\nabla_u f(\mathbf{x})$ and $\nabla_v f(\mathbf{x})$, respectively, so that $\nabla f(\mathbf{x}) = (\nabla_u f(\mathbf{x}), \nabla_v f(\mathbf{x}))$. We use $\langle \cdot, \cdot \rangle$ to denote the standard inner product, keeping the same notation for each of the vector spaces we consider. We assume that the spaces $\mathcal{X}_u$ and $\mathcal{X}_v$ are equipped with Euclidean norms, denoted by $\|\mathbf{u}\|_u, \|\mathbf{v}\|_v$. The norm in the space $\mathcal{X}$ is then defined by $\|\mathbf{x}\| = (\alpha\|\mathbf{u}\|_u^2 + \beta\|\mathbf{v}\|_v^2)^{1/2}$ where $\alpha, \beta > 0$. We use the notation $\|\mathbf{g}_u\|_{u,*}, \|\mathbf{g}_v\|_{v,*}$ and $\|\mathbf{g}\|_* := (\frac{1}{\alpha}\|\mathbf{g}_u\|_{u,*}^2 + \frac{1}{\beta}\|\mathbf{g}_v\|_{v,*}^2)^{1/2}$ to denote the corresponding dual norms.

**Stochastic Oracles with Noise Imbalance.** Following standard conventions, we assume that both players—the minimization player $\mathbf{u}$, and the maximization player $\mathbf{v}$—have access unbiased stochastic oracles $G_u(\mathbf{x}, \xi), G_v(\mathbf{x}, \xi)$, with the property $\mathbb{E}[G_u(\mathbf{x}, \xi)] = F(\mathbf{x}), \mathbb{E}[G_v(\mathbf{x}, \xi)] = F(\mathbf{x})$. We make the following assumption on the variance:

**Assumption 1.** *There exists finite constants* $\sigma_{uu}^2, \sigma_{vv}^2$ *such that for all* $\mathbf{x} \in \mathcal{X}$:

$$\mathbb{E}\left\|\left[(G_u(\mathbf{x}, \xi) - F(\mathbf{x}))\right]_u\right\|_{u,*}^2 \leqslant \sigma_{uu}^2 \qquad\qquad \mathbb{E}\left\|\left[(G_v(\mathbf{x}, \xi) - F(\mathbf{x}))\right]_v\right\|_{v,*}^2 \leqslant \sigma_{vv}^2.$$

*Here, we use the operator* $[\cdot]_i$ *to denote the coordinates corresponding to player* $i \in \{u, v\}$. *For convenience, we define* $\sigma^2 := \sigma_{uu}^2 + \sigma_{vv}^2$.

Note that we only require that the noise of self-gradients is bounded, i.e. $\sigma_{uu}^2$ and $\sigma_{vv}^2$. However, we do not make any assumption on the noise for the estimates of the gradients of the other player. Concretely, the variances $\sigma_{uv}^2 := \mathbb{E}\left\|\left[(G_u(\mathbf{x}, \xi) - F(\mathbf{x}))\right]_v\right\|_{v,*}^2$ and $\sigma_{vu}^2 :=$

$\mathbb{E}\left\|\left[(G_v(\mathbf{x}, \xi) - F(\mathbf{x}))\right]_u\right\|_{u,*}^2$ can be arbitrarily large, possibly unbounded. This is is contrast to other works on stochastic min-max optimization, that require the variance of both $G_u$ and $G_v$ to be bounded.

**Monotonicity and Smoothness.** We outline the necessary assumptions for establishing the convergence of our method.

**Assumption 2** (Strong monotonicity). *Operator $F$ from* (1) *is strongly monotone with parameter $\mu > 0$, i.e., for all $\mathbf{x}, \mathbf{x}' \in \mathcal{X}$, the following inequality holds:*

$$\langle F(\mathbf{x}) - F(\mathbf{x}'), \mathbf{x} - \mathbf{x}' \rangle \geq \mu \|\mathbf{x} - \mathbf{x}'\|^2.$$

**Assumption 3** (Lipschitz smoothness). *Operator $F$ from* (1) *is $L$-Lipschitz, i.e., for all $\mathbf{x}, \mathbf{x}' \in \mathcal{X}$, the following inequality holds:*

$$\|F(\mathbf{x}) - F(\mathbf{x}')\|_* \leq L \|\mathbf{x} - \mathbf{x}'\| . \tag{2}$$

In the sequel, we will also need to use more refined smoothness constant $L_c$ which is defined by the following inequalities holding for all $\mathbf{x} \equiv (\mathbf{u}, \mathbf{v}), \mathbf{x}' \equiv (\mathbf{u}', \mathbf{v}') \in \mathcal{X}$:

$$\begin{aligned}
\|\nabla_u f(\mathbf{u}, \mathbf{v}) - \nabla_u f(\mathbf{u}, \mathbf{v}')\|_{u,*} &\leq L_c \|\mathbf{v} - \mathbf{v}'\|_v , \\
\|\nabla_v f(\mathbf{u}, \mathbf{v}) - \nabla_v f(\mathbf{u}', \mathbf{v})\|_{v,*} &\leq L_c \|\mathbf{u} - \mathbf{u}'\|_u .
\end{aligned} \tag{3}$$

We will show in Section 4 that this constant plays an important role in communication acceleration as it captures how interactive the game is. In this work, we assume that there exists one constant $L_c$ which can be used for both inequalities.

For the reader's convenience, Tables 2 and 3 in the appendix summarize our notations.

## 3 DECOUPLED SGDA FOR TWO-PLAYER GAMES

In this section we introduce decoupled SGDA and explain its motivation.

**General Idea: Communication Efficient Strategy Exchange.** A standard way for solving (SP) is as follows:

$$\mathbf{x}_{t+1} = \mathbf{x}_t - \gamma G(\mathbf{x}_t, \xi) \quad \text{with} \quad G(\mathbf{x}, \xi) = \begin{pmatrix} \nabla_u f(\mathbf{u}, \mathbf{v}; \xi) \\ -\nabla_v f(\mathbf{u}, \mathbf{v}; \xi) \end{pmatrix} . \tag{SGDA}$$

However, in a distributed setting, the players need one *round of communication* to exchange their parameters $(\mathbf{u}_t, \mathbf{v}_t)$ in every step of the method.

To alleviate this communication issue, earlier works proposed so-called *local update methods* that reduce the amount of communication by performing local parameter updates for each player separately. For instance, both player could perform $K \geq 1$ updates on a local copy of the parameters. After every communication round, local variables are initialized as $\mathbf{x}_t^v = \mathbf{x}_t^u = \mathbf{x}_t$, and updated as:

$$\mathbf{x}_{t+K}^u = \mathbf{x}_t^u - \gamma \sum_{i=0}^{K-1} G_u(\mathbf{x}_{t+i}^u, \xi_{t+i}), \quad \mathbf{x}_{t+K}^v = \mathbf{x}_t^v - \gamma \sum_{i=0}^{K-1} G_v(\mathbf{x}_{t+i}^v, \xi_{t+i}). \quad \text{(local-SGDA)}$$

The local variables are then synchronized in a communication round, $\mathbf{x}_{t+K} := \frac{1}{2}\left(\mathbf{x}_{t+K}^u + \mathbf{x}_{t+K}^v\right)$. This is a standard approach in distributed optimization. However, this method does not apply to our setting, as we would need to assume that the stochastic noise of the oracles $G_u$ and $G_v$ is bounded.

We are considering a setting where the two players may not have access to their opponent's strategies or gradients, and only assume that the private components of the gradients have bounded variance, see Assumption 1. For this setting, we therefore propose that each player should only use the reliable information, that is $[G_u(\mathbf{x}, \xi)]_u$ for player $u$, and $[G_v(\mathbf{x}, \xi)]_v$ for player $v$. We can write our proposed method compactly as:

$$\mathbf{x}_K^r = \mathbf{x}_0^r - \gamma \sum_{t=0}^{K-1} G_0(\mathbf{x}_t^r, \xi_t) \quad \text{with} \quad G_0(\mathbf{x}_t^r, \xi) \equiv \begin{pmatrix} \nabla_u f(\mathbf{u}_t^r, \mathbf{v}_0^r; \xi) \\ -\nabla_v f(\mathbf{u}_0^r, \mathbf{v}_t^r; \xi) . \end{pmatrix} \quad \text{(decoupled-SGDA)}$$

Our operator has the property that $\mathbb{E}[G_0(\mathbf{x}, \xi)] = F_0(\mathbf{x})$ with bounded variance $\sigma^2$ where $F_0(\mathbf{x}) = (\nabla_u f(\mathbf{u}, \mathbf{v}_0), -\nabla_v f(\mathbf{u}_0, \mathbf{v}))$. Here, the index $t$ denotes the local update step in the current local update phase, and the superscript $r$ indexes the local phases. We also use $\mathbf{x}_0 \equiv (\mathbf{u}_0, \mathbf{v}_0)$ to denote the parameters of the players at the beginning of some round $r$. One communication round is needed for exchanging the updated parameters $(\mathbf{u}_K^r, \mathbf{v}_K^r)$ when passing to the next round. Next, we introduce one more assumption on operator $F_0$.

**Assumption 4** (Strong monotonicity). *Operator $F_0$ is strongly monotone with parameter $\mu_0 > 0$, i.e., for all $\mathbf{x}, \mathbf{x}' \in \mathcal{X}$, the following inequality holds:*

$$\langle F_0(\mathbf{x}) - F_0(\mathbf{x}'), \mathbf{x} - \mathbf{x}' \rangle \geqslant \mu_0 \|\mathbf{x} - \mathbf{x}'\|^2.$$

Note that one can show $\mu_0 = \min\{\frac{\mu_u}{\alpha}, \frac{\mu_v}{\beta}\}$ where $f$ is $\mu_u$-strongly convex in $\mathbf{u}$ and $\mu_v$-strongly-concave in $\mathbf{v}$ (see Lemma 6).

**Intuition.** To provide some intuition on why decoupled SGDA might work, consider that the objective of minimax games (SP) can be written as:

$$f(\mathbf{u}, \mathbf{v}) = g(\mathbf{u}) - h(\mathbf{v}) + r(\mathbf{u}, \mathbf{v}) \tag{4}$$

where $g(\mathbf{u})$ and $h(\mathbf{v})$ represent the independent contributions of each player, and $r(\mathbf{u}, \mathbf{v})$ captures the interaction between them.

In the special case when $r(\mathbf{u}, \mathbf{v}) \equiv 0$, i.e., there is no interaction, the problem does not require any communication: the optimal solution can be found by minimizing $g$ and $h$ separately. A method like SGDA is, therefore, not a good choice in this setting, as it requires communication in every step of the method, although this is unnecessary. If the contrast, when the coupling $r(\mathbf{u}, \mathbf{v})$ is significant, then optimizing $g$ and $h$ separately might not be a good strategy. Decoupled SGDA aims to find a balance between the two extremes. In the following, we will characterize some settings, where Decoupled SGDA provably uses significantly less communication rounds than SGDA, or other baselines (see also Table 1).

**Method.** We begin this section by providing details of our method. Decoupled SGDA has a round-wise update scheme allowing each player to share his parameters only once in a while. At the beginning of each round $r$, each player receives the most recent parameters of the other player. Then all players start taking $K$ local steps and updating **only** their own parameters using the information they received at the beginning of the round from other players. Note that our method is a general framework and one can use any first-order method to take local steps. As a baseline in this work, we consider simple GD updates. We formalize our method in Algorithm 1. For simplicity in notation, we now consider a two-player minimax game to motivate our method and highlight its differences from existing paradigms. The constants $\alpha, \beta$ are determined by the vector norm that we specify.

---

**Algorithm 1** Decoupled SGDA for two-player games

---

1: **Input:** step size $\gamma$, initialization $\mathbf{x}_0 = (\mathbf{u}_0, \mathbf{v}_0)$, $R$, $K$
2: **for** $r \in \{1, \ldots, R\}$ **do**
3:     **for** $t \in \{1, \ldots, K\}$ **do**
4:         Update local model $\mathbf{u}_{t+1}^r \leftarrow \mathbf{u}_t^r - \gamma(\alpha)^{-1}\nabla_u f(\mathbf{u}_t^r, \mathbf{v}_0^r; \xi)$
5:         Update local model $\mathbf{v}_{t+1}^r \leftarrow \mathbf{v}_t^r + \gamma(\beta)^{-1}\nabla_v f(\mathbf{u}_0^r, \mathbf{v}_t^r; \xi)$
6:     **end for**
7:     **Communicate** $(\mathbf{u}_K^r, \mathbf{v}_K^r)$ to each player
8: **end for**
9: **Output:** $\mathbf{x}_K^R = (\mathbf{u}_K^R, \mathbf{v}_K^R)$

---

**Extensions of Decoupled SGDA.** It is clear that our method is a general framework, providing flexibility for various modifications and adaptations. For instance, our method allows for any first-order update rule to be applied for the local steps like GDA, Extra Gradient (EG), and Optimistic Gradient Descent Ascent (OGDA). Note that in this work, we focused on GDA updates, leaving the analysis of other methods for future work. Moreover, in Section G, we present **Decoupled SGDA with Ghost Sequence**, where each player aims to estimate the other player's parameters using the so-called **Ghost Sequence**, which leads to further acceleration in terms of the number of rounds.

## 4 CONVERGENCE GUARANTEES

We first need to introduce the notion of **Weakly Coupled Games / Regime** and next we provide the convergence guarantee for our method.

**Definition 1** (Weakly Coupled and Fully Decoupled Games[1]). *Given a SCSC zero-sum minimax game $f(\mathbf{u}, \mathbf{v})$. We define the **coupling degree** parameter $\theta$ for this game as follows:*

$$\theta := \frac{L_c}{\mu_0}. \tag{5}$$

*This variable measures the level of interaction in the game. A smaller value of $\theta$ indicates less interaction. For any $f(\mathbf{u}, \mathbf{v})$, we say the game is **Weakly Coupled** if the following inequality holds:*

$$\frac{\theta}{c} \leqslant 1 \tag{6}$$

*where $c > 1$ is an absolute constant and will be specified based on the setting. We say the game is **Fully Decoupled** if we have $\theta = 0$ which implies $r(\mathbf{u}, \mathbf{v}) = 0$ (see Equation (4)).*

It's clear that fully decoupled games are the extreme case of weakly coupled games. In weakly coupled games, interaction between the two players is relatively minor compared to their individual self-interactions. This regime suggests that the influence of the $\mathbf{u}$ player on $\mathbf{v}$ (and vice versa) is sufficiently small, allowing the players' dynamics to be driven mainly by their own quadratic behavior, with minimal influence from interaction. In the fully decoupled games, the problem of finding the saddle point reduces to solving two independent minimization and maximization problems which is usually much easier and has been well-studied.

**Theorem 1.** *For any $R, K \geqslant \Omega\left(\frac{1}{\gamma\mu} \log\left(\frac{4}{\theta}\right)\right)$, after running Decoupled SGDA for a total of $T = KR$ iterations on a function $f$, with the stepsize $\gamma \leqslant \frac{\mu_0}{L^2}$ if the game is weakly coupled with $c = 4$ ($4\theta \leqslant 1$) or $\gamma \leqslant \min\left\{\frac{\mu}{L^2}, \frac{\mu}{KL\delta}, \frac{\mu}{K\delta^2}\right\}$ otherwise, we get a rate of:*

$$\mathbb{E}\big[\|\mathbf{x}_K^R - \mathbf{x}^\star\|^2\big] \leqslant D^2 \exp\left(-\max\left\{(1 - 4\theta)R, \frac{\gamma\mu}{2}KR\right\}\right) + \frac{\sigma^2\gamma}{\mu} \min\left\{\frac{8\theta}{1 - 4\theta}, 2\right\},$$

*where $\delta := \frac{L_c}{\sqrt{\alpha\beta}}$, $\sigma^2 := \sigma_{uu}^2 + \sigma_{vv}^2$, and $D = \|\mathbf{x}_0 - \mathbf{x}^\star\|$.*

**Corollary 2.** *Decoupled GDA with a stepsize of $\gamma = \frac{\mu_0}{L^2}$ converges to the saddle point **without** any communication on fully decoupled games ($\theta = 0$) if $K \to \infty$.*

The above result is an obvious case in which our method beats any first-order method that does not make use of local steps. For the sake of comparison, we define the condition numbers[2]: $\kappa_u = \frac{L_u}{\mu_u}$, $\kappa_v = \frac{L_v}{\mu_v}$ and $\kappa_{uv} = \kappa_{vu} = \frac{L_c}{\sqrt{\mu_u\mu_v}}$. Also we use $\kappa = \frac{L}{\mu}$. The most recent rate proposed for GDA Lee et al. (2024) needs $\mathcal{O}\left((\kappa_u + \kappa_v)\log(\frac{1}{\epsilon})\right)$ rounds of communication when the game is fully decoupled. A major drawback of GDA in this setting is that poor conditioning in one of the players (large $\kappa_u, \kappa_v$) increases the number of rounds significantly while our method overcomes this problem by utilizing local steps.

**Corollary 3.** *With the choice of $\gamma = \frac{\mu}{RL^2}$ if the game is weakly coupled and $\gamma = \min\left\{\frac{\mu}{32KL^2}, \frac{1}{\mu KR} \ln(\max\{2, \frac{\mu^2D^2}{\sigma^2}KR\})\right\}$ otherwise, we get*

$$\mathbb{E}\big[\|\mathbf{x}_K^R - \mathbf{x}^\star\|^2\big] \leqslant D^2 \exp\left(-\max\left\{(1 - 4\theta)R, \frac{\mu^2}{2L^2}R\right\}\right) + \sigma^2 \min\left\{\frac{8\theta}{RL^2(1 - 4\theta)}, \frac{1}{\mu^2KR}\right\}.$$

*Consequently, to reach $\mathbb{E}[\|\mathbf{x}_K^R - \mathbf{x}^\star\|^2] \leqslant \epsilon$, it suffices to perform $R = \max\{\frac{1}{4-\theta}\log(\frac{2D^2}{\epsilon}), \frac{16\theta}{\sigma^2L^2(1-4\theta)\epsilon}\}$ rounds with $K = \frac{L^2}{\mu\mu_0}\log(\frac{4}{\theta})$ if the game is weakly coupled, or $R = \frac{2L^2}{\mu^2}\log(\frac{D^2}{\epsilon})$ with $K = \frac{2}{\mu^2\epsilon}$ otherwise.*

---

[1]w.l.o.g. and for clarity, we assumed $\alpha = \beta = 1$. One can easily verify that $\theta := \frac{1}{\sqrt{\alpha\beta}} \cdot \frac{L_c}{\mu_0}$ in the general form (see Lemma 8).

[2]$\|\nabla_u f(\mathbf{u}, \mathbf{v}) - \nabla_u f(\mathbf{u}', \mathbf{v})\| \leqslant L_u\|\mathbf{u} - \mathbf{u}'\|$ and $\|\nabla_v f(\mathbf{u}, \mathbf{v}) - \nabla_v f(\mathbf{u}, \mathbf{v}')\| \leqslant L_v\|\mathbf{v} - \mathbf{v}'\|$

| Method | Communication Complexity *(Fully Decoupled)* | Communication Complexity *(General Bound)* | Speed Up |
|---|---|---|---|
| GDA Lee et al. (2024) | $(\kappa_u + \kappa_v)\log\frac{1}{\epsilon}$ | $(\kappa_u + \kappa_v + \kappa_{uv}^2)\log\frac{1}{\epsilon}$ | $\kappa_c \leqslant \frac{1}{4}$ (weakly coupled) |
| EG/OGDA Mokhtari et al. (2020) | $(\kappa_u + \kappa_v)\log\frac{1}{\epsilon}$ | $(\kappa_u + \kappa_v)\log\frac{1}{\epsilon}$ | $\kappa_c \leqslant \frac{1}{2}\sqrt{1 - \frac{1}{\max\{\kappa_u \kappa_v\}}}$ |
| APPA Lin et al. (2020) | $\sqrt{\kappa_u \kappa_v}\log^3\frac{1}{\epsilon}$ | $\sqrt{\kappa_u \kappa_v}\log^3\frac{1}{\epsilon}$ | $\kappa_c \leqslant \frac{1}{2}\sqrt{1 - \frac{1}{\sqrt{\kappa_u \kappa_v}}}$ |
| FOAM Kovalev & Gasnikov (2022) | $\sqrt{\kappa_u \kappa_v}\log\frac{1}{\epsilon}$ | $\sqrt{\kappa_u \kappa_v}\log\frac{1}{\epsilon}$ | $\kappa_c \leqslant \frac{1}{2}\sqrt{1 - \frac{1}{\sqrt{\kappa_u \kappa_v}}}$ |
| **Decoupled SGDA (ours)** | **0** | $\min\left\{\frac{1}{1-4\kappa_c}\log\frac{1}{\epsilon},\ \kappa^2\log\frac{1}{\epsilon}\right\}$ | — |

Table 1: Comparison of communication complexity (rounds complexity—ours, vs. iteration complexity–other methods') and the acceleration condition. *Speed up* lists the conditions under which Decoupled SGDA achieves acceleration relative to the respective method.

**Linear convergence.** Our rate benefits from an exponential convergence in terms of the number of rounds. The main feature of Theorem 1 is the **absence** of $\kappa_u$, $\kappa_v$, or $\kappa$ in the weakly coupled regime. This is particularly interesting because the condition number for each player can be extremely large, which would typically comes with a high communication overhead. However, in the weakly coupled regime, our rate depends on the coupling degree $\theta$ which can be very small or even zero.

**Noise term.** Another important aspect of our rate is its dependency on $\sigma$ in the noise term. As discussed in Assumption 1, one can assume that each player has access to a very accurate gradient oracle with respect to its own parameters, while a very noisy oracle is used to access the gradient with respect to the other player's parameters. Our method has **no** dependency on $\sigma_{uv}$ or $\sigma_{vu}$, which can be much larger than $\sigma$. In addition, the noise term is multiplied by the factor $\frac{\theta}{1-4\theta}$, which goes to zero when $\theta = 0$. Note that in this case, based on the lower bound $\Omega\left(\frac{1}{\gamma\mu}\ln\left(\frac{4}{\theta}\right)\right)$, we know that $K \to \infty$, which is expected as we also have a linear speed-up in terms of the number of local steps in the non-weakly coupled regime. Now, we state the communication complexity of our method:

**Corollary 4.** *For any $K \geqslant \Omega\left(\frac{1}{\gamma\mu}\ln\left(\frac{4}{\theta}\right)\right)$, after running Decoupled SGDA on a weakly coupled game with $c = 4$, we have the following communication complexity in order to achieve $\epsilon$ accuracy in the noiseless setting:*

$$\overbrace{\boxed{R = \mathcal{O}\left(\frac{1}{1-4\theta}\log\frac{1}{\epsilon}\right)}}^{\text{Decoupled GDA}} \quad \textbf{vs.} \quad \overbrace{\boxed{R = \mathcal{O}\left((\kappa_u + \kappa_v + \kappa_{uv}^2)\log\frac{1}{\epsilon}\right)}}^{\text{GDA}}$$

*Moreover, Decoupled SGDA in weakly coupled regime has **always** a better communication complexity compared to the baseline GDA. In another word, $\frac{1}{1-4\theta} \leqslant \kappa_u + \kappa_v + \kappa_{uv}^2$.*

Table 1 compares our method with other first-order methods in terms of **communication complexity** in both the fully decoupled and weakly coupled regimes. It is clear that in the fully decoupled regime, our method outperforms all other methods. Furthermore, it is expected to compare our method with GDA by considering it as the baseline because our method uses GD local updates (and not updates using EG or momentum). In Corollary 4, we stated that we always have a better complexity compared to GDA in the weakly coupled regime. However, we can show that under a slightly stronger assumption, our method achieves better communication complexity than the optimal first-order method for solving SCSC games.

**Corollary 5.** *For any SCSC zero-sum minimax game with coupling degree $\theta \leqslant \frac{1}{2}\sqrt{1 - \frac{1}{\sqrt{\kappa_u \kappa_v}}}$, our method achieves a better communication complexity than FOAM which is the optimal first-order method for solving SCSC games. In another word, if $\frac{1}{1-4\theta} \ll \sqrt{\kappa_u \kappa_v}$, our method achieves significant communication acceleration compared to FOAM.*

Corollary 5 shows our method can outperform the optimal first-order method in terms of the number of communication rounds. The assumption $\theta \leqslant \frac{1}{2}\sqrt{1 - \frac{1}{\sqrt{\kappa_u \kappa_v}}}$ is stronger than the weakly coupled assumption. One can verify that in the limiting case when $\max\{\kappa_u, \kappa_v\} \to \infty$, which means either one or both players have very poor conditioning, this assumption reduces to $\theta \leqslant \frac{1}{2}$, which is the definition of a weakly coupled game with $c = 2$. The main drawback of all existing methods is that they do not utilize the fact that the interactive part of the game might have a minor effect. For instance, the communication complexity of two popular methods, EG and OGDA, is given as $\mathcal{O}\left(\kappa \log\left(\frac{1}{\epsilon}\right)\right)$, as proposed in Mokhtari et al. (2020), where $\kappa$ depends on $\max\{L_u, L_v, L_c\}$, which is too pessimistic when the interaction between players is low. Even the method proposed by Lin et al. (2020), with complexity $\mathcal{O}\left(\sqrt{\kappa_u \kappa_v} \log^3\left(\frac{1}{\epsilon}\right)\right)$, which is near-optimal, and Kovalev & Gasnikov (2022), with complexity $\mathcal{O}\left(\sqrt{\kappa_u \kappa_v} \log\left(\frac{1}{\epsilon}\right)\right)$, which is the optimal method, match the lower bound of $\Omega\left(\sqrt{\kappa_u \kappa_v} \log\left(\frac{1}{\epsilon}\right)\right)$ proposed by Zhang et al. (2022b), have dependencies on $\kappa_u$ and $\kappa_v$ that can be significantly large. However, our rate depends on the quantity $\theta$, which can be very small or even zero when the interaction between players is low.

## 5 EXPERIMENTS

In this section, we evaluate the empirical performance of Decoupled SGDA. For all experiments described herein, we provide additional implementation details (and hyperparameters) in Section I.

**Quadratic Games**    Herein, we consider the following problem class:

$$\min_{\mathbf{u}} \max_{\mathbf{v}} \quad \frac{1}{2}\langle \mathbf{u}, \mathbf{Au} \rangle - \frac{1}{2}\langle \mathbf{v}, \mathbf{Bv} \rangle + \langle \mathbf{u}, \mathbf{Cv} \rangle, \tag{QG}$$

where $\mathbf{u}, \mathbf{v} \in \mathbb{R}^{\frac{d}{2}}$, and $\mathbf{A}, \mathbf{B}, \mathbf{C}$ are $\frac{d}{2} \times \frac{d}{2}$ positive definite matrices. We will use varying $\mathbf{C}$ to control the players' interaction.

Figure 1 illustrates the performances of Decoupled-SGDA on the (QG) for varying numbers of local steps $K$ and different intensities of the interactive term of (QG). The results show that as the interactive term weakens, Decoupled SGDA converges more quickly than the GDA baseline ($K = 1$). Additionally, with a stronger interactive term, increasing the number of local steps $K$ leads to faster convergence for the same number of synchronization rounds. Figure 2 depicts the performances over a spectrum of payoff functions controlled by the constant matrix $\mathbf{C}$ in (QG). In the Weakly Coupled Game regime, highlighted by shading, Decoupled SGDA outperforms the baseline GDA. In Figure 2 (right), we compare it with other optimization methods, demonstrating that Decoupled SGDA achieves similar results with significantly fewer communication rounds in the weakly coupled regime.

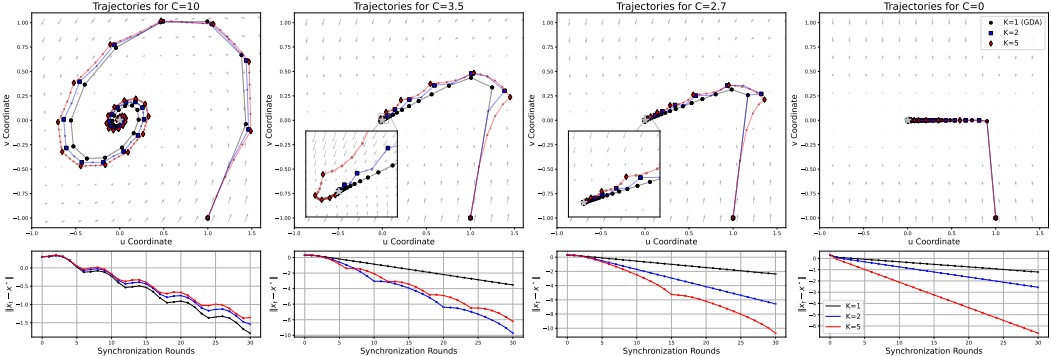

Figure 1: **Trajectories (top row) and distance to equilibrium over synchronization rounds (bottom row) of GDA ($K = 1$) and Decoupled SGDA with $K = \{2, 5\}$ on the** (QG) **problem ($d = 2$). C** in (QG) is a constant here—the larger, the stronger the interactive term. **Left-to-right:** decreasing the constant $c \in \{10, 3.5, 2, 7, 0\}$. The markers denote the local steps and star the solution. See § 5 for discussion.

**Communication Efficiency For Non-convex Functions**    While our theoretical focus was on SCSC games, in this section, we explore if our insights extend to broader problem instances. We

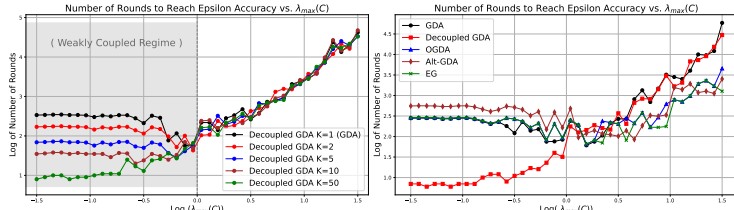

Figure 2: **Number of rounds (log-scale; lower is better) to reach epsilon accuracy for varying $\lambda_{\mathbf{max}}(\mathbf{C})$ in** (QG). **Left:** Decoupled GDA with different $K$-values and GDA ($K = 1$). **Right:** comparison between GDA, Decoupled GDA, Optimistic GDA (Popov, 1980), ALT–alternating GDA and Extragradient (Korpelevich, 1976).

focus on a *Toy GAN* non-convex game as follows:

$$\min_{\mathbf{u}} \max_{\mathbf{v}} \left\{ \mathbb{E}_{\phi \sim \mathcal{N}(0,\Sigma)}[\phi^T \mathbf{v}\phi] - \mathbb{E}_{\phi \sim \mathcal{N}(0,1)}[(\mathbf{u}\phi)^T \mathbf{v}(\mathbf{u}\phi)] + \lambda_1 \|\mathbf{u}\|^2 - \lambda_2 \|\mathbf{v}\|^2 \right\}, \quad \text{(toyGAN)}$$

where $\mathbf{u} \in \mathbb{R}^{d_1}$, $\mathbf{v} \in \mathbb{R}^{d_2}$.

Figure 3 shows the smallest gradient norm (lower is better) each algorithm can achieve for a fixed number of communication rounds, with varying values of $1/\lambda$. As $\lambda$ decreases, the regularization terms dominate, making the game less interactive (similar to the weakly coupled regime). When $\lambda$ increases, reducing interaction, Decoupled GDA achieves a much lower gradient norm with the same number of communication rounds. This demonstrates that Decoupled GDA efficiently solves non-convex problems in settings analogous to the weakly coupled regime by leveraging local updates to reduce communication. This experiment highlights the method's capabilities beyond SCSC games. The trajectory of Decoupled GDA iterations for this non-convex minimax problem can be found in Appendix H.1.

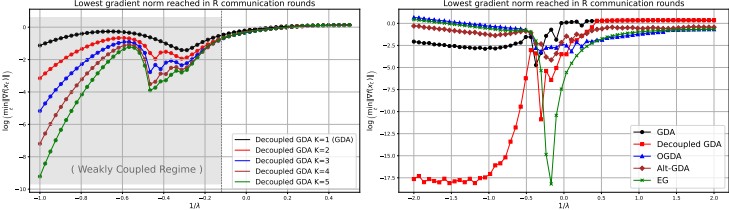

Figure 3: **Lowest gradient norm reached after a fixed number of communication rounds, for varying $1/\lambda$ in** (toyGAN). **Left:** Effect of $K$. **Right:** different optimization methods, GDA, Decoupled GDA, Optimistic GDA (Popov, 1980), ALT–alternating GDA and Extragradient (Korpelevich, 1976). See § 5 for discussion.

**Decoupled SGDA with gradient approximation**  Herein, we compare *Decoupled SGDA* with Federated Minimax, aka (local-SGDA). We focus on environments with gradient oracles with *unbalanced noise*. Each player has access to a gradient oracle that provides low-variance noise for their own gradients but high-variance noise for the remaining players. This reflects real-world challenges where shared information can be unreliable. In such cases, it is more effective to wait for synchronization for accurate updates, and between these periods, players should update only their own parameters. In particular, we focus on a modified quadratic game (QG) we used earlier, with each oracle adding zero-mean Gaussian noise to the full gradient. The variance of noise differs between gradients related to a player's own strategy (diagonal variance) and those related to other players' strategies (off-diagonal variance). Refer to Assumption 8 for a more rigorous definition. In both experiments, diagonal variance was fixed at 0.1, while off-diagonal variance was linearly increased from 1 to 10 in the second experiment.

Figure 4 illustrates the performances of *Decoupled SGDA* and *Local SGDA*, the latter being the most commonly used method for federated minimax problems (Deng & Mahdavi, 2021). It depicts the smallest gradient norm each algorithm achieves within a fixed number of communication rounds across different scenarios. The left plot demonstrates how both methods perform in games with varying levels of interaction. When the interaction is weaker, Decoupled SGDA achieves significantly lower gradient norms with the same number of communication rounds. The right plot highlights the

effect of noise variance, showing that while high noise negatively impacts Local SGDA, it has minimal to no effect on Decoupled SGDA. In the presence of imbalanced noise, the results suggest that switching from local SGDA to Decoupled SGDA is beneficial, even for highly interactive games.

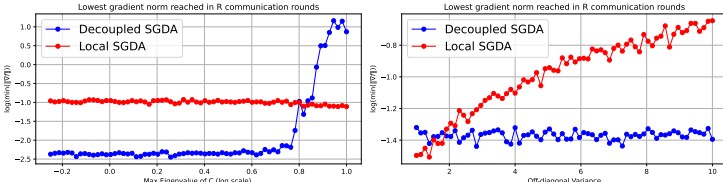

Figure 4: **Lowest gradient norm achieved by Decoupled SGDA and Local SGDA under a fixed number of communication rounds with unbalanced noisy gradient oracles. Left:** Comparison of *Decoupled SGDA* and *Federated Minimax* across varying values of $\|\mathbf{C}\|$. **Right:** Comparison of *Decoupled SGDA* and *Local SGDA* under different levels of off-diagonal variance noise. Refer to Section H.2 for detailed figures.

**Communication Efficiency in GAN Training**  Figure 5 compares Decoupled SGDA with baseline methods in terms of FID score reduction over several communication rounds. The plots show that Decoupled SGDA converges more quickly and requires fewer communication rounds compared to standard GDA and its variants. This is especially noticeable in the CIFAR-10 and SVHN datasets, where increasing the number of local steps (K) results in lower FID scores, demonstrating the efficiency of our approach in reducing communication while maintaining strong performance in complex, non-convex tasks like GAN training.

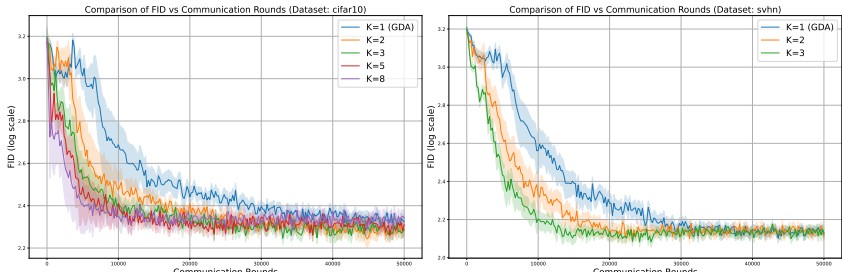

Figure 5: $y$-**axis: FID scores (log scale; lower is better) during GAN training, versus** $x$-**axis communication rounds. Left:** results on the *CIFAR-10* (Krizhevsky, 2009) dataset. **Right:** results on the *SVHN* (Netzer et al., 2011) dataset.

## 6 CONCLUSION

We proposed *Decoupled SGDA* as an effective optimization method for games with intermittent strategy communication, particularly in scenarios where interaction between players is weak, or noise levels are high. Through extensive theoretical and empirical analysis, we demonstrate that Decoupled SGDA not only outperforms traditional methods like Local SGDA in terms of communication efficiency and robustness in weakly coupled games but also extends its benefits beyond SCSC games to non-convex settings. The method's ability to handle varying levels of interaction and noise makes it highly adaptable, providing a valuable tool for federated and decentralized optimization problems.

Several future directions are possible. One can consider varying $K$ per player, extensions of other game optimization methods, such as extragradient, for classes beyond players having strongly convex utilities, among others. In addition, the proposed approach has the potential to address privacy-sensitive scenarios, as players can update their parameters independently without needing direct access to others' parameters, minimizing privacy risks posed by gradient sharing (see Zhu et al., 2019; Zhao et al., 2020; Wei et al., 2020, and references therein). Future work could further explore this potential in privacy-preserving applications, making Decoupled SGDA a valuable tool for decentralized optimization under privacy constraints.

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

# Table of Contents

# A   Summary of Parameters

Table 2: Definitions of $L$ and $\mu$ Terms for Two-Player Games

| Symbol | Definition | Mathematical Definition |
|:---:|:---:|:---:|
| $L$ | Smoothness parameter for operator $F(\mathbf{x})$ | $\|F(\mathbf{x}) - F(\mathbf{x}')\|_* \leqslant L \|\mathbf{x} - \mathbf{x}'\|$ |
| $L_u$ | Smoothness parameter with respect to $\mathbf{u}$ | $\|\nabla_u f(\mathbf{u}, \mathbf{v}) - \nabla_u f(\mathbf{u}', \mathbf{v})\|_{u,*} \leqslant L_u \|\mathbf{u} - \mathbf{u}'\|_u$ |
| $L_v$ | Smoothness parameter with respect to $\mathbf{v}$ | $\|\nabla_v f(\mathbf{u}, \mathbf{v}) - \nabla_v f(\mathbf{u}, \mathbf{v}')\|_{v,*} \leqslant L_v \|\mathbf{v} - \mathbf{v}'\|_v$ |
| $L_c$ | Interaction smoothness parameter | $\|\nabla_u f(\mathbf{u}, \mathbf{v}) - \nabla_u f(\mathbf{u}, \mathbf{v}')\|_{u,*} \leqslant L_c \|\mathbf{v} - \mathbf{v}'\|_v$ |
| $L_c$ | Interaction smoothness parameter | $\|\nabla_v f(\mathbf{u}, \mathbf{v}) - \nabla_v f(\mathbf{u}', \mathbf{v})\|_{v,*} \leqslant L_c \|\mathbf{u} - \mathbf{u}'\|_u$ |
| $\mu_u$ | Strong convexity parameter for $\mathbf{u}$ | $f(\mathbf{u}', \mathbf{v}) \geqslant f(\mathbf{u}, \mathbf{v}) + \langle \nabla_u f(\mathbf{u}, \mathbf{v}), \mathbf{u}' - \mathbf{u} \rangle + \frac{\mu_u}{2} \|\mathbf{u}' - \mathbf{u}\|_u^2$ |
| $\mu_v$ | Strong concavity parameter for $\mathbf{v}$ | $f(\mathbf{u}, \mathbf{v}') \leqslant f(\mathbf{u}, \mathbf{v}) + \langle \nabla_v f(\mathbf{u}, \mathbf{v}), \mathbf{v}' - \mathbf{v} \rangle - \frac{\mu_v}{2} \|\mathbf{v}' - \mathbf{v}\|_v^2$ |
| $\mu_0$ | Strong monotonicity parameter for $F(\mathbf{x})$ | $\langle F_0(\mathbf{x}) - F_0(\mathbf{x}'), \mathbf{x} - \mathbf{x}' \rangle \geqslant \mu_0 \|\mathbf{x} - \mathbf{x}'\|^2$ |
| $\mu$ | Strong monotonicity parameter for $F(\mathbf{x})$ | $\langle F(\mathbf{x}) - F(\mathbf{x}'), \mathbf{x} - \mathbf{x}' \rangle \geqslant \mu \|\mathbf{x} - \mathbf{x}'\|^2$ |

Table 3: Definitions of $L$ and $\mu$ Terms for $N$-Player Games

| Symbol | Definition | Mathematical Definition |
|:---:|:---:|:---:|
| $\hat{L}_i$ | Upper bound for diagonal elements $L_{ii}$ | $L_{ii} \leqslant \hat{L}_i$, refer to Matrix(16) |
| $\bar{L}_i$ | Upper bound for off-diagonal elements $L_{ij}$ for $i \neq j$ | $L_{ij} \leqslant \bar{L}_i$, refer to Matrix(16) |
| $L$ | Smoothness parameter for operator $F(\mathbf{x})$ | $\|F(\mathbf{x}) - F(\mathbf{x}')\|_* \leqslant L \|\mathbf{x} - \mathbf{x}'\|$ |
| $\mu_{\min}$ | Minimum strong convexity/concavity parameter | $\mu_{\min} := \min_{1 \leqslant i \leqslant N}\{\frac{\mu_i}{\alpha_i}\}$ |
| $\mu$ | Strong monotonicity parameter for $F(\mathbf{x})$ | $\langle F(\mathbf{x}) - F(\mathbf{x}'), \mathbf{x} - \mathbf{x}' \rangle \geqslant \mu \|\mathbf{x} - \mathbf{x}'\|^2$ |

# B   Missing Proofs for Section 4

**Lemma 6.** *The operator $F_0$ defined in* (decoupled-SGDA) *is $\mu_0$-strongly monotone where $\mu_0$ can be expressed as:*

$$\mu_0 = \min\left\{\frac{\mu_u}{\alpha}, \frac{\mu_v}{\beta}\right\}. \tag{7}$$

*Proof.* Recall that the function $f$ is $\mu_u$ strongly convex in $\mathbf{u}$ and $\mu_v$ strongly concave in $\mathbf{v}$ meaning that:

$$f(\mathbf{u}', \mathbf{v}) \geqslant f(\mathbf{u}, \mathbf{v}) + \langle \nabla_u f(\mathbf{u}, \mathbf{v}), \mathbf{u}' - \mathbf{u} \rangle + \frac{\mu_u}{2} \|\mathbf{u}' - \mathbf{u}\|_u^2$$

$$f(\mathbf{u}, \mathbf{v}') \leqslant f(\mathbf{u}, \mathbf{v}) + \langle \nabla_v f(\mathbf{u}, \mathbf{v}), \mathbf{v}' - \mathbf{v} \rangle - \frac{\mu_v}{2} \|\mathbf{v}' - \mathbf{v}\|_v^2$$

Next we have:

$$\langle F_0(\mathbf{x}) - F_0(\mathbf{x}'), \mathbf{x} - \mathbf{x}' \rangle$$
$$= \langle \nabla_u f(\mathbf{u}, \mathbf{v}_0) - \nabla_u f(\mathbf{u}', \mathbf{v}_0), \mathbf{u} - \mathbf{u}' \rangle + \langle \nabla_v f(\mathbf{u}_0, \mathbf{v}) - \nabla_v f(\mathbf{u}_0, \mathbf{v}'), \mathbf{v}' - \mathbf{v} \rangle$$
$$\geqslant \mu_u \|\mathbf{u} - \mathbf{u}'\|_u^2 + \mu_v \|\mathbf{v} - \mathbf{v}'\|_v^2 = \frac{\mu_u}{\alpha} \alpha \|\mathbf{u} - \mathbf{u}'\|_u^2 + \frac{\mu_v}{\beta} \beta \|\mathbf{v} - \mathbf{v}'\|_v^2$$
$$\geqslant \min\left\{\frac{\mu_u}{\alpha}, \frac{\mu_v}{\beta}\right\} \|\mathbf{x} - \mathbf{x}'\|^2. \qquad \square$$

**Lemma 7** (two-player). *For each $x \in \mathcal{X}$ and $\delta := \frac{L_c}{\sqrt{\alpha\beta}}$, we have*

$$\|F(\mathbf{x}) - F_0(\mathbf{x})\|_* \leqslant \delta \|\mathbf{x}_0 - \mathbf{x}\|.$$

*Proof.* Indeed,

$$\|F(\mathbf{x}) - F_0(\mathbf{x})\|_*^2 = \frac{1}{\alpha} \|\nabla_u f(\mathbf{u}, \mathbf{v}) - \nabla_u f(\mathbf{u}, \mathbf{v}_0)\|_{u,*}^2 + \frac{1}{\beta} \|\nabla_v f(\mathbf{u}, \mathbf{v}) - \nabla_v f(\mathbf{u}, \mathbf{v}_0)\|_{v,*}^2$$

$$\leqslant \frac{L_c^2}{\alpha} \|\mathbf{v} - \mathbf{v}_0\|_v^2 + \frac{L_c^2}{\beta} \|\mathbf{u} - \mathbf{u}_0\|_u^2$$

$$= \frac{L_c^2}{\beta\alpha} \beta \|\mathbf{v} - \mathbf{v}_0\|_v^2 + \frac{L_c^2}{\alpha\beta} \alpha \|\mathbf{u} - \mathbf{u}_0\|_u^2$$

$$\leqslant \max\left\{\frac{L_c^2}{\beta\alpha}, \frac{L_c^2}{\alpha\beta}\right\} \left[\beta \|\mathbf{v} - \mathbf{v}_0\|_v^2 + \alpha \|\mathbf{u} - \mathbf{u}_0\|_u^2\right] = \delta^2 \|\mathbf{x}_0 - \mathbf{x}\|^2. \qquad \square$$

**Lemma 8** (two-player). *Let* $\mathbf{x}', \mathbf{x}^\star \in \mathcal{X}$ *be such that* $F_0(\mathbf{x}') = 0$ *and* $F(\mathbf{x}^\star) = 0$. *Then,*

$$\|\mathbf{x}' - \mathbf{x}^\star\| \leqslant \theta \|\mathbf{x}_0 - \mathbf{x}^\star\|. \tag{8}$$

*Proof.* Recall that the function $f$ is $\mu_u$ strongly convex in $\mathbf{u}$ and $\mu_v$ strongly concave in $\mathbf{v}$ meaning that:

$$f(\mathbf{u}', \mathbf{v}) \geqslant f(\mathbf{u}, \mathbf{v}) + \langle \nabla_u f(\mathbf{u}, \mathbf{v}), \mathbf{u}' - \mathbf{u} \rangle + \frac{\mu_u}{2} \|\mathbf{u}' - \mathbf{u}\|_u^2$$

$$f(\mathbf{u}, \mathbf{v}') \leqslant f(\mathbf{u}, \mathbf{v}) + \langle \nabla_v f(\mathbf{u}, \mathbf{v}), \mathbf{v}' - \mathbf{v} \rangle - \frac{\mu_v}{2} \|\mathbf{v}' - \mathbf{v}\|_v^2$$

Next we have:

$$\|\mathbf{x}' - \mathbf{x}^\star\|^2 = \alpha \|\mathbf{u}' - \mathbf{u}^\star\|_u^2 + \beta \|\mathbf{v}' - \mathbf{v}^\star\|_v^2$$

$$\leqslant \frac{\alpha}{\mu_u^2} \|\nabla_u f(\mathbf{u}', \mathbf{v}_0) - \nabla_u f(\mathbf{u}^\star, \mathbf{v}_0)\|_{u,*}^2 + \frac{\beta}{\mu_v^2} \|\nabla_v f(\mathbf{u}_0, \mathbf{v}') - \nabla_v f(\mathbf{u}_0, \mathbf{v}^\star)\|_{v,*}^2$$

$$= \frac{\alpha}{\mu_u^2} \|\nabla_u f(\mathbf{u}^\star, \mathbf{v}^\star) - \nabla_u f(\mathbf{u}^\star, \mathbf{v}_0)\|_{u,*}^2 + \frac{\beta}{\mu_v^2} \|\nabla_v f(\mathbf{u}^\star, \mathbf{v}^\star) - \nabla_v f(\mathbf{u}_0, \mathbf{v}^\star)\|_{v,*}^2$$

$$\leqslant \frac{\alpha L_c^2}{\mu_u^2} \|\mathbf{v}_0 - \mathbf{v}^\star\|_v^2 + \frac{\beta L_c^2}{\mu_v^2} \|\mathbf{u}_0 - \mathbf{u}^\star\|_u^2$$

$$= \frac{\alpha L_c^2}{\beta\mu_u^2} \beta \|\mathbf{v}_0 - \mathbf{v}^\star\|_v^2 + \frac{\beta L_c^2}{\alpha\mu_v^2} \alpha \|\mathbf{u}_0 - \mathbf{u}^\star\|_u^2$$

$$= \frac{L_c^2}{\beta\alpha\frac{\mu_u^2}{\alpha^2}} \beta \|\mathbf{v}_0 - \mathbf{v}^\star\|_v^2 + \frac{L_c^2}{\alpha\beta\frac{\mu_v^2}{\beta^2}} \alpha \|\mathbf{u}_0 - \mathbf{u}^\star\|_u^2$$

$$\leqslant \frac{L_c^2}{\alpha\beta\mu_0^2} \beta \|\mathbf{v}_0 - \mathbf{v}^\star\|_v^2 + \frac{L_c^2}{\alpha\beta\mu_0^2} \alpha \|\mathbf{u}_0 - \mathbf{u}^\star\|_u^2$$

$$\leqslant \frac{1}{\alpha\beta} \frac{L_c^2}{\mu_0^2} \left[\beta \|\mathbf{v}_0 - \mathbf{v}^\star\|_v^2 + \alpha \|\mathbf{u}_0 - \mathbf{u}^\star\|_u^2\right] = \theta^2 \|\mathbf{x}_0 - \mathbf{x}^\star\|^2. \qquad \square$$

where we used the fact that $\mu_0 \leqslant \frac{\mu_u}{\alpha}$ and $\mu_0 \leqslant \frac{\mu_v}{\beta}$ from Lemma 6.

### B.1    PROOF OF THEOREM 1

We start with some auxiliary lemmas.

**Lemma 9** (**Consensus error**). *After running Decoupled SGDA for $K$ local steps at some round $r$ with a step-size of $\gamma \leqslant \frac{\mu}{32\delta LK}$, the consensus error can be upper bounded as follows:*

$$\mathbb{E} \|\mathbf{x}_{t+1} - \mathbf{x}_0\|^2 \leqslant \sum_{i=t+1-K}^{t} \frac{\mu^2}{64K\delta^2} \mathbb{E} \|\mathbf{x}_i - \mathbf{x}^\star\|^2 + 4K\gamma^2\sigma^2 \tag{9}$$

*Proof.*

$$\mathbb{E}\left\|\mathbf{x}_{t+1} - \mathbf{x}_0\right\|^2$$

$$= \mathbb{E}\left\|\mathbf{x}_t - \gamma\mathbf{B}^{-1}G_0(\mathbf{x}_t, \xi) - \mathbf{x}_0\right\|^2$$

$$\leqslant \mathbb{E}\left\|\mathbf{x}_t - \gamma\mathbf{B}^{-1}F_0(\mathbf{x}_t) - \mathbf{x}_0\right\|^2 + \gamma^2$$

$$\leqslant \left(1 + \frac{1}{K}\right)\mathbb{E}\left\|\mathbf{x}_t - \mathbf{x}_0\right\|^2 + 2K\gamma^2\mathbb{E}\left\|F_0(\mathbf{x}_t)\right\|_*^2 + \gamma^2\sigma^2$$

$$\leqslant \left(1 + \frac{1}{K}\right)\mathbb{E}\left\|\mathbf{x}_t - \mathbf{x}_0\right\|^2 + 2K\gamma^2\mathbb{E}\left\|F_0(\mathbf{x}_t) - F(\mathbf{x}_t) + F(\mathbf{x}_t)\right\|_*^2 + \gamma^2\sigma^2$$

$$\leqslant \left(1 + \frac{1}{K}\right)\mathbb{E}\left\|\mathbf{x}_t - \mathbf{x}_0\right\|^2 + 4K\gamma^2\mathbb{E}\left\|F_0(\mathbf{x}_t) - F(\mathbf{x}_t)\right\| + 4K\gamma^2\mathbb{E}\left\|F(\mathbf{x}_t)\right\|_*^2 + \gamma^2\sigma^2$$

$$\leqslant \left(1 + \frac{1}{K}\right)\mathbb{E}\left\|\mathbf{x}_t - \mathbf{x}_0\right\|^2 + 4K\delta^2\gamma^2\mathbb{E}\left\|\mathbf{x}_t - \mathbf{x}_0\right\|^2 + 4KL^2\gamma^2\mathbb{E}\left\|\mathbf{x}_t - \mathbf{x}^\star\right\|^2 + \gamma^2\sigma^2$$

With the choice of $\gamma \leqslant \frac{\mu}{32K\delta L}$ where $\delta := \frac{L_c}{\sqrt{\alpha\beta}}$, we get:

$$\mathbb{E}\left\|\mathbf{x}_{t+1} - \mathbf{x}_0\right\|^2$$

$$\leqslant \left(1 + \frac{1}{K}\right)\mathbb{E}\left\|\mathbf{x}_t - \mathbf{x}_0\right\|^2 + \frac{\mu^2}{256KL^2}\mathbb{E}\left\|\mathbf{x}_t - \mathbf{x}_0\right\|^2 + \frac{\mu^2}{256K\delta^2}\mathbb{E}\left\|\mathbf{x}_t - \mathbf{x}^\star\right\|^2 + \gamma^2\sigma^2$$

$$\leqslant \left(1 + \frac{1}{K}\right)\mathbb{E}\left\|\mathbf{x}_t - \mathbf{x}_0\right\|^2 + \frac{1}{256K}\mathbb{E}\left\|\mathbf{x}_t - \mathbf{x}_0\right\|^2 + \frac{\mu^2}{256K\delta^2}\mathbb{E}\left\|\mathbf{x}_t - \mathbf{x}^\star\right\|^2 + \gamma^2\sigma^2$$

$$\leqslant \left(1 + \frac{1}{K} + \frac{1}{256K}\right)\mathbb{E}\left\|\mathbf{x}_t - \mathbf{x}_0\right\|^2 + \frac{\mu^2}{256K\delta^2}\mathbb{E}\left\|\mathbf{x}_t - \mathbf{x}^\star\right\|^2 + \gamma^2\sigma^2$$

By unrolling the recursion for the last $K$ steps and considering the fact that $\left(1 + \frac{1}{K} + \frac{1}{256K}\right)^K \leqslant 4$ we get:

$$\mathbb{E}\left\|\mathbf{x}_{t+1} - \mathbf{x}_0\right\|^2 \leqslant \sum_{i=t+1-K}^{t}\frac{\mu^2}{64K\delta^2}\mathbb{E}\left\|\mathbf{x}_i - \mathbf{x}^\star\right\|^2 + 4K\gamma^2\sigma^2$$

$\square$

**Lemma 10.** *Let* $\mathbf{x}' = (\mathbf{u}', \mathbf{v}')$ *where* $\mathbf{u}' = \arg\min_{\mathbf{u}} f(\mathbf{u}, \mathbf{v}_0)$ *and* $\mathbf{v}' = \arg\max_{\mathbf{v}} f(\mathbf{u}_0, \mathbf{v})$. *Starting from* $(\mathbf{u}_0, \mathbf{v}_0)$, *we upper bound the distance to* $\mathbf{x}'$ *after* $K$ *local steps as follows:*

$$\left\|\mathbf{x}_{t+1} - \mathbf{x}'\right\|^2 \leqslant (1 - \gamma\mu_0)^K\mathbb{E}\left\|\mathbf{x}_0 - \mathbf{x}'\right\|^2 + \frac{\gamma\sigma^2}{\mu_0} \tag{10}$$

*Proof.*

$$\left\|\mathbf{x}_{t+1} - \mathbf{x}'\right\|^2$$

$$= \left\|\mathbf{x}_t - \gamma\mathbf{B}^{-1}G_0(\mathbf{x}_t, \xi_t) - \mathbf{x}'\right\|^2$$

$$= \left\|\mathbf{x}_t - \mathbf{x}'\right\|^2 + \gamma^2\left\|G_0(\mathbf{x}_t, \xi)\right\|_*^2 - 2\gamma\langle G_0(\mathbf{x}_t, \xi_t), \mathbf{x}_t - \mathbf{x}'\rangle$$

$$= \left\|\mathbf{x}_t - \mathbf{x}'\right\|^2 + \gamma^2\left\|G_0(\mathbf{x}_t, \xi) - F_0(\mathbf{x}_t) + F_0(\mathbf{x}_t)\right\|_*^2 - 2\gamma\langle G_0(\mathbf{x}_t, \xi) - F_0(\mathbf{x}_t) + F_0(\mathbf{x}_t), \mathbf{x}_t - \mathbf{x}'\rangle$$

By taking the conditional expectation on previous iterates we have:

$$\mathbb{E}_{\xi_t}\left\|\mathbf{x}_{t+1} - \mathbf{x}'\right\|^2$$

$$\leqslant \left\|\mathbf{x}_t - \mathbf{x}'\right\|^2 + \gamma^2\left\|F_0(\mathbf{x}_t) - F_0(\mathbf{x}')\right\|_*^2 - 2\gamma\langle F_0(\mathbf{x}_t) - F_0(\mathbf{x}'), \mathbf{x}_t - \mathbf{x}'\rangle + \sigma^2$$

$$\leqslant (1 + \gamma^2L^2 - 2\gamma\mu_0)\left\|\mathbf{x}_t - \mathbf{x}'\right\|^2 + \gamma^2\sigma^2$$

With the choice of $\gamma \leqslant \frac{\mu_0}{L^2}$ and taking the unconditional expectation we have:

$$\mathbb{E}\left\|\mathbf{x}_{t+1} - \mathbf{x}'\right\|^2 \leqslant (1 - \gamma\mu_0)\mathbb{E}\left\|\mathbf{x}_t - \mathbf{x}'\right\|^2 + \gamma^2\sigma^2$$

After unrolling the recursion for $K$ steps we have:

$$\mathbb{E}\left\|\mathbf{x}_{t+1} - \mathbf{x}'\right\|^2 \leqslant (1 - \gamma\mu_0)^K \mathbb{E}\left\|\mathbf{x}_0 - \mathbf{x}'\right\|^2 + \sum_{i=0}^{K}(1 - \gamma\mu_0)^i\gamma^2\sigma^2$$

$$\leqslant (1 - \gamma\mu_0)^K \mathbb{E}\left\|\mathbf{x}_0 - \mathbf{x}'\right\|^2 + \frac{\gamma\sigma^2}{\mu_0}$$

$$\square$$

**Lemma 11.** *Let $\{r_t\}_{t\geqslant 0}$ be a non-negative sequence of numbers that satisfy*

$$r_{t+1} \leqslant (1 - a\gamma)r_t + \frac{b}{K}\gamma \sum_{i=\max\{0,t-K+1\}}^{t} r_i + c\gamma^2,$$

*for constants $a > 0$, $b, c \geqslant 0$ and integer $K \geqslant 1$ and a parameter $\gamma \geqslant 0$, such that $a\gamma \leqslant \frac{1}{K}$. If $b \leqslant \frac{a}{4}$, then it holds*

$$r_t \leqslant \left(1 - \frac{a}{2}\gamma\right)^t r_0 + \frac{2c}{a}\gamma. \tag{11}$$

*Proof.* By assumption on $r_t$:

$$r_{t+1} \leqslant \left(1 - \frac{a\gamma}{2}\right)r_t - \frac{a\gamma}{2}r_t + \frac{b}{K}\gamma \sum_{i=\max\{0,t-K+1\}}^{t} r_i + c\gamma^2,$$

and by unrolling the recursion:

$$r_{t+1} \leqslant \left(1 - a\frac{\gamma}{2}\right)^t r_0 + \sum_{i=0}^{t}\left(1 - \frac{a\gamma}{2}\right)^{t-i}\left[-\frac{a\gamma}{2}r_i + \frac{b}{K}\gamma \sum_{j=\max\{0,i-K+1\}}^{i} r_j\right] + \sum_{i=0}^{t}\left(1 - \frac{a\gamma}{2}\right)^{t-i}c\gamma^2$$

$$\leqslant \left(1 - \frac{a\gamma}{2}\right)^t r_0 + \sum_{i=0}^{t}\left(1 - \frac{a\gamma}{2}\right)^{t-i}\left[-\frac{a\gamma}{2}r_i + \frac{b}{K}\gamma \sum_{j=\max\{0,i-K+1\}}^{i} r_j\right] + \frac{2c}{a}\gamma$$

$$= \left(1 - a\frac{\gamma}{2}\right)^t r_0 + \sum_{i=0}^{t}\left(1 - \frac{a\gamma}{2}\right)^{t-i}\left[-\frac{a\gamma}{2}r_i + \frac{b}{K}\gamma \sum_{j=\max\{0,i-K-1\}}^{i}\left(1 - \frac{a\gamma}{2}\right)^{i-j}r_i\right] + \frac{2c}{a}\gamma$$

where we used $\sum_{i=0}^{t}(1 - \frac{a\gamma}{2})^i \leqslant \frac{2}{a\gamma}$ (for $(\frac{a\gamma}{2}) < 1$) for the second inequality.

By estimating

$$-\frac{a\gamma}{2}r_i + \frac{b}{K}\gamma \sum_{j=\max\{0,i-K-1\}}^{i}(1 - \frac{a\gamma}{2})^{i-j}r_i \leqslant -\frac{a\gamma}{2}r_i + \frac{b}{K}\gamma \sum_{j=\max\{0,i-K-1\}}^{i}\left(1 - \frac{a\gamma}{2}\right)^{1-K}r_i$$

$$\leqslant -\frac{a\gamma}{2}r_i + b\gamma r_i\left(1 - \frac{a\gamma}{2}\right)^{1-K}r_i$$

$$\leqslant -\frac{a\gamma}{2}r_i + 2b\gamma r_i \leqslant 0,$$

with and $(1 - \frac{a\gamma}{2})^{1-K} \leqslant 2$ for $a\gamma \leqslant \frac{1}{K}$, and the assumption $b \leqslant \frac{a}{4}$ (and $r_i \geqslant 0$).

The validity of the inequality, $(1 - \frac{a\gamma}{2})^{1-K} \leqslant 2$ for $a\gamma \leqslant \frac{1}{K}$ can be shown in the following way:

$$\left(1 - \frac{a\gamma}{2}\right)^{1-K} \leqslant \left(1 - \frac{a\gamma}{2}\right)^{-K} \leqslant e^{\frac{a\gamma K}{2}}$$

For the last inequality above we used the approximation $(1 - x)^{-n} \leqslant e^{nx}$ for $x \geqslant 0$ and $n \geqslant 0$:

Given that $a\gamma \leqslant \frac{1}{K}$, we have:

$$e^{\frac{a\gamma K}{2}} \leqslant e^{\frac{1}{2}}.$$

Thus, we have

$$\left(1 - \frac{a\gamma}{2}\right)^{1-K} \leqslant 2$$

Going back to the main proof, we conclude

$$r_{t+1} \leqslant \left(1 - \frac{a\gamma}{2}\right)^t r_0 + \frac{2c}{a}\gamma.$$

as claimed. $\qquad\square$

Now we are ready to prove the following theorem.

**Theorem** (Decoupled SGDA for two-player Games). *For any $R, K \geqslant \Omega\left(\frac{1}{\gamma\mu}\log\left(\frac{4}{\theta}\right)\right)$, after running Decoupled SGDA for a total of $T = KR$ iterations on a function $f$, with the stepsize $\gamma \leqslant \frac{\mu_0}{L^2}$ if the game is weakly coupled with $c = 4$ ($4\theta \leqslant 1$) or $\gamma \leqslant \min\left\{\frac{\mu}{L^2}, \frac{\mu}{KL\delta}, \frac{\mu}{K\delta^2}\right\}$ otherwise, we get a rate of:*

$$\mathbb{E}\big[\|\mathbf{x}_K^R - \mathbf{x}^\star\|^2\big] \leqslant D^2 \exp\left(-\max\left\{(1-4\theta)R, \frac{\gamma\mu}{2}KR\right\}\right) + \frac{\sigma^2\gamma}{\mu}\min\left\{\frac{8\theta}{1-4\theta}, 2\right\},$$

*where $\delta := \frac{L_c}{\sqrt{\alpha\beta}}$, $\sigma^2 := \sigma_{uu}^2 + \sigma_{vv}^2$, and $D = \|\mathbf{x}_0 - \mathbf{x}^\star\|$.*

*Proof.* The proof consists of two parts. First we provide a convergence proof for our method when the game is not weakly coupled and we show a rate that almost matches the baseline GDA. Next, we assume that the game is weakly coupled and provide a proof which shows acceleration compared to GDA. The final rate would be the minimum between these two rates.

**Part 1.** In this part we, assume that the game is not weakly coupled. We start by upper bounding the iterate $\mathbf{x}$ at time step $t + 1$ from the equilibrium.

$$\mathbb{E}\|\mathbf{x}_{t+1} - \mathbf{x}^\star\|^2$$

$$\leqslant \mathbb{E}\left\|\mathbf{x}_t - \gamma\mathbf{B}^{-1}G_0(\mathbf{x}_t, \xi) - \mathbf{x}^\star\right\|^2 + \gamma^2\sigma^2$$

$$\leqslant \mathbb{E}\left\|\mathbf{x}_t - \gamma\mathbf{B}^{-1}F_0(\mathbf{x}_t) - \mathbf{x}^\star\right\|^2 + \gamma^2\sigma^2$$

$$= \mathbb{E}\left\|\mathbf{x}_t - \gamma\mathbf{B}^{-1}F(\mathbf{x}_t) - \mathbf{x}^\star + \gamma\mathbf{B}^{-1}F(\mathbf{x}_t) - \gamma\mathbf{B}^{-1}F_0(\mathbf{x}_t)\right\|^2 + \gamma^2\sigma^2$$

$$\leqslant \left(1 + \frac{\gamma\mu}{2}\right)\left[\mathbb{E}\left\|\mathbf{x}_t - \gamma\mathbf{B}^{-1}F(\mathbf{x}_t) - \mathbf{x}^\star\right\|^2\right] + \gamma\left(\gamma + \frac{2}{\mu}\right)\mathbb{E}\|F(\mathbf{x}_t) - F_0(\mathbf{x}_t)\|_*^2 + \gamma^2\sigma^2$$

$$= \left(1 + \frac{\gamma\mu}{2}\right)\left[\mathbb{E}\|\mathbf{x}_t - \mathbf{x}^\star\|^2 + \gamma^2\mathbb{E}\|F(\mathbf{x}_t) - F(\mathbf{x}^\star)\|_*^2 - 2\gamma\langle F(\mathbf{x}_t) - F(\mathbf{x}^\star), \mathbf{x}_t - \mathbf{x}^\star\rangle\right] +$$

$$\qquad \gamma\left(\gamma + \frac{2}{\mu}\right)\mathbb{E}\|F(\mathbf{x}_t) - F_0(\mathbf{x}_t)\|_*^2 + \gamma^2\sigma^2$$

$$= \left(1 + \frac{\gamma\mu}{2}\right)\left[(1 + \gamma^2L^2 - 2\gamma\mu)\mathbb{E}\|\mathbf{x}_t - \mathbf{x}^\star\|^2\right] + \gamma\left(\gamma + \frac{2}{\mu}\right)\mathbb{E}\|F(\mathbf{x}_t) - F_0(\mathbf{x}_t)\|_*^2 + \gamma^2\sigma^2$$

$$\leqslant \left(1 + \frac{\gamma\mu}{2}\right)\left[(1 - \gamma\mu)\mathbb{E}\|\mathbf{x}_t - \mathbf{x}^\star\|^2\right] + \frac{3\gamma}{\mu}\mathbb{E}\|F(\mathbf{x}_t) - F_0(\mathbf{x}_t)\|_*^2 + \gamma^2\sigma^2$$

$$\leqslant \left(1 - \frac{\gamma\mu}{2}\right)\mathbb{E}\|\mathbf{x}_t - \mathbf{x}^\star\|^2 + \frac{3\gamma\delta^2}{\mu}\mathbb{E}\|\mathbf{x}_t - \mathbf{x}_0\|^2 + \gamma^2\sigma^2$$

Where we assumed that $\gamma \leqslant \frac{\mu}{L^2}$. Now by using the upper bound on consensus error from Lemma 17 we get:

$$\mathbb{E}\left\|\mathbf{x}_{t+1} - \mathbf{x}^\star\right\|^2$$

$$= \left(1 - \frac{\gamma\mu}{2}\right)\mathbb{E}\left\|\mathbf{x}_t - \mathbf{x}^\star\right\|^2 + \frac{\gamma\mu}{16K}\sum_{i=\max\{0,t-K\}}^{t}\mathbb{E}\left\|\mathbf{x}_i - \mathbf{x}^\star\right\|^2 + \left(1 + \frac{12K\gamma\delta^2}{\mu}\right)\gamma^2\sigma^2$$

With the choice of $\gamma \leqslant \frac{\mu}{12K\delta^2}$ where $\delta := \frac{L_c}{\sqrt{\alpha\beta}}$ we have:

$$\mathbb{E}\left\|\mathbf{x}_{t+1} - \mathbf{x}^\star\right\|^2$$

$$\left(1 - \frac{\gamma\mu}{2}\right)\mathbb{E}\left\|\mathbf{x}_t - \mathbf{x}^\star\right\|^2 + \frac{\gamma\mu}{16K}\sum_{i=\max\{0,t-K\}}^{t}\mathbb{E}\left\|\mathbf{x}_i - \mathbf{x}^\star\right\|^2 + 2\frac{\gamma\sigma^2}{\mu}$$

By unrolling the recursion using Lemma 11 we get:

$$\mathbb{E}\left\|\mathbf{x}_K^R - \mathbf{x}^\star\right\|^2 \leqslant D^2\exp\left(-\frac{\mu^2}{L^2}R\right) + 2\frac{\gamma^2}{\mu}\sigma^2$$

**Part 2.** Here we write the proof based on the assumption that the game is weakly coupled. We start by upper bounding the following term:

$$\left\|\mathbf{x}_{t+1} - \mathbf{x}^\star\right\|^2 \leqslant 2\left\|\mathbf{x}_{t+1} - \mathbf{x}'\right\|^2 + 2\left\|\mathbf{x}' - \mathbf{x}^\star\right\|^2 \tag{12}$$

where $\mathbf{x}' = (\mathbf{u}', \mathbf{v}')$ and $\mathbf{u}' = \arg\min_{\mathbf{u}} f(\mathbf{u}, \mathbf{v}_0)$ and $\mathbf{v}' = \arg\max_{\mathbf{v}} f(\mathbf{u}_0, \mathbf{v})$. For the first term we use Lemma 10 and we get:

$$\left\|\mathbf{x}_{t+1} - \mathbf{x}'\right\|^2 \leqslant (1 - \gamma\mu_0)^K\mathbb{E}\left\|\mathbf{x}_0 - \mathbf{x}'\right\|^2 + \frac{\gamma\sigma^2}{\mu_0}$$

Putting this back in (12) gives us:

$$\mathbb{E}\left\|\mathbf{x}_{t+1} - \mathbf{x}^\star\right\|^2$$

$$\leqslant 2(1 - \gamma\mu_0)^K\mathbb{E}\left\|\mathbf{x}_0 - \mathbf{x}'\right\|^2 + 2\theta\,\mathbb{E}\left\|\mathbf{x}_0 - \mathbf{x}^\star\right\|^2 + \frac{2\gamma\sigma^2}{\mu_0}$$

$$\leqslant 4(1 - \gamma\mu_0)^K\mathbb{E}\left\|\mathbf{x}_0 - \mathbf{x}^\star\right\|^2 + 4(1 - \gamma\mu_0)^K\mathbb{E}\left\|\mathbf{x}' - \mathbf{x}^\star\right\|^2 + 2\theta\,\mathbb{E}\left\|\mathbf{x}_0 - \mathbf{x}^\star\right\|^2 + \frac{2\gamma\sigma^2}{\mu_0}$$

$$\leqslant 4(1 - \gamma\mu_0)^K\mathbb{E}\left\|\mathbf{x}_0 - \mathbf{x}^\star\right\|^2 + \left(4(1 - \gamma\mu_0)^K\theta + 2\theta\right)\mathbb{E}\left\|\mathbf{x}_0 - \mathbf{x}^\star\right\|^2 + \frac{2\gamma\sigma^2}{\mu_0}$$

$$\leqslant \left(4(1 - \gamma\mu_0)^K + 4(1 - \gamma\mu_0)^K\theta + 2\theta\right)\mathbb{E}\left\|\mathbf{x}_0 - \mathbf{x}^\star\right\|^2 + \frac{2\gamma\sigma^2}{\mu_0}$$

$$\leqslant \left(4\exp\left(-\gamma\mu_0 K\right) + 4\exp\left(-\gamma\mu_0 K\right)\theta + 2\theta\right)\mathbb{E}\left\|\mathbf{x}_0 - \mathbf{x}^\star\right\|^2 + \frac{2\gamma\sigma^2}{\mu_0}$$

Now we need to make sure that $4\exp\left(-\gamma\mu_0 K\right) \leqslant \theta \leqslant 1$ which implies that $K \geqslant \Omega\left(\frac{1}{\gamma\mu_0}\ln\left(\frac{4}{\theta}\right)\right)$. Next we have:

$$\mathbb{E}\left\|\mathbf{x}_{t+1} - \mathbf{x}^\star\right\|^2 \leqslant 4\theta\,\mathbb{E}\left\|\mathbf{x}_0 - \mathbf{x}^\star\right\|^2 + \frac{2\gamma\sigma^2}{\mu_0}$$

The above recursion can be re-written in terms of two consecutive rounds:

$$\mathbb{E}\left\|\mathbf{x}^{r+1} - \mathbf{x}^\star\right\|^2 \leqslant 4\theta\,\mathbb{E}\left\|\mathbf{x}^r - \mathbf{x}^\star\right\|^2 + \frac{2\gamma\sigma^2}{\mu_0}$$

After unrolling the recursion for $R$ rounds we have:

$$\mathbb{E}\left\|\mathbf{x}^R - \mathbf{x}^\star\right\|^2 \leqslant (4\theta)^R\,\mathbb{E}\left\|\mathbf{x}_0 - \mathbf{x}^\star\right\|^2 + \frac{2\gamma\sigma^2}{\mu_0}\sum_{i=1}^{R}(4\theta)^i$$

Note that we assumed the game is weakly coupled with parameter $c = 4$ which implies that $4\theta \leqslant 1$. Finally we have:

$$\mathbb{E}\left\|\mathbf{x}^R - \mathbf{x}^\star\right\|^2 \leqslant (4\theta)^R \,\mathbb{E}\left\|\mathbf{x}_0 - \mathbf{x}^\star\right\|^2 + \frac{2\gamma\sigma^2}{\mu_0}\sum_{i=1}^{R}(4\theta)^i$$

$$\leqslant D^2\,(4\theta)^R + \frac{8\gamma\sigma^2}{\mu_0}\cdot\frac{\theta}{1 - 4\theta}$$

$\square$

## C    DECOUPLED SGD FOR $N$-PLAYER GAMES

In this section, we generalize all previous results on two-player games to $N$-player games. We first introduce the notation that is needed to define $N$-player games and will be used to establish our convergence guarantees.

**Notation.**    We consider unconstrained $N$-player games where each player $\mathbf{x}^i$ belongs to the space $\mathcal{X}_i = \mathbb{R}^{d_i}$. The vector $\mathbf{x} = (\mathbf{x}^1, \ldots, \mathbf{x}^N) \in \mathbb{R}^d$ is defined in the space $\mathcal{X} = \mathcal{X}_1 \times \ldots \times \mathcal{X}_N = \mathbb{R}^d$ with $d = \sum_{i=1}^N d_i$. The space $\mathcal{X}_i$ for all $i \in [N]$ is equipped with certain Euclidean norms, $\|\mathbf{x}^i\|_i := \langle \mathbf{B}_i\mathbf{x}^i, \mathbf{x}^i\rangle^{1/2}$ where $\mathbf{B}_i$ is a positive definite matrix. The norm in the space $\mathcal{X}$ is then defined by $\|\mathbf{x}\| = (\sum_{i=1}^N \alpha_i\|\mathbf{x}^i\|_i^2)^{1/2}$ where $\alpha_i > 0$; thus, $\|\mathbf{x}\| = \langle \mathbf{B}\mathbf{x}, \mathbf{x}\rangle^{1/2}$, where $\mathbf{B}$ is the block-diagonal matrix with blocks $\alpha_i\mathbf{B}_i$. The dual norms are defined as: $\|\mathbf{g}_i\|_{i,*} := \max_{\|\mathbf{x}^i\|_i=1}\langle \mathbf{g}_i, \mathbf{x}^i\rangle = \langle \mathbf{g}_i, \mathbf{B}_i^{-1}\mathbf{g}_i\rangle^{1/2}$ ($\mathbf{g}_i \in \mathcal{X}_{d_i}$) and $\|\mathbf{g}\|_* := \max_{\|\mathbf{x}\|=1}\langle \mathbf{g}, \mathbf{x}\rangle = (\sum_{i=1}^N \frac{1}{\alpha_i}\|\mathbf{g}_i\|_{i,*}^2)^{1/2} = \langle \mathbf{g}, \mathbf{B}^{-1}\mathbf{g}\rangle^{1/2}$ ($\mathbf{g} \equiv (\mathbf{g}_1, \ldots, \mathbf{g}_N) \in \mathcal{X}$).

Similar to the work Nesterov (2012), we define the following partitioning of the identity matrix:

$$\mathbf{I}_d = (\mathbf{U}_1, \mathbf{U}_2, \ldots, \mathbf{U}_N) \in \mathbb{R}^{d\times d},\ d = \sum_{i=1}^N d_i,\ \mathbf{U}_i \in \mathbb{R}^{d\times d_i}$$

Now we can represent the vector $\mathbf{x}$ as follows:

$$\mathbf{x} = \sum_{i=1}^N \mathbf{U}_i\mathbf{x}^i \in \mathbb{R}^d.$$

We can extract the parameters of one player as follows:

$$\mathbf{x}^i = \mathbf{U}_i^\top\mathbf{x} \in \mathbb{R}^{d_i}$$

**Problem Formulation.**    An $N$-player games is defined as:

$$\left(\min_{\mathbf{x}^1} f_1(\mathbf{x}), \ldots, \min_{\mathbf{x}^N} f_N(\mathbf{x})\right) \qquad\qquad (N\text{-player})$$

Where $f_n \colon \mathcal{X}_{d_n} \to \mathbb{R}$.

The goal is to find the Nash Equilibrium in $\mathbf{x}^\star = (\mathbf{x}^{\star 1}, \ldots, \mathbf{x}^{\star N})$ like in the work Bravo et al. (2018), which has the property that if one player changes their strategy, their payoff function will increase. In other words, there is no incentive to change one strategy alone: for all $\mathbf{h}_n \in \mathcal{X}_n$, it holds that

$$f_n(\mathbf{x}^\star) \leqslant f_n(\mathbf{x}^\star + \mathbf{U}_n\mathbf{h}_n). \qquad\qquad (13)$$

Moreover, we define the operator $F : \mathcal{X} \to \mathcal{X}$ which denotes the stack of gradients with respect to each player's parameters as follows:

$$F(\mathbf{x}) := (\nabla_1 f_1(\mathbf{x}), \ldots, \nabla_N f_N(\mathbf{x}))$$

For the equilibrium, it holds that $F(\mathbf{x}^\star) = 0$. We can extract the partial gradient with respect to one player as follows:

$$\nabla_n f_n(\mathbf{x}) = \mathbf{U}_n^\top F(\mathbf{x}).$$

We now present the assumptions required for the convergence of our method.

**Assumption 5** (Lipschitz gradients). *Operator $F : \mathcal{X} \to \mathcal{X}$ is L-Lipschitz if for all $\mathbf{x}, \mathbf{x}' \in \mathcal{X}$, the following inequality holds:*

$$\|F(\mathbf{x}) - F(\mathbf{x}')\|_* \leqslant L\|\mathbf{x} - \mathbf{x}'\| \tag{14}$$

**Assumption 6** (Lipschitz partial gradients). *For each $n \in [N]$, there exist constants $\hat{L}_n, \bar{L}_n \geqslant 0$ such that, for any $\mathbf{x} \in \mathbb{R}^d$, any $\mathbf{h}_1 \in \mathbb{R}^{d_1}, \ldots, \mathbf{h}_N \in \mathbb{R}^{d_N}$ and any $n \in [N]$, it holds that*

$$\|\nabla_n f_n(\mathbf{x}) - \nabla_n f_n(\mathbf{x} + \mathbf{U}_n \mathbf{h}_n)\|_{n,*} \leqslant \hat{L}_n \|\mathbf{h}_n\|_n,$$
$$\|\nabla_n f_n(\mathbf{x}) - \nabla_n f_n(\mathbf{x} + \sum_{i \neq n} \mathbf{U}_i \mathbf{h}_i)\|_{n,*} \leqslant \bar{L}_n \|\sum_{i \neq n} \mathbf{U}_i \mathbf{h}_i\|. \tag{15}$$

For the $N$-player games, we can define the following matrix for the better understanding of the smoothness parameters:

$$\mathbf{L} = \begin{pmatrix} \hat{L}_1 & & & & \\ & \hat{L}_2 & & \boxed{\bar{L}_n} & \\ & & \ddots & & \\ & \boxed{\bar{L}_n} & & \hat{L}_{N-1} & \\ & & & & \hat{L}_N \end{pmatrix} \tag{16}$$

In the above matrix, the row number corresponds to the player with respect to whom we are taking the derivative, while the column number corresponds to the player that is fixed, with all other parameters changing. All the elements $L_{ii}$ on the main diagonal of the matrix measure the strength of each individual player, while the off-diagonal elements $L_{ij}$ for $i \neq j$ measure the interaction between players $i$ and $j$. We assume that all the diagonal elements are upper bounded by $\hat{L}_n$ and all off-diagonal elements are upper bounded by $\bar{L}_n$. Here $n$ is the player which is being fixed. The parameter $\bar{L}_n$ measures the interaction of the $n$th player with all other players.

**Assumption 7** (Strong monotonicity). *The operator $F : \mathcal{X} \to \mathcal{X}$ is said to be strongly monotone with parameter $\mu > 0$ if for all $\mathbf{x}, \mathbf{x}' \in \mathcal{X}$, the following inequality holds:*

$$\langle F(\mathbf{x}) - F(\mathbf{x}'), \mathbf{x} - \mathbf{x}' \rangle \geqslant \mu \|\mathbf{x} - \mathbf{x}'\|^2. \tag{17}$$

Also we define the more refined strong monotonicity constants $\mu_n > 0$ for a player $n \in [N]$ by the following inequality:

$$\langle \nabla_n f_n(\mathbf{x}) - \nabla_n f(\mathbf{x} + \mathbf{U}_n^\top \mathbf{d}_n), \mathbf{x}^n - \mathbf{x}'^n \rangle \geqslant \mu_n \|\mathbf{x}'^n - \mathbf{x}^n\|_n^2, \tag{18}$$

where $\mathbf{d}_n := \mathbf{x}'^n - \mathbf{x}^n$.

**Assumption 8.** *There exists finite constant $\bar{\sigma}^2$ such that for all $\mathbf{x} \in \mathcal{X}$:*

$$\mathbb{E}\left\|\left[(G_i(\mathbf{x}, \xi) - F(\mathbf{x}))\right]_i\right\|_{i,*}^2 \leqslant \sigma_{ii}^2$$

*Here, we use the operator $[\cdot]_i$ do denote the coordinates corresponding to player $i \in \{1, \ldots, N\}$. For convenience, we define $\bar{\sigma}^2 = \max_{1 \leqslant i \leqslant N} \sigma_{ii}^2$.*

Note that we only require that the noise of self-gradients is bounded, i.e. $\sigma_{ii}^2$. However, we do not make any assumption on the noise for the estimates of the gradients of the other player. Concretely, the variances $\sigma_{ij}^2 := \mathbb{E}\left[\left\|\left[(G_i(\mathbf{x}, \xi) - F(\mathbf{x}))\right]_j\right\|_{i,*}^2\right]$, $i \neq j$ can be arbitrarily large, possibly unbounded. This is is contrast to other works on stochastic min-max optimization, that require the variance $\sigma_{ij}$ to be bounded.

We can also define the noise matrix as follows:

$$\mathcal{M} = \begin{pmatrix} \bar{\sigma}^2 & & & & \\ & \bar{\sigma}^2 & & \boxed{\hat{\sigma}^2} & \\ & & \ddots & & \\ & \boxed{\hat{\sigma}^2} & & \bar{\sigma}^2 & \\ & & & & \bar{\sigma}^2 \end{pmatrix} \tag{19}$$

We assume that each player has access to a separate noisy gradient oracle, which provides the gradient with respect to all other parameters. More specifically, $\sigma_{ij}^2$ corresponds to the noise oracle of player $i$ that computes the gradient with respect to the parameters of player $j$. We assume that all the elements on the main diagonal are upper bounded by $\hat{\sigma}^2$ and all off-diagonal elements are upper bounded by $\hat{\sigma}^2$. Note that, in general, we expect that $\max\{\sigma_{ii}\} \ll \max\{\sigma_{ij} \text{ for } i \neq j\}$, especially in a distributed setting with multiple players. This is because, due to privacy concerns, it may be challenging to compute the gradient with respect to the parameters of other players. As a result, the gradient estimation between different players can be noisy and inaccurate, leading to larger values of $\sigma_{ij}$ for $i \neq j$. On the other hand, computing the gradient with respect to a player's own parameters is generally easier, and it is reasonable to assume that $\max\{\sigma_{ii}\}$ is relatively small. In some cases, we might even assume that $\sigma_{ii} = 0$ for all $i$, while $\max\{\sigma_{ij} \text{ for } i \neq j\}$ remains significantly large.

## C.1 METHOD

We are considering a setting where the $N$ players may not have access to their opponent's strategies or gradients, and only assume that the private components of the gradients have bounded variance, see Assumption 8. For this setting, we therefore propose that each player should only use the reliable information, that is $[G_i(\mathbf{x}, \xi)]_i$ for player $i \in [N]$. We can write our proposed method compactly as:

$$\mathbf{x}_{t+1}^r = \mathbf{x}_t^r - \gamma \mathbf{B}^{-1} G_{\mathbf{x}_0^r}(\mathbf{x}_t^r, \xi_t), \tag{20}$$

where

$$G_{\mathbf{x}_0}(\mathbf{x}, \xi) \equiv \left( \nabla_i f(\mathbf{x}_0 + \mathbf{U}_i \mathbf{U}_i^\top (\mathbf{x} - \mathbf{x}_0)); \xi) \right)_{1 \leqslant i \leqslant N}.$$

wHere, the index $t$ denotes the local update step in the current local update phase on player $i$, and the superscript $r$ indexes the local phases. One communication round is needed for exchanging the updated parameters $\mathbf{x}_K^r$ when passing to the next round. Note that $\mathbf{x}_t^{r,i} \in \mathcal{X}_i$ and $\mathbf{x}_t^r \in \mathcal{X}$.

**Assumption 9** (Strong monotonicity). *The operator $F : \mathcal{X} \to \mathcal{X}$ is said to be strongly monotone with parameter $\mu > 0$ if for all $\mathbf{x}, \mathbf{x}' \in \mathcal{X}$, the following inequality holds:*

$$\langle F(\mathbf{x}) - F(\mathbf{x}'), \mathbf{x} - \mathbf{x}' \rangle \geqslant \mu \|\mathbf{x} - \mathbf{x}'\|^2. \tag{21}$$

---

**Algorithm 2** Decoupled SGD for $N$-player games

---

1: **Input:** step size $\gamma$, initialization $\mathbf{x}_0 = (\mathbf{x}_0^1, \ldots, \mathbf{x}_0^N), R, K$
2: **for** $r \in \{1, \ldots, R\}$ **do**
3:      **for** $t \in \{1, \ldots, K\}$ **do**
4:          **for** $n \in \{1, \ldots, N\}$ **in parallel do**
5:              Update local model $\mathbf{x}_{t+1}^{n,r} \leftarrow \mathbf{x}_t^{n,r} - \gamma \mathbf{B}^{-1} G_{\mathbf{x}_0}(\mathbf{x}_t^r)$
6:          **end for**
7:      **end for**
8:      **Communicate** $\left[\mathbf{x}_K^{1,r}, \ldots, \mathbf{x}_K^{N,r}\right]^\top$ to all players
9: **end for**
10: **Output:** $\mathbf{x}_K^R = (\mathbf{x}_K^{1,R}, \ldots, \mathbf{x}_K^{N,R})$

---

## C.2 CONVERGENCE GUARANTEE

Now we out to a change in the definition of weakly coupled games in $N$-player setting.

**Definition 2** (Weakly Coupled and Fully Decoupled Games). *Given an $N$-player game in the form of N-player. We define the **coupling degree** parameter $\theta$ for this game as follows:*

$$\boxed{\theta := \max_{1 \leqslant i \leqslant N} \left( \sum_{j \neq i} \frac{\alpha_i \bar{L}_i^2}{\mu_i^2} \right)^{1/2}} \tag{22}$$

*This variable measures the level of interaction in the game. A smaller value of $\theta$ indicates less interaction. We say the game is **Weakly Coupled** if the following inequality holds:*

$$\frac{\theta}{c} \leqslant 1 \tag{23}$$

*where $c > 1$ is an absolute constant and will be specified based on the setting. We say the game is **Fully Decoupled** if we have $\theta = 0$ which implies each player is minimizing their own pay-off function independently.*

**Theorem 12.** *For any $R, K \geqslant \Omega\left(\frac{1}{\gamma\mu}\ln\left(\frac{4}{\theta}\right)\right)$, after running Decoupled SGDA for a total of $T = KR$ iterations on a an $N$-player game defined in 2, with a stepsize of $\gamma \leqslant \frac{\mu_{\min}}{L^2}$ if the game is weakly coupled with $c = 4$ ($4\theta \leqslant 1$) or $\gamma \leqslant \min\left\{\frac{\mu}{L^2}, \frac{\mu}{KL\delta}, \frac{\mu}{K\delta^2}\right\}$ otherwise, we get a rate of:*

$$\mathbb{E}\big[\|\mathbf{x}_K^R - \mathbf{x}^\star\|^2\big] \leqslant D^2 \exp\Big(-\max\Big\{(1-4\theta)R, \frac{\gamma\mu}{2}KR\Big\}\Big) + \frac{\bar{\sigma}^2\gamma}{\mu}\min\Big\{\frac{8\theta}{1-4\theta}, 2\Big\},$$

*where $\delta := \max_{1\leqslant i\leqslant N}\big(\sum_{j\neq i}\frac{\bar{L}_j^2}{\alpha_j}\big)^{1/2}$, $\mu_{\min} = \min_{1\leqslant i\leqslant N}\{\frac{\mu_i}{\alpha_i}\}$, and $D = \|\mathbf{x}_0 - \mathbf{x}^\star\|$.*

**Corollary 13.** *With the choice of $\gamma = \frac{\mu_{\min}}{RL^2}$ if the game is weakly coupled and $\gamma = \min\{\frac{\mu}{32NKL^2}, \frac{1}{\mu KR}\ln(\max\{2, \frac{\mu^2 D^2}{\bar{\sigma}^2}KR\})\}$ otherwise, we get*

$$\mathbb{E}\big[\|\mathbf{x}_K^R - \mathbf{x}^\star\|^2\big] \leqslant D^2 \exp\Big(-\max\Big\{(1-4\theta)R, \frac{\mu^2}{2L^2}R\Big\}\Big) + \bar{\sigma}^2\min\left\{\frac{8\theta}{RL^2(1-4\theta)}, \frac{1}{\mu^2 KR}\right\}.$$

### C.3  MISSING PROOFS FOR SECTION C.2

Before establishing the convergence results, we first need a couple of auxiliary lemmas for $N$-player games.

**Lemma 14.** *The operator $F_0$ defined in (20) is strongly monotone: for each $\mathbf{x}, \mathbf{x}' \in \mathcal{X}$, we have*

$$\langle F_0(\mathbf{x}) - F_0(\mathbf{x}'), \mathbf{x} - \mathbf{x}'\rangle \geqslant \min_{1\leqslant i\leqslant N}\Big\{\frac{\mu_i}{\alpha_i}\Big\}\|\mathbf{x} - \mathbf{x}'\|^2. \tag{24}$$

*Proof.* Indeed,

$$\begin{aligned}
\langle F_0(\mathbf{x}) &- F_0(\mathbf{x}'), \mathbf{x} - \mathbf{x}'\rangle \\
&= \sum_{i=1}^N \langle \nabla_i f(\mathbf{x}) - \nabla_i f(\mathbf{x} + \mathbf{U}_i^\top \mathbf{d}_i), \mathbf{x}^i - \mathbf{x}'^i\rangle \\
&\geqslant \sum_{i=1}^N \mu_i\|\mathbf{x} - \mathbf{x}'\|_i^2 = \sum_{i=1}^N \frac{\mu_i}{\alpha_i}\alpha_i\|\mathbf{x} - \mathbf{x}'^i\|_i^2 \\
&\geqslant \min_{1\leqslant i\leqslant N}\Big\{\frac{\mu_i}{\alpha_i}\Big\}\|\mathbf{x} - \mathbf{x}'\|^2. \qquad \square
\end{aligned}$$

where $\mathbf{d}_i := \mathbf{x}'^i - \mathbf{x}^i$.

**Lemma 15** (N-player). *For the points $\mathbf{x}', \mathbf{x}^\star \in \mathcal{X}$ that satisfy $F_0(\mathbf{x}') = 0$ and $F(\mathbf{x}^\star) = 0$, we have that:*

$$\left\|\mathbf{x}' - \mathbf{x}^\star\right\| \leqslant \theta\left\|\mathbf{x}_0 - \mathbf{x}^\star\right\| \tag{25}$$

*Where we defined $\theta := \max_{1\leqslant i\leqslant N}\big(\sum_{j\neq i}\frac{\alpha_i \bar{L}_i^2}{\mu_i^2}\big)^{1/2}$.*

*Proof.* Let's define $\mathbf{h}_i := \mathbf{x}_0^i - \mathbf{x}^i$, $\mathbf{d}_i := \mathbf{x}'^i - \mathbf{x}^i$, $\mathbf{r}_i := \mathbf{x}^{\star i} - \mathbf{x}^i$, $\mathbf{s}_i := \mathbf{x}_0^i - \mathbf{x}^{\star i}$. We first introduce the point $\mathbf{x}' \in \mathbb{R}^d$ as follows:

$$\mathbf{x}' = (\mathbf{x}'^1, \dots, \mathbf{x}'^N), \quad \mathbf{x}'^i = \arg\min_{\mathbf{x}^i \in \mathbb{R}^{d_i}} f_i\Big(\mathbf{x} + \sum_{j\neq i}\mathbf{U}_j\mathbf{h}_j\Big)$$

$$\left\| \mathbf{x}' - \mathbf{x}^\star \right\|^2$$

$$= \sum_{i=1}^N \alpha_i \left\| \mathbf{x}'^i - \mathbf{x}^\star \right\|_i^2$$

$$\leqslant \sum_{i=1}^N \frac{\alpha_i}{\mu_i^2} \left\| \nabla_i f_i \left( \mathbf{x} + \mathbf{U}_i \mathbf{d}_i + \sum_{j \neq i} \mathbf{U}_j \mathbf{h}_j \right) - \nabla_i f_i \left( \mathbf{x} + \mathbf{U}_i \mathbf{r}_i + \sum_{j \neq i} \mathbf{U}_j \mathbf{h}_j \right) \right\|_{i,*}^2$$

$$= \sum_{i=1}^N \frac{\alpha_i}{\mu_i^2} \left\| \nabla_i f_i(\mathbf{x}^\star) - \nabla_i f_i \left( \mathbf{x} + \mathbf{U}_i \mathbf{r}_i + \sum_{j \neq i} \mathbf{U}_j \mathbf{h}_j \right) \right\|_{i,*}^2 \leqslant \sum_{i=1}^N \frac{\alpha_i \bar{L}_i^2}{\mu_i^2} \left\| \sum_{j \neq i} \mathbf{U}_j \mathbf{s}_j \right\|^2$$

$$\leqslant \sum_{i=1}^N \frac{\alpha_i \bar{L}_i^2}{\mu_i^2} \sum_{j \neq i} \alpha_j \left\| \mathbf{s}_j \right\|_j^2 \leqslant \sum_{i=1}^N \beta_i \alpha_i \left\| \mathbf{s}_i \right\|_i^2 \leqslant \theta^2 \left\| \mathbf{s} \right\|^2$$

Where $\beta_i = \sum_{j \neq i} \frac{\alpha_i \bar{L}_i^2}{\mu_i^2}$ and $\theta := \max_{1 \leqslant i \leqslant N} \sqrt{\beta_i}$. $\qquad\qquad\square$

**Lemma 16** ($N$-player). *For the operators $F$ and $F_0$ and $\delta := \max_{1 \leqslant i \leqslant N} \left( \sum_{j \neq i} \frac{\bar{L}_j^2}{\alpha_j} \right)^{1/2}$, we have*

$$\| F(\mathbf{x}) - F_0(\mathbf{x}) \|_* \leqslant \delta \| \mathbf{x} - \mathbf{x}_0 \| . \tag{26}$$

*Proof.* We first define $\mathbf{h}_i := \mathbf{x}_0^i - \mathbf{x}^i$ and $\mathbf{v} := \sum_{j \neq i} \mathbf{U}_j \mathbf{h}_j$. Next we have:

$$\| F_0(\mathbf{x}) - F(\mathbf{x}) \|_*^2 = \sum_{i=1}^N \frac{1}{\alpha_i} \| \nabla_i f(\mathbf{x}) - \nabla_i f(\mathbf{x} + \mathbf{v}) \|_{i,*}^2 \leqslant \sum_{i=1}^N \frac{\bar{L}_i^2}{\alpha_i} \| \mathbf{v} \|^2$$

$$= \sum_{i=1}^N \frac{\bar{L}_i^2}{\alpha_i} \sum_{j \neq i} \alpha_j \| \mathbf{h}_j \|_j^2 = \sum_{i=1}^N \beta_i \alpha_i \| \mathbf{h}_i \|_i^2 ,$$

where $\beta_i = \sum_{j \neq i} \frac{\bar{L}_j^2}{\alpha_j}$. Defining now $\delta^2 = \max_{1 \leqslant i \leqslant N} \beta_i$, we get $\sum_{i=1}^N \beta_i \alpha_i \| \mathbf{h}_i \|_i^2 \leqslant \delta^2 \| \mathbf{h} \|^2$. $\quad\square$

**Lemma 17** (**Consensus error**). *After running Decoupled SGD for $K$ local steps at some round $r$ with a step-size of $\gamma \leqslant \frac{\mu}{32 \delta L K}$, the consensus error can be upper bounded as follows:*

$$\mathbb{E} \left\| \mathbf{x}_{t+1} - \mathbf{x}_0 \right\|^2 \leqslant \sum_{i=t+1-K}^t \frac{\mu^2}{64 K \delta^2} \mathbb{E} \left\| \mathbf{x}_i - \mathbf{x}^\star \right\|^2 + 4 K \gamma^2 \bar{\sigma}^2 \tag{27}$$

*Proof.* The proof is almost identical to the two-player case with two differences. Firstly, the variance of the stochastic noise is upper bounded by $\bar{\sigma}^2$. Secondly, the upper bound on the term $\| F(\mathbf{x}) - F_0(\mathbf{x}) \|_*$ should be obtained from Lemma 16. $\qquad\square$

With the use of these Lemmas, one can easily extend the proof of two player game to the general $N$-player games.

# D  DECOUPLED GDA FOR QUADRATIC GAMES

To provide an extra insight for the results we showed so far and support them with a separate analysis, we additionally consider quadratic functions in this section which are a sub-class of SCSC functions. A general quadratic game can be defined as:

$$f(\mathbf{u}, \mathbf{v}) = \frac{1}{2} \langle \mathbf{u}, \mathbf{A}\mathbf{u} \rangle - \frac{1}{2} \langle \mathbf{v}, \mathbf{B}\mathbf{v} \rangle + \langle \mathbf{u}, \mathbf{C}\mathbf{v} \rangle , \tag{QG}$$

where $\mathbf{A} \in \mathbb{S}_{++}^{d_u}$ and $\mathbf{B} \in \mathbb{S}_{++}^{d_v}$ and $\mathbf{C} \in \mathbb{R}^{d_u \times d_v}$. The matrix $\mathbf{C}$ can be seen as the interaction between two players as it's the only term which involves both $\mathbf{u}$ and $\mathbf{v}$.

**Definition 3.** *Consider a function $f : \mathbb{R}^{d_u} \times \mathbb{R}^{d_v} \to \mathbb{R}$ in the form of (QG) for some $\mathbf{u} \in \mathbb{R}^{d_u}, \mathbf{v} \in \mathbb{R}^{d_v}$. The Lipschitzness parameters can be defined as:*

$$L_u := \lambda_{max}(\mathbf{A}), \quad L_v := \lambda_{max}(\mathbf{B}), \quad L_{uv} := L_{vu} = \|\mathbf{C}\|$$

*and the strong convexity/concavity parameters can be define as:*

$$\mu_u := \lambda_{min}(\mathbf{A}), \quad \mu_v := \lambda_{min}(\mathbf{B})$$

*and the condition numbers can be defined as:*

$$\kappa_u := \frac{L_u}{\mu_u}, \quad \kappa_v := \frac{L_v}{\mu_v}$$

Recall that we defined a general two player game as $f(\mathbf{u}, \mathbf{v}) = g(\mathbf{u}) - h(\mathbf{v}) + r(\mathbf{u}, \mathbf{v})$. For the class of quadratic games, we can be more specific as functions $g(\cdot)$ and $h(\cdot)$ are quadratic functions and $r(\cdot)$ is just a linear term. Moreover, we can be more accurate about the smoothness and strong convexity parameters as they are correspond to the maximum and minimum singular values of the matrices $\mathbf{A}, \mathbf{B}$ and $\mathbf{C}$. So the class of quadratic games can be written as $\mathcal{F}(\lambda_{\min}(\mathbf{A}), \lambda_{\min}(\mathbf{B}), \lambda_{\max}(\mathbf{A}), \lambda_{\max}(\mathbf{B}), \|\mathbf{C}\|)$.

**Lemma 18.** *Given a two-player quadratic game $f(\mathbf{u}, \mathbf{v}) \in \mathcal{F}(\mu, \mu_u, \mu_v, L, L_u, L_v, L_{uv}, L_{vu})$ in the form of (QG). At some round $r$ after $K$ local steps with a stepsize of $\gamma \leqslant \max\{\frac{1}{L_u}, \frac{1}{L_v}\}$ on each player, the exact iterate generated by Decoupled GDA is given as follows:*

$$\mathbf{x}_K^r = \left[\mathbf{Q}^K + \mathbf{E}\right] \mathbf{x}_0^r$$

$$\mathbf{Q} := \begin{pmatrix} (\mathbf{I} - \gamma\mathbf{A}) & \mathbf{0} \\ \mathbf{0} & (\mathbf{I} - \gamma\mathbf{B}) \end{pmatrix}, \quad \mathbf{E} := \begin{pmatrix} \mathbf{0} & -\mathbf{E}_u \\ \mathbf{E}_v & \mathbf{0} \end{pmatrix} \tag{28}$$

$$\mathbf{E}_u := \left[\mathbf{I} - (\mathbf{I} - \gamma\mathbf{A})^K\right]\mathbf{A}^{-1}\mathbf{C}, \quad \mathbf{E}_v := \left[\mathbf{I} - (\mathbf{I} - \gamma\mathbf{B})^K\right]\mathbf{B}^{-1}\mathbf{C}^\top$$

*After taking the norm of both sides we have:*

$$\|\mathbf{x}_K^r\| \leqslant \max\left\{(1 - \gamma\lambda_{min}(\mathbf{A}))^K, (1 - \gamma\lambda_{min}(\mathbf{B}))^K\right\} + \|\mathbf{C}\| \cdot \max\left\{\delta(\mathbf{A}), \delta(\mathbf{B})\right\}^{1/2}$$

$$\delta(\mathbf{A}) := \frac{(1 - (1 - \gamma\lambda_{max}(\mathbf{A}))^K)^2}{\lambda_{min}^2(\mathbf{A})}, \quad \delta(\mathbf{B}) := \frac{(1 - (1 - \gamma\lambda_{max}(\mathbf{B}))^K)^2}{\lambda_{min}^2(\mathbf{B})} \tag{29}$$

**Remark 19.** *For a quadratic game in the form of (QG), the saddle point is $\mathbf{x}^\star = (0, 0)$. We expect our method to shrink the norm of $\mathbf{x}_t^r$ in each round by a factor less than $1$ so that we converge to the saddle point.*

Lemma 18 shows the dynamics of Decoupled GDA for quadratic functions. We can decompose the exact iterates and write it as the sum of two matrices $\mathbf{Q}$ and $\mathbf{E}$. As $\mathbf{Q}$ is a diagonal matrix to the power of $K$ and we have that $\gamma \leqslant \max\{\frac{1}{L_u}, \frac{1}{L_v}\}$, we know that when $K \to \infty$ then $\mathbf{Q} \to \mathbf{0}$. The second matrix $\mathbf{E}$ can be seen as an error matrix which is caused by the interactive part of the game. It is clear that if the game is fully decoupled which implies $\mathbf{C} = \mathbf{0}$, we get the trivial result that we converge only with local steps **without** the need for communicating. However, for the case that we have this interactive term and the game is weakly coupled we have to upper bound the norm of this error matrix to derive the convergence rate. The next Theorem shows the convergence rate of decoupled GDA on quadratic games.

**Theorem 20.** *For any $R$ and $K = \Omega\left(\frac{L_{max}}{\mu_{min}} \log\left(\frac{\mu_{min} - L_{uv}}{\mu_{min}}\right)\right)$ with a stepsize of $\gamma \leqslant \frac{1}{L_{max}}$ which ensures $\left(1 - \frac{\mu_{min}}{L_{max}}\right)^K + \frac{L_{xy}}{\mu} \leqslant 1$ and $\max\left\{1 - (1 - \gamma\lambda_{max}(\mathbf{A}))^K, 1 - (1 - \gamma\lambda_{max}(\mathbf{B}))^K\right\} \leqslant 1$, after running Decoupled GDA for a total of $T = KR$ iterations on a quadratic game $f(\mathbf{u}, \mathbf{v}) \in$*

$\mathcal{F}(\mu, \mu_u, \mu_v, L, L_u, L_v, L_{uv}, L_{vu})$ *in the form of* (QG) *assuming the game is weakly coupled with* $c = 1$, *we get a rate of:*

$$\left\| \mathbf{x}^R - \mathbf{x}^\star \right\| \leqslant D \left( \exp\left( -\frac{\mu_{min}}{L_{max}} K \right) + \frac{L_{uv}}{\mu_{min}} \right)^R \tag{30}$$

*Where* $L_{max} := \max\{L_u, L_v\}$ *and* $\mu_{min} := \min\{\mu_u, \mu_v\}$.

Theorem 20 clearly shows the effect of local steps and communication rounds which gives more insights about our method compared to the SCSC case. We can see that the first term in the rate goes to zero with taking more local steps while there is another term that is not affected by local steps. It's indeed intuitive as we don't expect our method to converge with only local steps in general. The remaining error is do to the interactive part. Moreover, in this Theorem we get a better constant factor $c = 1$ compared to $c = 4$ in the SCSC case. All the previous results discussed for SCSC case can be applied to the quadratic setting as well.

### D.1 MISSING PROOFS FROM SECTION D

We first introduce some auxiliary lemmas that are needed for proofs.

**Lemma 21.** *Let* $\mathbf{A}$ *be a positive definite matrix and* $\gamma \geqslant 0$. *Then matrices* $\mathbf{A}^{-1}$ *and* $(\mathbf{I} - \gamma\mathbf{A})$ *are commutative meaning that:*

$$\mathbf{A}^{-1}(\mathbf{I} - \gamma\mathbf{A}) = (\mathbf{I} - \gamma\mathbf{A})\mathbf{A}^{-1} \tag{31}$$

*Proof.*

$$\mathbf{A}^{-1}(\mathbf{I} - \gamma\mathbf{A})$$
$$= \mathbf{A}^{-1} - \gamma\mathbf{I}$$
$$= (\mathbf{I} - \gamma\mathbf{A})\mathbf{A}^{-1}$$

$\square$

**Lemma 22.** *Let* $\mathbf{A}$ *be a positive definite matrix and* $\gamma \geqslant 0$. *Then matrices* $\mathbf{A}^{-1}$ *and* $(\mathbf{I} - \gamma\mathbf{A})^K$ *are commutative meaning that:*

$$\mathbf{A}^{-1}(\mathbf{I} - \gamma\mathbf{A})^K = (\mathbf{I} - \gamma\mathbf{A})^K \mathbf{A}^{-1} \tag{32}$$

*Proof.* By induction we assume that this statement holds for $K$ which means $\mathbf{A}^{-1}(\mathbf{I} - \gamma\mathbf{A})^K = (\mathbf{I} - \gamma\mathbf{A})^K \mathbf{A}^{-1}$. Now we show that this statement holds for $K + 1$.

$$\mathbf{A}^{-1}(\mathbf{I} - \gamma\mathbf{A})^{K+1}$$
$$= \mathbf{A}^{-1}(\mathbf{I} - \gamma\mathbf{A})(\mathbf{I} - \gamma\mathbf{A})^K$$
$$= (\mathbf{I} - \gamma\mathbf{A})\mathbf{A}^{-1}(\mathbf{I} - \gamma\mathbf{A})^K$$
$$= (\mathbf{I} - \gamma\mathbf{A})(\mathbf{I} - \gamma\mathbf{A})^K \mathbf{A}^{-1}$$
$$= (\mathbf{I} - \gamma\mathbf{A})^{K+1} \mathbf{A}^{-1}$$

For the case of $K = 1$ we use the previous Lemma. $\square$

**Lemma 23.** *Let* $\mathbf{A}$ *be a positive definite matrix and* $\gamma \geqslant 0$. *Then we have that:*

$$\mathbf{A}^{-1}\left((\mathbf{I} - \gamma\mathbf{A})^K - \mathbf{I}\right) = \left((\mathbf{I} - \gamma\mathbf{A})^K - \mathbf{I}\right)\mathbf{A}^{-1} \tag{33}$$

*Proof.*

$$\mathbf{A}^{-1}\left((\mathbf{I} - \gamma\mathbf{A})^K - \mathbf{I}\right)$$
$$= \mathbf{A}^{-1}(\mathbf{I} - \gamma\mathbf{A})^K - \mathbf{A}^{-1}$$
$$= (\mathbf{I} - \gamma\mathbf{A})^K \mathbf{A}^{-1} - \mathbf{A}^{-1}$$
$$= \left((\mathbf{I} - \gamma\mathbf{A})^K - \mathbf{I}\right)\mathbf{A}^{-1}$$

$\square$

## D.2 EXPLICIT ITERATES GENERATED BY DECOUPLED GDA

**Lemma 24.** *Given a general quadratic game in the following form:*

$$f(\mathbf{u}, \mathbf{v}) = \frac{1}{2}\mathbf{u}^\top \mathbf{A}\mathbf{u} - \frac{1}{2}\mathbf{v}^\top \mathbf{B}\mathbf{v} + \mathbf{u}^\top \mathbf{C}\mathbf{v}$$

*After $k$ steps of Decoupled GDA at some round $r$ we can compute the explicit form of iterates as follows:*

$$\mathbf{u}_k^r = -\mathbf{A}^{-1}\mathbf{C}\mathbf{v}_0^r + \mathbf{A}^{-1}(\mathbf{I} - \gamma\mathbf{A})^k (\mathbf{A}\mathbf{u}_0^r + \mathbf{C}\mathbf{v}_0^r)$$

$$\mathbf{v}_k^r = \mathbf{B}^{-1}\mathbf{C}^\top \mathbf{u}_0^r + \mathbf{B}^{-1}(\mathbf{I} - \gamma\mathbf{B})^k (\mathbf{B}\mathbf{v}_0^r - \mathbf{C}^\top \mathbf{u}_0^r)$$

*Proof.* We use induction for the proof of this section. By using the update rule of Local GDA we would have,

$$
\begin{aligned}
\mathbf{u}_{k+1}^r &= \mathbf{u}_k - \gamma\nabla_u f(\mathbf{u}_k^r, \mathbf{v}_0^r) \\
&= \mathbf{u}_k - \gamma(\mathbf{A}\mathbf{u}_k^r + \mathbf{C}\mathbf{v}_0^r) \\
&= -\mathbf{A}^{-1}\mathbf{C}\mathbf{v}_0^r + \mathbf{A}^{-1}(\mathbf{I} - \gamma\mathbf{A})^k (\mathbf{A}\mathbf{u}_0^r + \mathbf{C}\mathbf{v}_0^r) \\
&\quad - \gamma\left(\mathbf{A}\left[-\mathbf{A}^{-1}\mathbf{C}\mathbf{v}_0^r + \mathbf{A}^{-1}(\mathbf{I} - \gamma\mathbf{A})^k (\mathbf{A}\mathbf{u}_0^r + \mathbf{C}\mathbf{v}_0)\right] + \mathbf{C}\mathbf{v}_0^r\right) \\[2mm]
&= -\mathbf{A}^{-1}\mathbf{C}\mathbf{v}_0^r + \mathbf{A}^{-1}(\mathbf{I} - \gamma\mathbf{A})^k (\mathbf{A}\mathbf{u}_0^r + \mathbf{C}\mathbf{v}_0^r) \\
&\quad - \gamma\left(-\mathbf{C}\mathbf{v}_0^r + (\mathbf{I} - \gamma\mathbf{A})^k (\mathbf{A}\mathbf{u}_0^r + \mathbf{C}\mathbf{v}_0^r) + \mathbf{C}\mathbf{v}_0^r\right) \\[2mm]
&= -\mathbf{A}^{-1}\mathbf{C}\mathbf{v}_0^r + \mathbf{A}^{-1}(\mathbf{I} - \gamma\mathbf{A})^k (\mathbf{A}\mathbf{u}_0^r + \mathbf{C}\mathbf{v}_0^r) - \gamma(\mathbf{I} - \gamma\mathbf{A})^k (\mathbf{A}\mathbf{u}_0^r + \mathbf{C}\mathbf{v}_0^r) \\
&= -\mathbf{A}^{-1}\mathbf{C}\mathbf{v}_0^r + (\mathbf{A}^{-1} - \gamma\mathbf{I})\left[(\mathbf{I} - \gamma\mathbf{A})^k (\mathbf{A}\mathbf{u}_0^r + \mathbf{C}\mathbf{v}_0^r)\right] \\
&= -\mathbf{A}^{-1}\mathbf{C}\mathbf{v}_0^r + \mathbf{A}^{-1}(\mathbf{I} - \gamma\mathbf{A})\left[(\mathbf{I} - \gamma\mathbf{A})^k (\mathbf{A}\mathbf{u}_0^r + \mathbf{C}\mathbf{v}_0^r)\right] \\
&= -\mathbf{A}^{-1}\mathbf{C}\mathbf{v}_0^r + \mathbf{A}^{-1}(\mathbf{I} - \gamma\mathbf{A})^{k+1} (\mathbf{A}\mathbf{u}_0^r + \mathbf{C}\mathbf{v}_0^r)
\end{aligned}
$$

Now we only need to show that our claim also works for $k = 0$,

$$
\begin{aligned}
\mathbf{u}_0^r &= -\mathbf{A}^{-1}\mathbf{C}\mathbf{v}_0^r + \mathbf{A}^{-1}(\mathbf{I} - \gamma\mathbf{A})^0 (\mathbf{A}\mathbf{u}_0^r + \mathbf{C}\mathbf{v}_0^r) \\
&= -\mathbf{A}^{-1}\mathbf{C}\mathbf{v}_0^r + \mathbf{u}_0^r + \mathbf{A}^{-1}\mathbf{C}\mathbf{v}_0^r \\
&= \mathbf{u}_0^r
\end{aligned}
$$

Also, we do the computation with respect to $\mathbf{v}$:

$$\mathbf{v}_k^r = \mathbf{B}^{-1}\mathbf{C}^\top \mathbf{u}_0^r + \mathbf{B}^{-1}(\mathbf{I} - \gamma\mathbf{B})^k (\mathbf{B}\mathbf{v}_0^r - \mathbf{C}^\top \mathbf{u}_0^r)$$

By using the update rule of Decoupled GDA we get:

$$
\begin{aligned}
\mathbf{v}_{k+1}^r &= \mathbf{v}_k - \gamma \nabla_{\mathbf{v}} f(\mathbf{u}_0^r, \mathbf{u}_k^r) \\
&= \mathbf{v}_k + \gamma \left( -\mathbf{B}\mathbf{v}_k^r + \mathbf{C}^\top \mathbf{u}_0^r \right) \\
&= \mathbf{v}_k - \gamma \left( \mathbf{B}\mathbf{v}_k^r - \mathbf{C}^\top \mathbf{u}_0^r \right) \\
&= \mathbf{B}^{-1}\mathbf{C}^\top \mathbf{u}_0^r + \mathbf{B}^{-1} \left( \mathbf{I} - \gamma\mathbf{B} \right)^k \left( \mathbf{B}\mathbf{v}_0^r - \mathbf{C}^\top \mathbf{u}_0 \right) \\
&\qquad - \gamma \left( \mathbf{B} \left[ \mathbf{B}^{-1}\mathbf{C}^\top \mathbf{u}_0^r + \mathbf{B}^{-1} \left( \mathbf{I} - \gamma\mathbf{B} \right)^k \left( \mathbf{B}\mathbf{v}_0^r - \mathbf{C}^\top \mathbf{u}_0^r \right) \right] - \mathbf{C}^\top \mathbf{u}_0^r \right) \\[2mm]
&= \mathbf{B}^{-1}\mathbf{C}^\top \mathbf{u}_0^r + \mathbf{B}^{-1} \left( \mathbf{I} - \gamma\mathbf{B} \right)^k \left( \mathbf{B}\mathbf{v}_0^r - \mathbf{C}^\top \mathbf{u}_0 \right) \\
&\qquad - \gamma \left( \mathbf{C}^\top \mathbf{u}_0^r + \left( \mathbf{I} - \gamma\mathbf{B} \right)^k \left( \mathbf{B}\mathbf{v}_0^r - \mathbf{C}^\top \mathbf{u}_0^r \right) - \mathbf{C}^\top \mathbf{u}_0^r \right) \\[2mm]
&= \mathbf{B}^{-1}\mathbf{C}^\top \mathbf{u}_0^r + \mathbf{B}^{-1} \left( \mathbf{I} - \gamma\mathbf{B} \right)^k \left( \mathbf{B}\mathbf{v}_0^r - \mathbf{C}^\top \mathbf{u}_0 \right) - \gamma \left( \mathbf{I} - \gamma\mathbf{B} \right)^k \left( \mathbf{B}\mathbf{v}_0^r - \mathbf{C}^\top \mathbf{u}_0^r \right) \\
&= \mathbf{B}^{-1}\mathbf{C}^\top \mathbf{u}_0^r + \left( \mathbf{B}^{-1} - \gamma\mathbf{I} \right) \left[ \left( \mathbf{I} - \gamma\mathbf{B} \right)^k \left( \mathbf{B}\mathbf{v}_0^r - \mathbf{C}^\top \mathbf{u}_0^r \right) \right] \\
&= \mathbf{B}^{-1}\mathbf{C}^\top \mathbf{u}_0^r + \mathbf{B}^{-1} \left( \mathbf{I} - \gamma\mathbf{B} \right) \left[ \left( \mathbf{I} - \gamma\mathbf{B} \right)^k \left( \mathbf{B}\mathbf{v}_0^r - \mathbf{C}^\top \mathbf{u}_0^r \right) \right] \\
&= \mathbf{B}^{-1}\mathbf{C}^\top \mathbf{u}_0^r + \mathbf{B}^{-1} \left( \mathbf{I} - \gamma\mathbf{B} \right)^{k+1} \left( \mathbf{B}\mathbf{v}_0^r - \mathbf{C}^\top \mathbf{u}_0^r \right)
\end{aligned}
$$

Now we only need to show this our claim also works for $k = 0$,

$$
\begin{aligned}
\mathbf{v}_0^r &= \mathbf{B}^{-1}\mathbf{C}^\top \mathbf{u}_0^r + \mathbf{B}^{-1} \left( \mathbf{I} - \gamma\mathbf{B} \right)^0 \left( \mathbf{B}\mathbf{v}_0^r - \mathbf{C}^\top \mathbf{u}_0^r \right) \\
&= \mathbf{B}^{-1}\mathbf{C}^\top \mathbf{u}_0^r + \mathbf{v}_0^r - \mathbf{B}^{-1}\mathbf{C}^\top \mathbf{u}_0^r \\
&= \mathbf{v}_0^r
\end{aligned}
$$

$\square$

### D.3 PROOF OF LEMMA 18

Given a two-player quadratic game $f(\mathbf{u}, \mathbf{v}) \in \mathcal{F}(\mu, \mu_u, \mu_v, L, L_u, L_v, L_{uv}, L_{vu})$ in the form of (QG). At some round $r$ after $k$ local steps with a stepsize of $\gamma \leqslant \max\{\frac{1}{L_u}, \frac{1}{L_v}\}$ on each player, the exact iterate generated by Decoupled GDA is given as follows:

$$
\mathbf{x}_k^r = \left[ \mathbf{Q}^k + \mathbf{E} \right] \mathbf{x}_0^r
$$

$$
\mathbf{Q} := \begin{pmatrix} (\mathbf{I} - \gamma\mathbf{A}) & \mathbf{0} \\ \mathbf{0} & (\mathbf{I} - \gamma\mathbf{B}) \end{pmatrix}, \quad \mathbf{E} := \begin{pmatrix} \mathbf{0} & -\mathbf{E}_u \\ \mathbf{E}_v & \mathbf{0} \end{pmatrix} \tag{34}
$$

$$
\mathbf{E}_u := \left[ \mathbf{I} - (\mathbf{I} - \gamma\mathbf{A})^k \right] \mathbf{A}^{-1}\mathbf{C}, \quad \mathbf{E}_v := \left[ \mathbf{I} - (\mathbf{I} - \gamma\mathbf{B})^k \right] \mathbf{B}^{-1}\mathbf{C}^\top
$$

After taking the norm of both sides we have:

$$
\| \mathbf{x}_k^r \| \leqslant \max \left\{ (1 - \gamma\lambda_{\min}(\mathbf{A}))^k, (1 - \gamma\lambda_{\min}(\mathbf{B}))^k \right\} + \| \mathbf{C} \|^2 \cdot \max \left\{ \delta(\mathbf{A}), \delta(\mathbf{B}) \right\}
$$

$$
\delta(\mathbf{A}) := \frac{(1 - (1 - \gamma\lambda_{\max}(\mathbf{A}))^k)^2}{\lambda_{\min}^2(\mathbf{A})}, \quad \delta(\mathbf{B}) := \frac{(1 - (1 - \gamma\lambda_{\max}(\mathbf{B}))^k)^2}{\lambda_{\min}^2(\mathbf{B})} \tag{35}
$$

*Proof.* From Lemma 24 we can write the explicit iterates for the variable $\mathbf{x}$:

$$\|\mathbf{x}_k^r\| = \left\| \begin{pmatrix} (\mathbf{I} - \gamma\mathbf{A}) & \mathbf{0} \\ \mathbf{0} & (\mathbf{I} - \gamma\mathbf{B}) \end{pmatrix}^k + \begin{pmatrix} \mathbf{0} & -\mathbf{E}_u \\ \mathbf{E}_v & \mathbf{0} \end{pmatrix} \right\| \cdot \|\mathbf{x}_0^r\|$$

$$\leqslant \left\| \begin{pmatrix} (\mathbf{I} - \gamma\mathbf{A}) & \mathbf{0} \\ \mathbf{0} & (\mathbf{I} - \gamma\mathbf{B}) \end{pmatrix}^k \right\| + \left\| \begin{pmatrix} \mathbf{0} & -\mathbf{E}_u \\ \mathbf{E}_v & \mathbf{0} \end{pmatrix} \right\| \cdot \|\mathbf{x}_0^r\|$$

$$\leqslant \max\left\{ (1 - \gamma\lambda_{\min}(\mathbf{A}))^k, (1 - \gamma\lambda_{\min}(\mathbf{B}))^k \right\} \cdot \|\mathbf{x}_0^r\| + \left\| \begin{pmatrix} \mathbf{0} & -\mathbf{E}_u \\ \mathbf{E}_v & \mathbf{0} \end{pmatrix} \right\| \cdot \|\mathbf{x}_0^r\|$$

For computing the norm of the error matrix we need to compute $\sqrt{\lambda_{\max}(\mathbf{E}^\top\mathbf{E})}$. We first form $\mathbf{E}^\top\mathbf{E}$:

$$\mathbf{E}^\top\mathbf{E} = \begin{pmatrix} \mathbf{E}_v^\top\mathbf{E}_v & \mathbf{0} \\ \mathbf{0} & \mathbf{E}_u^\top\mathbf{E}_u \end{pmatrix}$$

So we have:

$$\lambda_{\max}(\mathbf{E}^\top\mathbf{E}) = \max\left\{ \lambda_{\max}(\mathbf{E}_u^\top\mathbf{E}_u), \lambda_{\max}(\mathbf{E}_v^\top\mathbf{E}_v) \right\}$$

For computing the $\lambda_{\max}(\mathbf{E}_u^\top\mathbf{E}_u)$ we have:

$$\lambda_{\max}\left(\mathbf{E}_u^\top\mathbf{E}_u\right) = \lambda_{\max}\left( \mathbf{C}^\top\mathbf{A}^{-\top} \left[\mathbf{I} - (\mathbf{I} - \gamma\mathbf{A})^k\right]^\top \left[\mathbf{I} - (\mathbf{I} - \gamma\mathbf{A})^k\right] \mathbf{A}^{-1}\mathbf{C} \right)$$

$$\leqslant \|\mathbf{C}\|^2 \lambda_{\max}\left( \mathbf{A}^{-\top} \left[\mathbf{I} - (\mathbf{I} - \gamma\mathbf{A})^k\right]^\top \left[\mathbf{I} - (\mathbf{I} - \gamma\mathbf{A})^k\right] \mathbf{A}^{-1} \right)$$

$$\leqslant \|\mathbf{C}\|^2 \lambda_{\max}\left(\mathbf{A}^{-\top}\right) \lambda_{\max}\left(\left[\mathbf{I} - (\mathbf{I} - \gamma\mathbf{A})^k\right]^\top\right) \lambda_{\max}\left(\left[\mathbf{I} - (\mathbf{I} - \gamma\mathbf{A})^k\right]\right) \lambda_{\max}\left(\mathbf{A}^{-1}\right)$$

$$\leqslant \|\mathbf{C}\|^2 \frac{(1 - (1 - \gamma\lambda_{\max}(\mathbf{A}))^k)^2}{\lambda_{\min}^2(\mathbf{A})}$$

$$\leqslant \frac{\|\mathbf{C}\|^2}{\lambda_{\min}^2(\mathbf{A})}$$

We have the same computation with respect to player $\mathbf{v}$ as well which gives us:

$$\lambda_{\max}\left(\mathbf{E}_v^\top\mathbf{E}_v\right) = \|\mathbf{C}\|^2 \frac{(1 - (1 - \gamma\lambda_{\max}(\mathbf{B}))^k)^2}{\lambda_{\min}^2(\mathbf{B})}$$

$$\leqslant \frac{\|\mathbf{C}\|^2}{\lambda_{\min}^2(\mathbf{B})}$$

$\square$

## D.4 PROOF OF THEOREM 20

For any $R$ and $K = \Omega\left(\frac{L_{\max}}{\mu_{\min}} \log\left(\frac{\mu_{\min} - L_{uv}}{\mu_{\min}}\right)\right)$ with a stepsize of $\gamma \leqslant \frac{1}{L_{\max}}$ which ensures $\left(1 - \frac{\mu_{\min}}{L_{\max}}\right)^K + \frac{L_{xy}}{\mu} \leqslant 1$ and $\max\left\{1 - (1 - \gamma\lambda_{\max}(\mathbf{A}))^k, 1 - (1 - \gamma\lambda_{\max}(\mathbf{B}))^k\right\} \leqslant 1$, after running Decoupled GDA for a total of $T = KR$ iterations on a quadratic game $f(\mathbf{u}, \mathbf{v}) \in \mathcal{F}(\mu, \mu_u, \mu_v, L, L_u, L_v, L_{uv}, L_{vu})$ in the form of (QG) assuming the game is weakly coupled with $c = 1$, we get a rate of:

$$\left\|\mathbf{x}^R - \mathbf{x}^\star\right\| \leqslant B\left(\exp\left(-\frac{\mu_{\min}}{L_{\max}}K\right) + \frac{L_{uv}}{\mu_{\min}}\right)^R \tag{36}$$

Where $L_{\max} := \max\{L_u, L_v\}$ and $\mu_{\min} := \min\{\mu_u, \mu_v\}$.

*Proof.* Using previous Lemmas we have:

$$\|\mathbf{x}_k^r\|$$

$$\leqslant \max\left\{(1 - \gamma\lambda_{\min}(\mathbf{A}))^K, (1 - \gamma\lambda_{\min}(\mathbf{B}))^K\right\} \cdot \|\mathbf{x}_0^r\| + \left\|\begin{pmatrix} \mathbf{0} & -\mathbf{E}_u \\ \mathbf{E}_v & \mathbf{0} \end{pmatrix}\right\| \cdot \|\mathbf{x}_0^r\|$$

$$\leqslant \max\left\{(1 - \gamma\lambda_{\min}(\mathbf{A}))^K, (1 - \gamma\lambda_{\min}(\mathbf{B}))^K\right\} \cdot \|\mathbf{x}_0^r\| + \|\mathbf{C}\| \max\left\{\frac{1}{\lambda_{\min}(\mathbf{A})}, \frac{1}{\lambda_{\min}(\mathbf{B})}\right\} \cdot \|\mathbf{x}_0^r\|$$

$$\leqslant \left((1 - \gamma\mu_{\min})^K + \frac{\|\mathbf{C}\|}{\mu_{\min}}\right) \cdot \|\mathbf{x}_0^r\|$$

After unrolling the above recursion for $R$ rounds we get:

$$\|\mathbf{x}_k^r\| \leqslant B\left((1 - \gamma\mu_{\min})^K + \frac{\|\mathbf{C}\|}{\mu_{\min}}\right)^R$$

$\square$

# E ADDITIONAL RELATED WORKS & DISCUSSION

## E.1 DECENTRALIZED OPTIMIZATION

The key difference between decentralized and distributed minimax approaches is the presence of a central server. In the former, there is no central server, and nodes communicate directly with their neighbors, whereas in the latter, a central server aggregates the parameters. Our method belongs to the category of distributed methods. However, we will discuss later on that our approach is completely different from the general idea of distributed / federated optimization.

Decentralized optimization is widely studied for the case of minimization (Xiao & Boyd, 2004; Tsitsiklis, 1984) with the goal of not relying on a central node or server. This idea is also applied to the case of minimax optimization problems. The paper Liu et al. (2020) is the first who studied non-convex-non-concave decentralized minimax. They also used the idea of optimistic gradient descent and achieved a rate of $\mathcal{O}(\epsilon^{-12})$. In Xian et al. (2021), authors proposed an algorithm called DM-HSGD for non-convex decentralized minimax by utilizing variance reduction and achieved a rate of $\mathcal{O}(\kappa^3\epsilon^{-3})$. Recently, authors in Liu et al. (2023) proposed an algorithm named Precision for the non-convex-strongly-concave objectives which has a two-stage local updates and gives a rate of $\mathcal{O}(\frac{1}{T})$.

## E.2 COMPARISON BETWEEN DECOUPLED SGDA AND FEDERATED MINIMAX (LOCAL SGDA)

In this section, we aim to highlight the key differences between our method and existing distributed or decentralized methods in the literature. As mentioned earlier, our method can be classified as distributed, though it has a major difference from others. In fact, this difference lies in the problem formulation.

**Decentralized / Distributed minimax formulation.** In these settings, we aim to solve the following finite-sum optimization problem over $M$ clients:

$$f(\mathbf{u}, \mathbf{v}) = \frac{1}{M}\sum_{m=1}^{M} f_m(\mathbf{u}, \mathbf{v}) \tag{37}$$

In the above formulation, it is assumed that each client has a different data distribution $\mathcal{D}_m$ and tries to solve the game based on this data. It means that each client keeps updating **both** $\mathbf{u}$ and $\mathbf{v}$ at the same time for several steps. Then the server aggregates the parameters and sends them back to clients. The ultimate goal is to find the saddle point $\mathbf{x}^\star = (\mathbf{u}^\star, \mathbf{v}^\star)$ of the global function $f$, as if the entire dataset $\mathcal{D} = \mathcal{D}_1 \cup \cdots \cup \mathcal{D}_M$ were on a single machine running GDA on it. In this

setting, each client is allowed to update both players meaning that it has access to the gradient of $f_m$ with respect to $\mathbf{u}$ and $\mathbf{v}$. However in our approach, instead of splitting the data over clients, we split the parameter space. It means one machine is responsible for **only** updating $\mathbf{u}$ and another for $\mathbf{v}$. Our method also allows to have several machines for $\mathbf{u}$ and several machines for $\mathbf{v}$. We discuss this setting in Section **??**. An important point to consider is that the notions of *client* and *player* should not be intermixed. When the number of players is fixed, the distributed minimax approach essentially runs several instances $(f_m)$ of the main game $(f)$ in parallel to ultimately find the saddle point of $f$. In contrast, our method directly finds the saddle point of $f$ by splitting the parameter space across different machines. Figure 7 illustrates the difference between these two methods. s

**Decoupled GDA**

$$\begin{cases} \mathbf{u}_{k+1}^r = \mathbf{u}_k^r - \gamma \nabla_u f(\mathbf{u}_k^r, \mathbf{v}_0^r) \\ \mathbf{v}_{k+1}^r = \mathbf{v}_k^r + \gamma \nabla_v f(\mathbf{u}_0^r, \mathbf{v}_k^r) \end{cases}$$

**GDA**

$$\begin{cases} \mathbf{u}_{k+1} = \mathbf{u}_k - \gamma \nabla_u f(\mathbf{u}_k, \mathbf{v}_k) \\ \mathbf{v}_{k+1} = \mathbf{v}_k + \gamma \nabla_v f(\mathbf{u}_k, \mathbf{v}_k) \end{cases}$$

**Federated Minimax**

$$\begin{cases} \mathbf{u}_{k+1}^{r,m} = \mathbf{u}_k^{r,m} - \gamma \nabla_u f_m(\mathbf{u}_k^{r,m}, \mathbf{v}_k^{r,m}) \\ \mathbf{v}_{k+1}^{r,m} = \mathbf{v}_k^{r,m} + \gamma \nabla_v f_m(\mathbf{u}_k^{r,m}, \mathbf{v}_k^{r,m}) \end{cases}$$

Figure 6: Comparison of different gradient descent ascent (GDA) approaches: Decoupled GDA, standard GDA, and Federated Minimax. The top box represents Decoupled GDA, where $\mathbf{u}$ and $\mathbf{v}$ gradients are separated, while the bottom left and right boxes represent the standard GDA and Federated Minimax approaches, respectively.

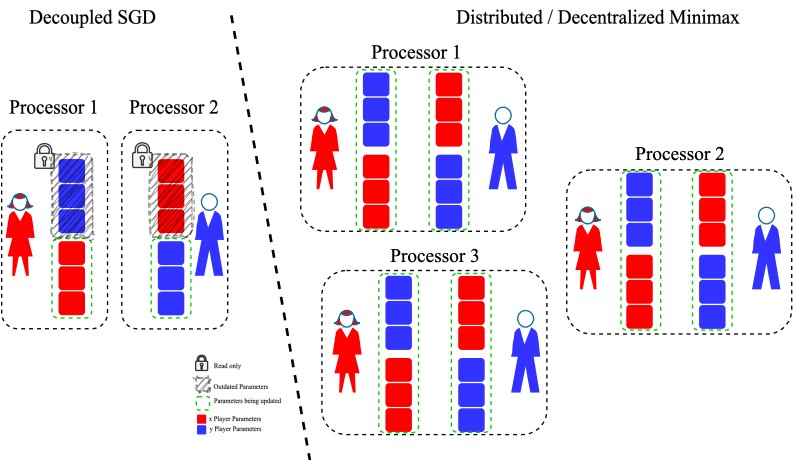

Figure 7: Comparison of our method with the federated minimax formulation: Our method splits the parameter space, while the federated formulation splits the data. Moreover, our method only allows each player to access the gradient with respect to their own parameters, whereas in federated minimax, each player can compute the gradient with respect to both their own parameters and the other player's parameters.

# F FEDERATED DECOUPLED SGDA

## F.1 COMPARING DECOUPLED SGDA WITH FEDERATED LEARNING FOR MINIMAX OPTIMIZATION

Federated learning (FL) builds on the foundational work in distributed minimization, exploring various settings. In the context of minimax optimization, methods like Local SGD have been extended to achieve convergence rates for different classes of functions in both heterogeneous and homogeneous regimes. FL methods for games differ from the setting considered in this work. In FL, multiple copies of all strategies (parameters) are trained locally on different machines and datasets and periodically aggregated. FL is suited for scenarios where a single local machine runs a multi-player algorithm and has access to all players' loss functions, with "collaboration" built into the design. In contrast, our method suits competitive distributed players (local machines) where each player has noisy or outdated strategies of the remaining players. For further discussion, see Appendix E. Additionally, federated learning assumes balanced noise across players, which is not required in our setting; revisited in § 2 and § 5. Finally, in § 4, we identify a class of games where our approach leads to faster convergence, even if fully centralized training is possible, which class similarly arises in non-convex settings–§ 5.In the rest of this section, we study Federated Decoupled SGDA, which is a combination of Federated Minimax and Decoupled SGDA algorithms, and can benefit from the advantages of both approaches. In the next section we propose this method with details.

## F.2 FEDERATED DECOUPLED SGDA METHOD

In this section, we study an extension of our algorithm for distributed setting. For simplicity and in order to be aligned with other works Deng & Mahdavi (2021); Sharma et al. (2022), we consider two-player zero-sum minimax games. Our results for the distributed setting can be extended to the $N$-player case.

**Notation and Problem Definition**  In distributed minimax optimization, we aim to solve the following problem:

$$\min_{\mathbf{u} \in \mathbb{R}^{d_u}} \max_{\mathbf{v} \in \mathbb{R}^{d_v}} \left[ f(\mathbf{u}, \mathbf{v}) = \frac{1}{M} \sum_{m=1}^{M} f_m(\mathbf{u}, \mathbf{v}) = \frac{1}{M} \sum_{m=1}^{M} \mathbb{E}_{\xi_m \sim \mathcal{D}_m} f_m(\mathbf{u}, \mathbf{v}, \xi_m) \right] \tag{38}$$

In this setting, we assume that each player's data is distributed across $M$ clients/processors. So each processor has access to a function $f_m(\mathbf{x}, \mathbf{y})$ on which it can perform stochastic gradient steps. The variance of the stochastic noise is uniformly bounded by $\sigma^2$. We denote $\mathbf{u}_k^{m,r}$ and $\mathbf{v}_k^{m,r}$ as the parameters of players $\mathbf{u}$ and $\mathbf{v}$ on client $m$ in some round $r$ after $k$ local steps. We also use the notation $\bar{\mathbf{u}}_k^r = \frac{1}{M} \sum_{m=1}^{M} \mathbf{u}_k^{m,r}$ and $\bar{\mathbf{v}}_k^r = \frac{1}{M} \sum_{m=1}^{M} \mathbf{v}_k^{m,r}$ to denote the average of parameters over clients at some round $r$ after $k$ local steps. Data distribution across processors can be either homogeneous or heterogeneous. In the heterogeneous regime, which is the case of study in this paper, each processor holds a different payoff function. To measure the heterogeneity of the problem, it's common to use the following assumption:

**Assumption 10.** *There exists a constant $\zeta_\star > 0$ satisfying the following inequality in distributed minimax games:*

$$\max \left\{ \sup_m \|\nabla_u f_m(\mathbf{x}^\star)\|^2, \sup_m \|\nabla_v f_m(\mathbf{x}^\star)\|^2 \right\} \leqslant \zeta_\star^2 \tag{39}$$

Assumption 10 is very common in federated learning and it has been used in many works Koloskova et al. (2020); Deng & Mahdavi (2021); Khaled et al. (2020). Another common assumption in the literature Woodworth et al. (2020b); Patel et al. (2024) is gradient similarity $\zeta$ for every point $\mathbf{x} \in \mathbb{R}^d$ which is a stronger assumption and cannot be satisfied for quadratic functions. In this work, we use Assumption 10 to provide our convergence guarantee for our method.

We also define operators $F^0(\mathbf{x})$, $F_m^0(\mathbf{x})$ are defined as follows:

$$F_m^0(\mathbf{x}_k^{m,r}) := \begin{pmatrix} \nabla_u f_m(\mathbf{u}_k^{m,r}, \mathbf{v}_0^{m,r}) \\ -\nabla_v f_m(\mathbf{u}_0^{m,r}, \mathbf{v}_k^{m,r}) \end{pmatrix}, \quad F^0(\bar{\mathbf{x}}_k^r) := \begin{pmatrix} \frac{1}{M}\sum_{m=1}^M \nabla_u f_m(\bar{\mathbf{u}}_k^r, \bar{\mathbf{v}}_0^r) \\ -\frac{1}{M}\sum_{m=1}^M \nabla_v f_m(\bar{\mathbf{u}}_0^r, \bar{\mathbf{v}}_k^r) \end{pmatrix} \tag{40}$$

In this work, we assume that the operator $F^0$ is $\mu$-strongly monotone.

**Remark 25.** *Note that in general $F_m^0(\mathbf{x}^\star) \neq 0$ and $F^0(\mathbf{x}^\star) \neq 0$. However, if we had the common operator with the most recent parameters, we could have said $F_m(\mathbf{x}^\star) \neq 0$ and $F(\mathbf{x}^\star) = 0$.*

---

**Algorithm 3** Decoupled SGDA for two-player federated minimax games

---

1: **Input:** step size $\gamma$, initialization $\mathbf{u}_0, \mathbf{v}_0$
2: **Initialize:** $\forall m \in [M]$, $\mathbf{u}_0^{r,0} \leftarrow \mathbf{u}_0$, $\quad \mathbf{y}_0^{r,0} \leftarrow \mathbf{v}_0$
3: **for** $r \in \{1, \dots, R\}$ **do**
4: $\quad$ $\forall m \in [M]$, $\mathbf{u}_0^{m,r} \leftarrow \bar{\mathbf{u}}_0^r$, $\quad \mathbf{y}_0^{m,r} \leftarrow \bar{\mathbf{v}}_0^r$
5: $\quad$ **for** $k \in \{0, \dots, K-1\}$ **do**
6: $\quad\quad$ **for** $m \in \{1, \dots, M\}$ **in parallel do**
7: $\quad\quad\quad$ Update local model $\mathbf{u}_{k+1}^{m,r} \leftarrow \mathbf{u}_k^{m,r} - \gamma\nabla f(\mathbf{u}_k^{m,r}, \mathbf{v}_0^{m,r})$
8: $\quad\quad\quad$ Update local model $\mathbf{v}_{k+1}^{m,r} \leftarrow \mathbf{v}_k^{m,r} + \gamma\nabla f(\mathbf{u}_0^{m,r}, \mathbf{v}_k^{m,r})$
9: $\quad\quad$ **end for**
10: $\quad$ **end for**
11: $\quad$ $\bar{\mathbf{u}}_0^{r+1} \leftarrow \frac{1}{M}\sum_{m=1}^M \mathbf{u}_K^{m,r}$, $\quad \bar{\mathbf{v}}_0^{r+1} \leftarrow \frac{1}{M}\sum_{m=1}^M \mathbf{v}_K^{m,r}$
12: $\quad$ **Communicate** $\bar{\mathbf{u}}_K^r$ to all processors with $\mathbf{v}$ player and $\bar{\mathbf{v}}_K^r$ to all processors with $\mathbf{u}$ player
13: **end for**
14: **Output:** $\bar{\mathbf{u}}_K^R, \bar{\mathbf{v}}_K^R$

---

In Algorithm 3, we discuss the distributed version of our method, where two players $\mathbf{u}$ and $\mathbf{v}$ have their data distributed across $M$ processors each. At every round, each set of processors update their local models while having access to an **outdated** version of the other opponent parameters which was received at the beginning of the round. By the end of the round, both set of $\mathbf{u}$ and $\mathbf{v}$ processors send the their parameters to a central server which will compute the average of the parameters and send them back to all processors.

**Theorem 26.** *For any $K, R, L > 0, \mu > 0$ after running Decoupled SGDA for a total of $T = KR$ iterations on the problems in the form of (38) in a distributed setting with $2M$ clients using a stepsize of $\gamma \leqslant \frac{\mu}{32L^2K}$, assuming that $\|\mathbf{x}_0 - \mathbf{x}^\star\|^2 \leqslant D^2$, we have the following convergence rate:*

$$\mathbb{E}\left[\left\|\bar{\mathbf{x}}_K^R - \mathbf{x}^\star\right\|^2\right] \leqslant B^2 \exp\left(-\frac{\gamma\mu KR}{2}\right) + \frac{96K^2L^2\gamma^2\zeta_\star^2}{\mu^2} + \frac{6KL^2\gamma^2\sigma^2}{\mu^2} + \frac{2\gamma\sigma^2}{M\mu} \tag{41}$$

**Corollary 27.** *After choosing a stepsize of $\gamma = \min\left\{\frac{\mu}{32NKL^2}, \frac{\ln(\max\{2, \mu^2 B^2 KR/\sigma^2\})}{\mu KR}\right\}$, we get a rate of:*

$$\mathbb{E}\left[\left\|\bar{\mathbf{x}}_K^R - \mathbf{x}^\star\right\|^2\right] = \tilde{\mathcal{O}}\left(B^2\exp\left(-\frac{\mu^2}{L^2}R\right) + \frac{L^2\zeta_\star^2}{\mu^4R^2} + \frac{L^2\sigma^2}{\mu^4KR^2} + \frac{\sigma^2}{\mu^2MKR}\right) \tag{42}$$

| Method | Heterogeneous | Homogeneous |
|--------|---------------|-------------|
| Local SGDA | $\mathcal{O}\left(\frac{L^6}{\mu^6 R^3} + \frac{\sigma^2}{\mu^2 MKR} + \frac{L^2\zeta_\star^2}{\mu^3 MKR} + \frac{L^2\sigma^2}{\mu^3 MKR}\right)$ | $\tilde{\mathcal{O}}\left(\frac{1}{K^2R^2} + \frac{\sigma^2}{\mu^2 nKR} + \frac{L^2\sigma^2}{\mu^4 MKR} + \frac{L^2\sigma^2}{\mu^4 MK^2R^2}\right)$ |
| Local SGD | $\mathcal{O}\left(LB^2\exp\left(-\frac{\mu}{L}R\right) + \frac{\sigma^2}{\mu MKR} + \frac{L\zeta_\star^2}{\mu^2 R^2} + \frac{L\sigma^2}{\mu^2 KR^2}\right)$ | $\mathcal{O}\left(LB^2\exp\left(-\frac{\mu}{L}KR\right) + \frac{\sigma^2}{\mu KMR} + \frac{Q^2\sigma^4}{\mu^5 K^2R^4}\right)$ |
| **Ours** | $\tilde{\mathcal{O}}\left(B^2\exp\left(-\frac{\mu^2}{L^2}R\right) + \frac{L^2\zeta_\star^2}{\mu^4 R^2} + \frac{L^2\sigma^2}{\mu^4 KR^2} + \frac{\sigma^2}{\mu^2 MKR}\right)$ | - |

Table 4: Comparison of Methods in Heterogeneous and Homogeneous Settings

Table 4 compares state-of-art rates for Local SGD, Local SGDA with Federated Decoupled SGDA. It's clear that our rate matches the tightest known upper bound for Local SGD in heterogeneous

regime with $\zeta_\star$ assumption. Note that the condition number $\kappa^2$ in the first term of our rate matches the GDA conditioning. Local SGD achieves a better conditioning of $\kappa$ due to the fact that the problem is minimization (not minimax). Moreover, our conditioning is much better that Deng & Mahdavi (2021) with the conditioning $\kappa^6$. In addition, it seems that in the rate of Deng & Mahdavi (2021), the term that captures heterogeneity $\zeta_\star$ decreases with taking local steps. However, it contracts some lower bounds on Local SGD proposed in Patel et al. (2024).

### F.3 MISSING PROOFS FOR SECTION F

**Lemma 28** (Consensus Error). *After running Decoupled Local SGDA for $k$ local steps at some round $r$ with a step-size of $\gamma \leqslant \frac{\mu}{32L^2K}$, the error $\Psi(\mathbf{x}_k^{m,r}) + \Phi(\bar{\mathbf{x}}_k^r)$ can be upper bounded as follows:*

$$\mathbb{E}[\Psi(\mathbf{x}_k^{m,r}) + \Phi(\bar{\mathbf{x}}_k^r)] \leqslant \sum_{i=1}^{K} \frac{\mu^2}{8KL^2} \left\| \bar{\mathbf{x}}_i^r - \mathbf{x}^\star \right\|^2 + 32K^2\gamma^2\zeta_\star^2 + \frac{2K\gamma^2\sigma^2}{M} + 2K\gamma^2\sigma^2 \quad (43)$$

In this setting, we have two different errors related to the use of outdated gradients and deviation from the average iterates. Total error is the sum of both errors. We define the consensus error in this setting as follows:

$$\Psi(\mathbf{u}_k^{m,r}) := \frac{1}{M} \sum_{m=1}^{M} \left\| \mathbf{u}_k^{m,r} - \bar{\mathbf{u}}_k^r \right\|^2, \quad \Psi(\mathbf{v}_k^{m,r}) := \frac{1}{M} \sum_{m=1}^{M} \left\| \mathbf{v}_k^{m,r} - \bar{\mathbf{v}}_k^r \right\|^2$$

$$\Phi(\bar{\mathbf{u}}_k^r) := \left\| \bar{\mathbf{u}}_0^r - \bar{\mathbf{u}}_k^r \right\|^2, \quad \Phi(\bar{\mathbf{v}}_k^r) := \left\| \bar{\mathbf{v}}_0^r - \bar{\mathbf{v}}_k^r \right\|^2$$

$$\Psi(\mathbf{x}_k^{m,r}) = \Psi(\mathbf{u}_k^{m,r}) + \Psi(\mathbf{v}_k^{m,r}), \quad \Phi(\bar{\mathbf{x}}_k^r) = \Phi(\bar{\mathbf{u}}_k^r) + \Phi(\bar{\mathbf{v}}_k^r)$$

The total consensus error can be computed by summing both errors with respect to $\mathbf{u}$ and $\mathbf{v}$:

$$\text{Consensus error} := \boxed{\begin{array}{c} \Psi(\mathbf{u}_k) + \Psi(\mathbf{v}_k) \\ \hline \text{error caused by} \\ \text{deviation from average} \end{array}} + \boxed{\begin{array}{c} \Phi(\mathbf{u}_k) + \Phi(\mathbf{v}_k) \\ \hline \text{error caused by} \\ \text{outdated gradients} \end{array}}$$

In the following, the upper bound for consensus error in different settings will be discussed. Note that in the case of multi client, we get different upper bounds based on the assumption on data heterogeneity.

*Proof.*

$$\mathbb{E}[\Psi(\mathbf{u}_{k+1}^{m,r}) + \Phi(\bar{\mathbf{u}}_{k+1}^{r})]$$

$$= \frac{1}{M} \sum_{m=1}^{M} \mathbb{E} \left\| \mathbf{u}_k^{m,r} - \gamma \nabla_{\mathbf{u}} f_m(\mathbf{u}_k^{m,r}, \mathbf{v}_0^{m,r}; \xi_m) - \bar{\mathbf{u}}_k^r + \frac{\gamma}{M} \sum_{m=1}^{M} \nabla_{\mathbf{u}} f_m(\mathbf{u}_k^{m,r}, \mathbf{v}_0^{m,r}; \xi_m) \right\|^2 +$$

$$\mathbb{E} \left\| \bar{\mathbf{u}}_0^r - \bar{\mathbf{u}}_k^r + \frac{\gamma}{M} \sum_{m=1}^{M} \nabla_u f_m(\mathbf{u}_k^{m,r}, \mathbf{v}_0^{m,r}; \xi_m) \right\|^2$$

$$= \frac{1}{M} \sum_{m=1}^{M} \mathbb{E} \left\| \mathbf{u}_k^{m,r} - \gamma \nabla_u f_m(\mathbf{u}_k^{m,r}, \mathbf{v}_0^{m,r}) - \bar{\mathbf{u}}_k^r + \frac{\gamma}{M} \sum_{m=1}^{M} \nabla_u f_m(\mathbf{u}_k^{m,r}, \mathbf{v}_0^{m,r}) \right\|^2 +$$

$$\mathbb{E} \left\| \bar{\mathbf{u}}_0^r - \bar{\mathbf{u}}_k^r + \frac{\gamma}{M} \sum_{m=1}^{M} \nabla_u f_m(\mathbf{u}_k^{m,r}, \mathbf{v}_0^{m,r}) \right\|^2 + \frac{\gamma^2 \sigma^2}{M} + \gamma^2 \sigma^2$$

$$\leqslant \left(1 + \frac{1}{K}\right) \mathbb{E}[\Psi(\mathbf{u}_k^{m,r}) + \Phi(\bar{\mathbf{u}}_k^r)] + \frac{2K\gamma^2}{M} \sum_{m=1}^{M} \mathbb{E} \left\| \nabla_u f_m(\mathbf{u}_k^{m,r}, \mathbf{v}_0^{m,r}) - \frac{1}{M} \sum_{m=1}^{M} \nabla_u f_m(\mathbf{u}_k^{m,r}, \mathbf{v}_0^{m,r}) \right\|^2 +$$

$$\frac{2K\gamma^2}{M} \sum_{m=1}^{M} \mathbb{E} \left\| \nabla_u f_m(\mathbf{u}_k^{m,r}, \mathbf{v}_0^{m,r}) \right\|^2 + \frac{\gamma^2 \sigma^2}{M} + \gamma^2 \sigma^2$$

$$\leqslant \left(1 + \frac{1}{K}\right) \mathbb{E}[\Psi(\mathbf{u}_k^{m,r}) + \Phi(\bar{\mathbf{u}}_k^r)] + \frac{4K\gamma^2}{M} \sum_{m=1}^{M} \mathbb{E} \left\| \nabla_u f_m(\mathbf{u}_k^{m,r}, \mathbf{v}_0^{m,r}) \right\|^2 + \frac{\gamma^2 \sigma^2}{M} + \gamma^2 \sigma^2$$

$$= \left(1 + \frac{1}{K}\right) \mathbb{E}[\Psi(\mathbf{u}_k^{m,r}) + \Phi(\bar{\mathbf{u}}_k^r)]+$$

$$\frac{4K\gamma^2}{M} \sum_{m=1}^{M} \mathbb{E} \left\| \nabla_u f_m(\mathbf{u}_k^{m,r}, \mathbf{v}_0^{m,r}) - \nabla_u f_m(\bar{\mathbf{u}}_k^r, \bar{\mathbf{v}}_k^r) + \nabla_u f_m(\bar{\mathbf{u}}_k^r, \bar{\mathbf{v}}_k^r) \right\|^2 + \frac{\gamma^2 \sigma^2}{M} + \gamma^2 \sigma^2$$

$$\leqslant \left(1 + \frac{1}{K}\right) \mathbb{E}[\Psi(\mathbf{u}_k^{m,r}) + \Phi(\bar{\mathbf{u}}_k^r)] + \frac{8K\gamma^2}{M} \sum_{m=1}^{M} \mathbb{E} \left\| \nabla_u f_m(\mathbf{u}_k^{m,r}, \mathbf{v}_0^{m,r}) - \nabla_u f_m(\bar{\mathbf{u}}_k^r, \bar{\mathbf{v}}_k^r) \right\|^2 +$$

$$\frac{8K\gamma^2}{M} \sum_{m=1}^{M} \mathbb{E} \left\| \nabla_u f_m(\bar{\mathbf{u}}_k^r, \bar{\mathbf{v}}_k^r) \right\|^2 + \frac{\gamma^2 \sigma^2}{M} + \gamma^2 \sigma^2$$

$$\leqslant \left(1 + \frac{1}{K}\right) \mathbb{E}[\Psi(\mathbf{u}_k^{m,r}) + \Phi(\bar{\mathbf{u}}_k^r)] + 8KL^2\gamma^2 \mathbb{E}[\Psi(\mathbf{u}_k^{m,r})] + 8KL^2\gamma^2 \mathbb{E}[\Phi(\bar{\mathbf{v}}_k^r)]+$$

$$\frac{8K\gamma^2}{M} \sum_{m=1}^{M} \mathbb{E} \left\| \nabla_u f_m(\bar{\mathbf{u}}_k^r, \bar{\mathbf{v}}_k^r) \right\|^2 + \frac{\gamma^2 \sigma^2}{M} + \gamma^2 \sigma^2$$

$$= \left(1 + \frac{1}{K}\right) \mathbb{E}[\Psi(\mathbf{u}_k^{m,r}) + \Phi(\bar{\mathbf{u}}_k^r)] + 8KL^2\gamma^2 \mathbb{E}[\Psi(\mathbf{u}_k^{m,r}) + \Phi(\bar{\mathbf{v}}_k^r)]+$$

$$\frac{8K\gamma^2}{M} \sum_{m=1}^{M} \mathbb{E} \left\| \nabla_u f_m(\bar{\mathbf{u}}_k^r, \bar{\mathbf{v}}_k^r) - \nabla_u f_m(\mathbf{u}^\star, \mathbf{v}^\star) + \nabla_u f_m(\mathbf{u}^\star, \mathbf{v}^\star) \right\|^2 + \frac{\gamma^2 \sigma^2}{M} + \gamma^2 \sigma^2$$

$$\left(1 + \frac{1}{K}\right) \mathbb{E}[\Psi(\mathbf{u}_k^{m,r}) + \Phi(\bar{\mathbf{u}}_k^r)] + 8KL^2\gamma^2 \mathbb{E}[\Psi(\mathbf{u}_k^{m,r}) + \Phi(\bar{\mathbf{v}}_k^r)]+$$

$$\frac{16K\gamma^2}{M} \sum_{m=1}^{M} \mathbb{E} \left\| \nabla_u f_m(\bar{\mathbf{u}}_k^r, \bar{\mathbf{v}}_k^r) - \nabla_u f_m(\mathbf{u}^\star, \mathbf{v}^\star) \right\|^2 + 16K\gamma^2 \zeta_\star^2 + \frac{\gamma^2 \sigma^2}{M} + \gamma^2 \sigma^2$$

we continue:

$$\mathbb{E}[\Psi(\mathbf{u}_{k+1}^{m,r}) + \Phi(\bar{\mathbf{u}}_{k+1}^{r})] \leqslant \left(1 + \frac{1}{K}\right) \mathbb{E}[\Psi(\mathbf{u}_{k}^{m,r}) + \Phi(\bar{\mathbf{u}}_{k}^{r})] + 8KL^2\gamma^2 \mathbb{E}[\Psi(\mathbf{u}_{k}^{m,r}) + \Phi(\bar{\mathbf{v}}_{k}^{r})] +$$

$$16KL^2\gamma^2 \mathbb{E}\|\bar{\mathbf{x}}_k^r - \mathbf{x}^\star\|^2 + 16K\gamma^2\zeta_\star^2 + \frac{\gamma^2\sigma^2}{M} + \gamma^2\sigma^2$$

After doing the same computation with respect to $\mathbf{v}$ we get:

$$\mathbb{E}[\Psi(\mathbf{v}_{k+1}^{m,r}) + \Phi(\bar{\mathbf{v}}_{k+1}^{r})]$$

$$\leqslant \left(1 + \frac{1}{K}\right) \mathbb{E}[\Psi(\mathbf{v}_{k}^{m,r}) + \Phi(\bar{\mathbf{v}}_{k}^{r})] + 8KL^2\gamma^2 \mathbb{E}[\Psi(\mathbf{v}_{k}^{m,r}) + \Phi(\bar{\mathbf{u}}_{k}^{r})] +$$

$$16KL^2\gamma^2 \mathbb{E}\|\bar{\mathbf{x}}_k^r - \mathbf{x}^\star\|^2 + 16K\gamma^2\zeta_\star^2 + \frac{\gamma^2\sigma^2}{M} + \gamma^2\sigma^2$$

Now we sum up both inequalities and we get:

$$\mathbb{E}[\Psi(\mathbf{x}_{k+1}^{m,r}) + \Phi(\bar{\mathbf{x}}_{k+1}^{r})]$$

$$\leqslant \left(1 + \frac{1}{K}\right) \mathbb{E}[\Psi(\mathbf{x}_{k}^{m,r}) + \Phi(\bar{\mathbf{x}}_{k}^{r})] + 8KL^2\gamma^2 \mathbb{E}[\Psi(\mathbf{x}_{k}^{m,r}) + \Phi(\bar{\mathbf{x}}_{k}^{r})] +$$

$$32KL^2\gamma^2 \mathbb{E}\|\bar{\mathbf{x}}_k^r - \mathbf{x}^\star\|^2 + 32K\gamma^2\zeta_\star^2 + \frac{2\gamma^2\sigma^2}{M} + 2\gamma^2\sigma^2$$

With the choice of $\gamma \leqslant \frac{\mu}{32L^2K}$ we simplify the above inequality as:

$$\mathbb{E}[\Psi(\mathbf{x}_{k+1}^{m,r}) + \Phi(\bar{\mathbf{x}}_{k+1}^{r})]$$

$$\leqslant \left(1 + \frac{1}{K} + \frac{1}{128K}\right) \mathbb{E}[\Psi(\mathbf{x}_{k}^{m,r}) + \Phi(\bar{\mathbf{x}}_{k}^{r})] + \frac{\mu^2}{32KL^2} \mathbb{E}\|\bar{\mathbf{x}}_k^r - \mathbf{x}^\star\|^2 + 32K\gamma^2\zeta_\star^2 + \frac{2\gamma^2\sigma^2}{M} + 2\gamma^2\sigma^2$$

After unrolling the recursion for the last $K$ steps and considering the fact that $\left(1 + \frac{1}{K} + \frac{1}{128K}\right)^K \leqslant 4$ we have:

$$\mathbb{E}[\Psi(\mathbf{x}_{k+1}^{m,r}) + \Phi(\bar{\mathbf{x}}_{k+1}^{r})] \leqslant \sum_{i=1}^{K} \frac{\mu^2}{8KL^2} \mathbb{E}\|\bar{\mathbf{x}}_i^r - \mathbf{x}^\star\|^2 + 32K^2\gamma^2\zeta_\star^2 + \frac{2K\gamma^2\sigma^2}{M} + 2K\gamma^2\sigma^2$$

$\square$

### F.4  PROOF OF THEOREM 26

For any $K, R, L > 0, \mu > 0$ after running Decoupled SGDA for a total of $T = KR$ iterations on the problems in the form of (38) in a distributed setting with $2M$ clients using a stepsize of $\gamma \leqslant \frac{\mu}{32L^2K}$, assuming that $\|\mathbf{x}_0 - \mathbf{x}^\star\|^2 \leqslant B^2$, we have the following convergence rate:

$$\mathbb{E}\|\bar{\mathbf{x}}_K^R - \mathbf{x}^\star\|^2 \leqslant B^2 \exp\left(-\frac{\gamma\mu KR}{2}\right) + \frac{96K^2L^2\gamma^2\zeta_\star^2}{\mu^2} + \frac{6KL^2\gamma^2\sigma^2}{\mu^2} + \frac{2\gamma\sigma^2}{M\mu} \qquad (44)$$

*Proof.* We start by upper bounding the distance between the average iterate $\bar{\mathbf{u}}^r_{k+1}$ and the saddle point.

$$\mathbb{E}\left\|\bar{\mathbf{u}}^r_{k+1} - \mathbf{u}^\star\right\|^2$$

$$= \mathbb{E}\left\|\bar{\mathbf{u}}^r_k - \frac{\gamma}{M}\sum_{m=1}^M \nabla_u f_m(\mathbf{u}^{m,r}_k, \mathbf{v}^{m,r}_0; \xi_m) - \mathbf{u}^\star\right\|^2$$

$$\leqslant \mathbb{E}\left\|\bar{\mathbf{u}}^r_k - \frac{\gamma}{M}\sum_{m=1}^M \nabla_u f_m(\mathbf{u}^{m,r}_k, \mathbf{v}^{m,r}_0) - \mathbf{u}^\star\right\|^2 + \frac{\gamma^2\sigma^2}{M}$$

$$= \mathbb{E}\left\|\bar{\mathbf{u}}^r_k + \frac{\gamma}{M}\sum_{m=1}^M \nabla_u f_m(\bar{\mathbf{u}}^r_k, \bar{\mathbf{v}}^r_k) - \frac{\gamma}{M}\sum_{m=1}^M \nabla_u f_m(\mathbf{u}^{m,r}_k, \mathbf{v}^{m,r}_0) - \frac{\gamma}{M}\sum_{m=1}^M \nabla_u f_m(\bar{\mathbf{u}}^r_k, \bar{\mathbf{v}}^r_k) - \mathbf{u}^\star\right\|^2 + \frac{\gamma^2\sigma^2}{M}$$

$$\leqslant \left(1 + \frac{\gamma\mu}{2}\right)\mathbb{E}\left\|\bar{\mathbf{u}}^r_k - \frac{\gamma}{M}\sum_{m=1}^M \nabla_u f_m(\bar{\mathbf{u}}^r_k, \bar{\mathbf{v}}^r_k) - \mathbf{u}^\star\right\|^2 +$$

$$\left(1 + \frac{2}{\gamma\mu}\right)\frac{\gamma^2}{M}\sum_{m=1}^M \mathbb{E}\left\|\nabla_u f_m(\bar{\mathbf{u}}^r_k, \bar{\mathbf{v}}^r_k) - \nabla_u f_m(\mathbf{u}^{m,r}_k, \mathbf{v}^{m,r}_0)\right\|^2 + \frac{\gamma^2\sigma^2}{M}$$

For the first term in the above inequality we have:

$$\left(1 + \frac{\gamma\mu}{2}\right)\mathbb{E}\left\|\bar{\mathbf{u}}^r_k - \frac{\gamma}{M}\sum_{m=1}^M \nabla_u f_m(\bar{\mathbf{u}}^r_k, \bar{\mathbf{v}}^r_k) - \mathbf{u}^\star\right\|^2$$

$$= \left(1 + \frac{\gamma\mu}{2}\right)\mathbb{E}\left\|\bar{\mathbf{u}}^r_k - \gamma\nabla_u f(\bar{\mathbf{u}}^r_k, \bar{\mathbf{v}}^r_k) - \mathbf{u}^\star\right\|^2$$

$$= \left(1 + \frac{\gamma\mu}{2}\right)\mathbb{E}\left[\left\|\bar{\mathbf{u}}^r_k - \mathbf{u}^\star\right\|^2 + \gamma^2\left\|\nabla_u f(\bar{\mathbf{u}}^r_k, \bar{\mathbf{v}}^r_k)\right\|^2 - 2\gamma\langle\bar{\mathbf{u}}^r_k - \mathbf{u}^\star, \nabla_u f(\bar{\mathbf{u}}^r_k, \bar{\mathbf{v}}^r_k)\rangle\right]$$

$$\leqslant \left(1 + \frac{\gamma\mu}{2}\right)\mathbb{E}\left[(1 + \gamma^2 L^2)\left\|\bar{\mathbf{u}}^r_k - \mathbf{u}^\star\right\|^2 - 2\gamma\langle\bar{\mathbf{u}}^r_k - \mathbf{u}^\star, \nabla_u f(\bar{\mathbf{u}}^r_k, \bar{\mathbf{v}}^r_k)\rangle\right]$$

For the second term we also have:

$$\left(1 + \frac{2}{\gamma\mu}\right)\frac{\gamma^2}{M}\sum_{m=1}^M \mathbb{E}\left\|\nabla_u f_m(\bar{\mathbf{u}}^r_k, \bar{\mathbf{v}}^r_k) - \nabla_u f_m(\mathbf{u}^{m,r}_k, \mathbf{v}^{m,r}_0)\right\|^2$$

$$\leqslant \left(1 + \frac{2}{\gamma\mu}\right)\frac{L^2\gamma^2}{M}\sum_{m=1}^M \mathbb{E}\left\|\bar{\mathbf{u}}^r_k - \mathbf{u}^{m,r}_k\right\|^2 + \left(1 + \frac{2}{\gamma\mu}\right)\frac{L^2\gamma^2}{M}\sum_{m=1}^M \mathbb{E}\left\|\bar{\mathbf{v}}^r_k - \bar{\mathbf{v}}^r_0\right\|^2$$

$$= \left(1 + \frac{2}{\gamma\mu}\right)L^2\gamma^2\,\mathbb{E}\left[\Psi(\mathbf{u}^{m,r}_k)\right] + \left(1 + \frac{2}{\gamma\mu}\right)L^2\gamma^2\,\mathbb{E}[\Phi(\bar{\mathbf{v}}^r_k)]$$

Where in the last line, we used the fact that $\mathbf{v}_0^{m,r} = \bar{\mathbf{v}}_0^r$. We then repeat the same computation with respect to $\mathbf{v}$.

$$\mathbb{E}\left\|\bar{\mathbf{v}}_{k+1}^r - \mathbf{v}^\star\right\|^2 =$$

$$= \mathbb{E}\left\|\bar{\mathbf{v}}_k^r + \frac{\gamma}{M}\sum_{m=1}^M \nabla_v f_m(\mathbf{u}_0^{m,r}, \mathbf{v}_k^{m,r}; \xi_m) - \mathbf{v}^\star\right\|^2$$

$$\leqslant \mathbb{E}\left\|\bar{\mathbf{v}}_k^r + \frac{\gamma}{M}\sum_{m=1}^M \nabla_v f_m(\mathbf{u}_0^{m,r}, \mathbf{v}_k^{m,r}) - \mathbf{v}^\star\right\|^2 + \frac{\gamma\sigma^2}{M}$$

$$= \mathbb{E}\left\|\bar{\mathbf{v}}_k^r + \frac{\gamma}{M}\sum_{m=1}^M \nabla_v f_m(\mathbf{u}_0^{m,r}, \mathbf{v}_k^{m,r}) - \frac{\gamma}{M}\sum_{m=1}^M \nabla_v f_m(\bar{\mathbf{u}}_k^r, \bar{\mathbf{v}}_k^r) + \frac{\gamma}{M}\sum_{m=1}^M \nabla_v f_m(\bar{\mathbf{u}}_k^r, \bar{\mathbf{v}}_k^r) - \mathbf{v}^\star\right\|^2 + \frac{\gamma\sigma^2}{M}$$

$$\leqslant \left(1 + \frac{\gamma\mu}{2}\right)\mathbb{E}\left\|\bar{\mathbf{v}}_k^r + \frac{\gamma}{M}\sum_{m=1}^M \nabla_v f_m(\bar{\mathbf{u}}_k^r, \bar{\mathbf{v}}_k^r) - \mathbf{v}^\star\right\|^2 +$$

$$\left(1 + \frac{2}{\gamma\mu}\right)\frac{\gamma^2}{M}\sum_{m=1}^M \mathbb{E}\left\|\nabla_v f_m(\bar{\mathbf{u}}_k^r, \bar{\mathbf{v}}_k^r) - \nabla_v f_m(\mathbf{u}_0^{m,r}, \mathbf{v}_k^{m,r})\right\|^2 + \frac{\gamma\sigma^2}{M}$$

For the first term in the above inequality we have:

$$\left(1 + \frac{\gamma\mu}{2}\right)\mathbb{E}\left\|\bar{\mathbf{v}}_k^r + \frac{\gamma}{M}\sum_{m=1}^M \nabla_v f_m(\bar{\mathbf{u}}_k^r, \bar{\mathbf{v}}_k^r) - \mathbf{v}^\star\right\|^2$$

$$= \left(1 + \frac{\gamma\mu}{2}\right)\mathbb{E}\left\|\bar{\mathbf{v}}_k^r + \gamma\nabla_v f(\bar{\mathbf{u}}_k^r, \bar{\mathbf{v}}_k^r) - \mathbf{v}^\star\right\|^2$$

$$= \left(1 + \frac{\gamma\mu}{2}\right)\mathbb{E}\left[\|\bar{\mathbf{v}}_k^r - \mathbf{v}^\star\|^2 + \gamma^2\|\nabla_v f(\bar{\mathbf{u}}_k^r, \bar{\mathbf{v}}_k^r)\|^2 - 2\gamma\langle\mathbf{v}^\star - \bar{\mathbf{v}}_k^r, \nabla_v f(\bar{\mathbf{u}}_k^r, \bar{\mathbf{v}}_k^r)\rangle\right]$$

$$\leqslant \left(1 + \frac{\gamma\mu}{2}\right)\mathbb{E}\left[(1 + \gamma^2 L^2)\|\bar{\mathbf{v}}_k^r - \mathbf{v}^\star\|^2 - 2\gamma\langle\mathbf{v}^\star - \bar{\mathbf{v}}_k^r, \nabla_v f(\bar{\mathbf{u}}_k^r, \bar{\mathbf{v}}_k^r)\rangle\right]$$

For the second term we also have:

$$\left(1 + \frac{2}{\gamma\mu}\right)\frac{\gamma^2}{M}\sum_{m=1}^M \mathbb{E}\left\|\nabla_v f_m(\bar{\mathbf{u}}_k^r, \bar{\mathbf{v}}_k^r) - \nabla_v f_m(\mathbf{u}_0^{m,r}, \mathbf{v}_k^{m,r})\right\|^2$$

$$\leqslant \left(1 + \frac{2}{\gamma\mu}\right)\frac{L^2\gamma^2}{M}\sum_{m=1}^M \mathbb{E}\|\bar{\mathbf{u}}_k^r - \bar{\mathbf{u}}_0^r\|^2 + \left(1 + \frac{2}{\gamma\mu}\right)\frac{L^2\gamma^2}{M}\sum_{m=1}^M \mathbb{E}\|\bar{\mathbf{v}}_k^r - \mathbf{v}_k^{m,r}\|^2$$

$$= \left(1 + \frac{2}{\gamma\mu}\right)L^2\gamma^2\,\mathbb{E}[\Phi(\bar{\mathbf{u}}_k^r)] + \left(1 + \frac{2}{\gamma\mu}\right)L^2\gamma^2\,\mathbb{E}[\Psi(\mathbf{v}_k^{m,r})]$$

Summing up the results from the inequalities with respect to $\mathbf{u}$ and $\mathbf{v}$ gives us:

$$\mathbb{E}\left\|\bar{\mathbf{x}}_{k+1}^r - \mathbf{x}^\star\right\|^2$$

$$\leqslant \left(1 + \frac{\gamma\mu}{2}\right)\mathbb{E}\left[(1 + \gamma^2 L^2)\|\bar{\mathbf{x}}_k^r - \mathbf{x}^\star\|^2 - 2\gamma\langle\bar{\mathbf{x}}_k^r - \mathbf{x}^\star, F(\bar{\mathbf{x}}_k^r)\rangle\right] + \gamma\left(\gamma L^2 + \frac{2L^2}{\mu}\right)\mathbb{E}\left[\Phi(\bar{\mathbf{x}}_k^r) + \Psi(\mathbf{x}_k^{m,r})\right] + \frac{\gamma^2\sigma^2}{M}$$

$$\leqslant \left(1 + \frac{\gamma\mu}{2}\right)\mathbb{E}\left[(1 + \gamma^2 L^2)\|\bar{\mathbf{x}}_k^r - \mathbf{x}^\star\|^2 - 2\gamma\mu\|\bar{\mathbf{x}}_k^r - \mathbf{x}^\star\|^2\right] + \gamma\left(\gamma L^2 + \frac{2L^2}{\mu}\right)\mathbb{E}\left[\Phi(\bar{\mathbf{x}}_k^r) + \Psi(\mathbf{x}_k^{m,r})\right] + \frac{\gamma^2\sigma^2}{M}$$

$$= \left(1 + \frac{\gamma\mu}{2}\right)\mathbb{E}\left[(1 - 2\gamma\mu + \gamma^2 L^2)\|\bar{\mathbf{x}}_k^r - \mathbf{x}^\star\|^2\right] + \gamma\left(\gamma L^2 + \frac{2L^2}{\mu}\right)\mathbb{E}\left[\Phi(\bar{\mathbf{x}}_k^r) + \Psi(\mathbf{x}_k^{m,r})\right] + \frac{\gamma^2\sigma^2}{M}$$

With the choice of $\gamma \leqslant \frac{\mu}{16L^2}$ we have:

$$\mathbb{E}\left\|\bar{\mathbf{x}}_{k+1}^r - \mathbf{x}^\star\right\|^2$$

$$\leqslant \left(1 - \frac{23\gamma\mu}{16}\right)\mathbb{E}\|\bar{\mathbf{x}}_k^r - \mathbf{x}^\star\|^2 + \frac{33\gamma L^2}{16\mu}\mathbb{E}\left[\Phi(\bar{\mathbf{x}}_k^r) + \Psi(\mathbf{x}_k^{m,r})\right] + \frac{\gamma^2\sigma^2}{M}$$

$$\leqslant \left(1 - \frac{23\gamma\mu}{16}\right)\mathbb{E}\|\bar{\mathbf{x}}_k^r - \mathbf{x}^\star\|^2 + \frac{33\gamma\mu}{128K}\sum_{i=1}^K \|\bar{\mathbf{x}}_i^r - \mathbf{x}^\star\|^2 + \frac{96K^2L^2\gamma^3\zeta_\star^2}{\mu} + \frac{7KL^2\gamma^3\sigma^2}{\mu M} + \frac{6KL^2\gamma^3\sigma^2}{\mu} + \frac{\gamma^2\sigma^2}{M}$$

We change the current notation for simplicity in proof by substituting $r$ and $k$ with $t$. $t$ varies from $0$ to $T = KR$, iterating over all rounds and local steps:

$$\mathbb{E} \|\bar{\mathbf{x}}_{t+1} - \mathbf{x}^\star\|^2 \leqslant \left(1 - \frac{23\gamma\mu}{16}\right) \mathbb{E} \|\bar{\mathbf{x}}_t - \mathbf{x}^\star\|^2 + \frac{33\gamma\mu}{128K} \sum_{i=\max\{0,t-K+1\}}^{t} \|\bar{\mathbf{x}}_i - \mathbf{x}^\star\|^2$$

$$+ \frac{96K^2L^2\gamma^3\zeta_\star^2}{\mu} + \frac{7KL^2\gamma^3\sigma^2}{\mu M} + \frac{6KL^2\gamma^3\sigma^2}{\mu} + \frac{\gamma^2\sigma^2}{M}$$

Here we use the Lemma 11 with the following parameters,

$$s_t = \mathbb{E} \|\bar{\mathbf{x}}_t - \mathbf{x}^\star\|^2 \;,\; a = \frac{23\mu}{16} \;,\; b = \frac{33\mu}{128} \;,\; c = \frac{96K^2L^2\gamma\zeta_\star^2}{\mu} + \frac{7KL^2\gamma\sigma^2}{\mu M} + \frac{6KL^2\gamma\sigma^2}{\mu} + \frac{\gamma\sigma^2}{M}$$

The final inequality is:

$$\mathbb{E} \|\bar{\mathbf{x}}_t - \mathbf{x}^\star\|^2 \leqslant \left(1 - \frac{23\gamma\mu}{32}\right)^t \mathbb{E} \|\mathbf{x}_0 - \mathbf{x}^\star\|^2 + \frac{32}{23\mu} \left(\frac{96K^2L^2\gamma\zeta_\star^2}{\mu} + \frac{7KL^2\gamma\sigma^2}{\mu M} + \frac{6KL^2\gamma\sigma^2}{\mu} + \frac{\gamma\sigma^2}{M}\right)\gamma$$

$$\leqslant \left(1 - \frac{\gamma\mu}{2}\right)^t \mathbb{E} \|\mathbf{x}_0 - \mathbf{x}^\star\|^2 + \frac{96K^2L^2\gamma^2\zeta_\star^2}{\mu^2} + \frac{7KL^2\gamma^2\sigma^2}{\mu^2 M} + \frac{6KL^2\gamma^2\sigma^2}{\mu^2} + \frac{\gamma\sigma^2}{M\mu}$$

Recall that we assumed $\gamma = \frac{\mu}{32KL^2}$ so we have:

$$\mathbb{E} \|\bar{\mathbf{x}}_T - \mathbf{x}^\star\|^2 \leqslant \left(1 - \frac{\gamma\mu}{2}\right)^{KR} \mathbb{E} \|\mathbf{x}_0 - \mathbf{x}^\star\|^2 + \frac{96K^2L^2\gamma^2\zeta_\star^2}{\mu^2} + \frac{6KL^2\gamma^2\sigma^2}{\mu^2} + \frac{2\gamma\sigma^2}{M\mu}$$

By setting $t = T = RK$, we get:

$$\mathbb{E} \|\bar{\mathbf{x}}_T - \mathbf{x}^\star\|^2 \leqslant \left(1 - \frac{\gamma\mu}{2}\right)^{KR} \mathbb{E} \|\mathbf{x}_0 - \mathbf{x}^\star\|^2 + \frac{96K^2L^2\gamma^2\zeta_\star^2}{\mu^2} + \frac{6KL^2\gamma^2\sigma^2}{\mu^2} + \frac{2\gamma\sigma^2}{M\mu}$$

$$\leqslant \exp\left(-\frac{\gamma\mu}{2}KR\right) \mathbb{E} \|\mathbf{x}_0 - \mathbf{x}^\star\|^2 + \frac{96K^2L^2\gamma^2\zeta_\star^2}{\mu^2} + \frac{6KL^2\gamma^2\sigma^2}{\mu^2} + \frac{2\gamma\sigma^2}{M\mu}$$

We can see that with this inequality we can only guarantee convergence to a neighborhood of $\mathbf{x}^\star$. To obtain a convergence the final, as discussed in Stich (2019), we need to choose the step size carefully. If $\frac{\mu}{32KL^2} \geqslant \frac{\ln(\max\{2, \mu^4\|\mathbf{x}_0-\mathbf{x}^\star\|^2 T^2/\sigma^2\})}{\mu T}$ then we choose $\gamma = \frac{\ln(\max\{2, \mu^4\|\mathbf{x}_0-\mathbf{x}^\star\|^2 T^2/\sigma^2\})}{\mu T}$, otherwise if $\frac{\mu}{32KL^2} < \frac{\ln(\max\{2, \mu^4\|\mathbf{x}_0-\mathbf{x}^\star\|^2 T^2/\sigma^2\})}{\mu T}$ then we choose $\gamma = \frac{\mu}{32KL^2}$

we can see that with these choices, we would have:

$$\mathbb{E} \|\bar{\mathbf{x}}_T - \mathbf{x}^\star\|^2 = \tilde{\mathcal{O}} \left(\exp\left(-\frac{\mu^2}{64L^2}R\right) \|\mathbf{x}_0 - \mathbf{x}^\star\|^2 + \frac{K^2L^2\zeta_\star^2}{\mu^4 T^2} + \frac{KL^2\sigma^2}{\mu^4 T^2} + \frac{2\sigma^2}{M\mu^2 T}\right)$$

$\square$

# G  DECOUPLED SGDA WITH GHOST SEQUENCE

In this section, we introduce a new extension to the Decoupled SGDA algorithm called *Ghost Sequence*. The base Decoupled SGDA algorithm, explained earlier, is designed to take advantage of problems with a dominant separable component. It minimizes communication complexity by reusing outdated strategies, which has already been analyzed theoretically in the prevous sections.

However, we can push this idea further by not just reusing old strategies but also **predicting** the opponent's next move. This smarter approach opens up a new line of research, where more advanced methods can be explored for estimating the opponent's strategy, offering directions for future work.

To demonstrate the potential of this approach, we propose *Decoupled SGDA with Ghost Sequence*. The main idea is for each player to predict (or approximate) the next move of the opponent based on their previous actions and behaviour. This is achieved by computing the difference between

successive strategies during synchronization. Using this information, each player can update both their own and their opponent's parameters, leading to improved performance. As shown in Figure 8, Decoupled SGDA with Ghost Sequence can greatly improve the algorithm's performance. It also achieves faster communication, even in highly interactive games, and does not require the problem to be weakly coupled.

For more details, refer to Algorithm 4.

---

**Algorithm 4** Decoupled SGDA with Ghost Sequence

1: **Input:** Step size $\gamma$, initial strategies $\mathbf{x}_0 = (\mathbf{u}_0, \mathbf{v}_0)$, total rounds $R$, local updates $K$
2: **for** $r \in \{1, \ldots, R\}$ **do**
3:      Calculate guess $\Delta_{\mathbf{u}}^r \leftarrow \frac{1}{K}(\mathbf{u}_0^r - \mathbf{u}_K^{r-1})$
4:      Calculate guess $\Delta_{\mathbf{v}}^r \leftarrow \frac{1}{K}(\mathbf{v}_0^r - \mathbf{v}_K^{r-1})$
5:      **for** $t \in \{0, \ldots, K-1\}$ **do**
6:          Update ghost sequence $\tilde{\mathbf{v}}_{t+1}^r \leftarrow \tilde{\mathbf{v}}_{t+1}^r + \Delta_{\mathbf{v}}^r$
7:          Update local strategy $\mathbf{u}_{t+1}^r \leftarrow \mathbf{u}_t^r - \gamma \nabla_u f(\mathbf{u}_t^r, \tilde{\mathbf{v}}_{t+1}^r; \xi_t^r)$
8:          Update ghost sequence $\tilde{\mathbf{u}}_{t+1}^r \leftarrow \tilde{\mathbf{u}}_{t+1}^r + \Delta_{\mathbf{u}}^r$
9:          Update local strategy $\mathbf{v}_{t+1}^r \leftarrow \mathbf{v}_t^r + \gamma \nabla_v f(\tilde{\mathbf{u}}_{t+1}^r, \mathbf{v}_t^r; \xi_t^r)$
10:      **end for**
11:      **Communicate** $(\mathbf{u}_K^r, \mathbf{v}_K^r)$ to other players
12: **end for**
13: **Output:** Final strategies $\mathbf{x}_K^R = (\mathbf{u}_K^R, \mathbf{v}_K^R)$

---

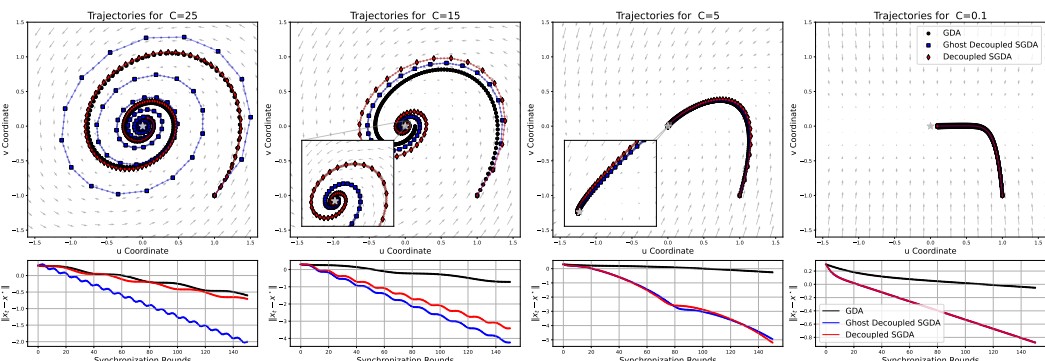

Figure 8: Trajectories and convergence comparison of GDA,*Decoupled SGDA* and *Decoupled SGDA with Ghost Sequence* with different values of $\mathbf{C} = C\mathbf{I}$ (interaction strength). The top row shows the trajectories of the different algorithms for $K = \{1, 5\}$ over varying values of $C \in \{25, 15, 5, 0.1\}$. As $C$ decreases, trajectories become more stable, with the Decoupled SGDA with Ghost Sequence (blue) showing more efficient convergence compared to GDA (black) and Decoupled SGDA (red). The bottom row presents the synchronization rounds versus distance to equilibrium for each configuration, highlighting faster convergence of Decoupled SGDA with Ghost Sequence under larger $\mathbf{C}$ values, while Decoupled SGDA with Ghost Sequence and Decoupled SGDA converge similarly for small $\mathbf{C}$.

# H ADDITIONAL EXPERIMENTS

## H.1 FINDING THE STATIONARY POINT DECOUPLED SGDA FOR NON-CONVEX FUNCTIONS

Here, we add one more figure for the toy GAN problem to provide further insight into the behavior of Decoupled SGDA.

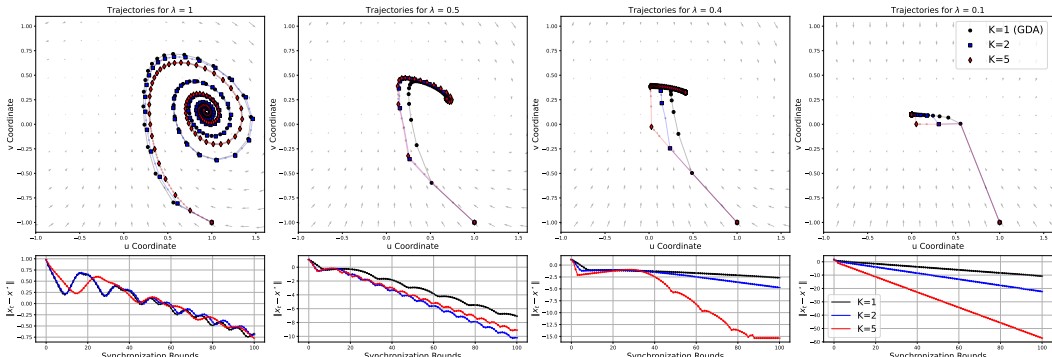

Figure 9: **Trajectories (top row) and distance to equilibrium over synchronization rounds (bottom row) of GDA ($K = 1$) and Decoupled SGDA with $K = \{2, 5\}$ on the** (toyGAN) **problem** ($d = 2$). **C** in (QG) is a constant here—the larger, the stronger the interactive term. **Left-to-right:** decreasing the constant $c \in \{10, 3.5, 2, 7, 0\}$.

## H.2 MORE FIGURES DECOUPLED SGDA WITH GRADIENT APPROXIMATION

In this experiment (Figure 10), Decoupled SGDA achieves lower gradient norms in fewer communication rounds compared to Local SGDA, especially as interaction noise increases (larger c). Decoupled SGDA shows much more stability in high-noise environments, highlighting its effectiveness in dealing with noisy gradients when compared to federated minimax settings.

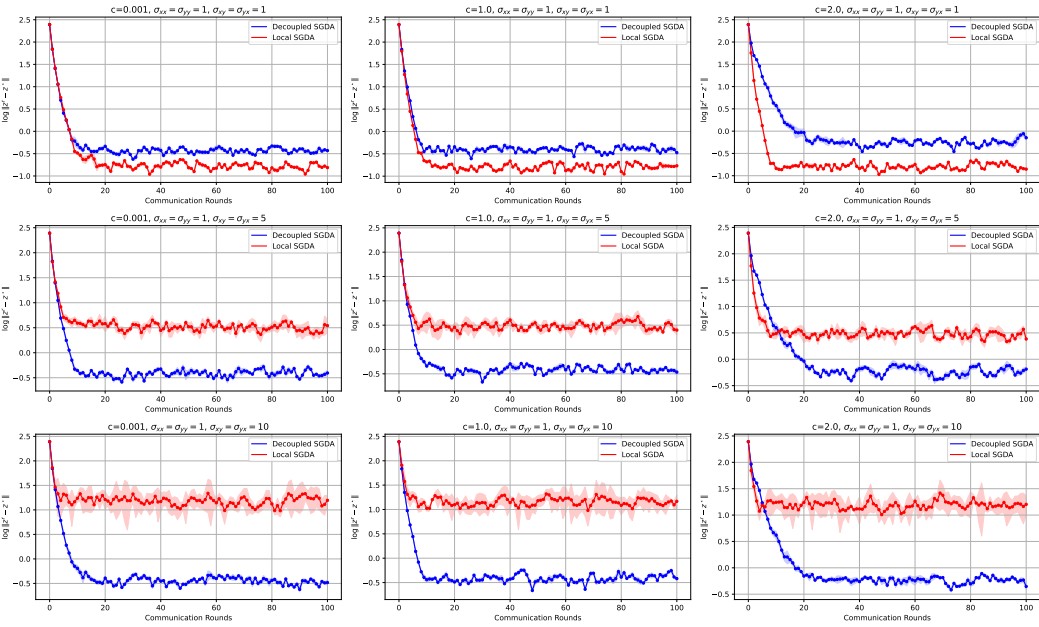

Figure 10: **Comparison of Decoupled SGDA and Local SGDA under different noise settings.** Each plot shows the smallest gradient norm achieved by both algorithms over 100 communication rounds, with varying interaction levels and noise variances. **Top Row:** Different settings of noise variances in off-diagonal entries (interaction noise). **Left to Right:** Increasing values of the constant $c$ controlling the interactive term's strength in the game. Decoupled SGDA consistently outperforms Local SGDA in scenarios where off-diagonal noise is significant, achieving lower gradient norms with fewer communication rounds.

# I  EXPERIMENTAL SETUP

## I.1  FINDING THE SADDLE POINT OF QUADRATIC GAMES

In the first experiment , we conducted tests with a dimensionality of $D = 2$ over $R = 31$ synchronization rounds. The values of $K$ tested were 1, 2, and 5, alongside parameter combinations $(a, b, c)$ set as $(1, 10, 10)$, $(1, 10, 3.5)$, $(1, 10, 2.7)$, and $(1, 10, 0)$. For each combination, we explored gamma values uniformly spaced in the interval $[0.0001, 0.1]$. The algorithm initializes $x$ and $y$ at 1 and $-1$ respectively and updates these variables based on the gradients $g_x$ and $g_y$ computed using the defined parameters.

For the second experiment, in the left figure, eigenvalues were sampled logarithmically between $10^{-1.5}$ and $10^{1.5}$, with random symmetric positive definite matrices generated for each. We tested agent counts $K$ as $[1, 2, 5, 10, 50]$ and learning rates $\gamma$ from $10^{-10}$ to 1. The algorithm ran for $R = 10^5$ rounds, adjusted based on eigenvalue size, to measure the average distance from equilibrium until it fell below $\epsilon = 10^{-6}$. Results were plotted to illustrate the relationship between $\lambda_{max}(C)$ and the number of rounds required for convergence.

For the left figure, we generated random symmetric positive definite matrices as oracles, varying the maximum eigenvalue of the matrix $C$ using logarithmic spacing between $10^{-1.5}$ and $10^{1.5}$. The accuracy threshold is set to $\epsilon = 10^{-4}$. We evaluated five algorithms: GDA, Decoupled GDA, Optimistic, Alternating Gradient Descent, and Extragradient, with $K$ fixed at 50. Each algorithm was executed for $R = 10^5$ rounds, determined based on the maximum eigenvalue, and their performance was assessed by the number of rounds required to achieve $\epsilon$ accuracy.

## I.2  DECOUPLED SGDA WITH GRADIENT APPROXIMATION

In this experiment, we analyze the performance of Decoupled and Local Stochastic Gradient Descent (SGDA) algorithms under varying conditions. We define oracles based on random symmetric positive definite matrices, with a fixed number of rounds $R = 100$ and $K = 40$. The maximum eigenvalues of matrices $C$ are sampled logarithmically between $10^{-0.25}$ and $10^1$, while off-diagonal variances range linearly from 1 to 10. For each maximum eigenvalue, we generate corresponding matrices and evaluate the algorithms across five trials to determine the lowest gradient norm achieved. Results are aggregated and visualized in two plots: one depicting the relationship between the maximum eigenvalue of $C$ and the minimum gradient norm, and the other illustrating the effect of varying off-diagonal variance on algorithm performance.

## I.3  COMMUNICATION EFFICIENCY OF DECOUPLED SGDA FOR NON-CONVEX FUNCTIONS

In this experiment, we investigate the performance of Decoupled Single Oracle GDA under various settings of $\lambda$ and $K$. We evaluate the gradient norm achieved over $R = 100$ communication rounds. The $\lambda$ values are sampled logarithmically between $10^{-4.5}$ and $10^3$, while $K$ values range from 1 to 5. For each combination of $\lambda$ and $K$, we compute the lowest gradient norm over 5 independent trials. The gradient norms are averaged and plotted, with vertical lines marking the transition to the weakly coupled regime at $\lambda = 50$. The final results show the relationship between $\lambda$ and the minimum gradient norm for different values of $K$, highlighting the weakly coupled regime.

## I.4  COMMUNICATION EFFICIENCY OF DECOUPLED SGDA IN GAN TRAINING

In this experiment, a Generative Adversarial Network (GAN) was trained using the CIFAR-10 and SVHN datasets, both resized to $32 \times 32$ pixels. The GAN was trained with a learning rate of $1 \times 10^{-4}$, a batch size of 256, and 50,000 rounds of updates. The hidden dimension size for the generator was 128. For evaluation, 256 samples were used to compute the Fréchet Inception Distance (FID) every 200 iterations. Both the generator and discriminator were optimized using the Adam optimizer, with a learning rate scheduler that decayed by a factor of 0.95 every 1000 steps. Additionally, a gradient penalty term was applied to stabilize training. The generator's latent space dimension was set to 100, and its Exponential Moving Average (EMA) was maintained with a decay factor of 0.999 for evaluation purposes. Training was conducted using CUDA on an NVIDIA L4 GPU.

The Generator uses a series of transposed convolutions, starting from a 100-dimensional latent vector, to generate a $32 \times 32 \times 3$ image, with BatchNorm and ReLU, ending with a Tanh activation. The Discriminator applies four convolutional layers to downsample the input, using LeakyReLU and BatchNorm, and outputs a real/fake probability through a Sigmoid activation.

