# OpenReview forum: "Decoupled SGDA for Games with Intermittent Strategy Communication"
_ICLR.cc/2025/Conference — Submitted to ICLR 2025_

### Official Review · Reviewer_6DZc · 2024-11-01

**Soundness:** 3
**Presentation:** 3
**Contribution:** 2
**Rating:** 6
**Confidence:** 3

**Summary:**

The paper presents the Decoupled SGDA approach for multiplayer games with intermittent communication, where players only occasionally update their strategies based on the actions of their competitors. This model addresses scenarios where continuous communication is impractical due to noise, outdated strategy information, or communication constraints. The authors investigate its convergence properties under SCSC conditions and extend the analysis to weakly coupled games, demonstrating significant reductions in communication costs. They also conduct extensive experiments, comparing Decoupled SGDA to traditional federated minimax and standard SGDA methods, highlighting the proposed approach’s superior performance in both quadratic minimax and non-convex settings.

**Strengths:**

* The method achieves notable communication efficiency when the game is weakly decoupled.
* The analysis covers the SCSC setting and N-player games.
* Strong experimental results verify the theory.

**Weaknesses:**

* The analysis is restricted to strongly convex and strongly monotone games, is it possible to extend to a more general setting?
* It seems the theoretical proof is standard and the technical contribution is weak.

**Questions:**

* In line 287, the function r(u,v) is not clear. Is this the same as r(u,v) in Eq. (5)?
* Regarding the 'Ghost sequence' proposed in Appendix G, could you explain the update of the ghost sequence in Algorithm 4?

---

> ### Author Response · Authors · 2024-11-22
>
> Thanks a lot for your time reviewing our paper and for your feedback.
>
> >The analysis is restricted to strongly convex and strongly monotone games, is it possible to extend to a more general setting?
>
> It is possible to use regularization technique in order to derive a rate for convex-concave games from the current results for strongly-convex-strongly-concave games. However, we leave the direct proof for convex-concave games for the future works.
>
> >It seems the theoretical proof is standard and the technical contribution is weak.
>
>  While the approach on a high level is standard, the difficulty is in identifying regimes and quantifying the acceleration—both of which are novel aspects of this work. Particularly, prior works rely on the assumption $\||\nabla f(x_1,y_1) - \nabla f(x_2,y_2) \||^2 \leq L^2[\||x_1-x_2\||^2 + \||y_1-y_2\||^2]$. We do not rely solely on this assumption, as it can be pessimistic when the interaction between players is low.
>   This necessitates a novel proof approach, particularly leveraging Equation 3 from the paper. Thus, we consider identifying this weakly coupled regime as the novelty of our work.
>
> >In line 287, the function r(u,v) is not clear. Is this the same as r(u,v) in Eq. (5)?
>
> Yes, it is the same $r(u, v)$ as in Eq. 5, thanks for your helpful comment. We added a reference to Eq. 5 in the revised version.
>
> >Regarding the 'Ghost sequence' proposed in Appendix G, could you explain the update of the ghost sequence in Algorithm 4?
>
> In Algorithm 1, each player assumes that the other player’s strategy remains fixed during their local steps. Specifically, in round $r$, player $u$ uses the gradient $\nabla f(u_t^r, v_0^r)$, where $v_0^r$ is kept constant throughout all local steps. Between communication rounds, player $ u $ does not receive updates about player $v $’s strategy, relying only on the information available from the last communication round. Algorithm 4 differs from Algorithm 1 in how it approximates the other player’s strategy during local steps. In Algorithm 4, player $ u$ uses a ghost sequence $ \hat{v} $ to dynamically approximate player$ v $’s strategy. The ghost sequence is updated during each local step by adding
> $$ \Delta\_{v}^r := \frac{1}{K} (v^r_0 - v^{r-1}\_0),$$
> where $\Delta_{v}^r$represents an estimated average change in player $ v $’s strategy between communication rounds.
>
> In this approach, Algorithm 4 adjusts dynamically to the other player’s strategy changes, rather than treating it as fixed. While this is a simple way to build the ghost sequence, exploring more advanced methods is an interesting future research direction. Algorithm 4 sets a basis framework for the development of other advanced methods. Although this is a heuristic, our experiments showed that it works well in practice. We leave the theoretical analysis of this method to future work.
>
> We would like to thank the reviewer again for their comment and time. Please consider increasing your score if you believe your concerns have been resolved. Otherwise, please let us know so that we can provide more explanations.

---

### Official Review · Reviewer_FnWw · 2024-11-04

**Soundness:** 3
**Presentation:** 4
**Contribution:** 3
**Rating:** 8
**Confidence:** 4

**Summary:**

This paper proposes an algorithm called Decoupled SGDA which extends stochastic GDA for the setting where information about players' strategies is noisy or outdated. In this algorithm, players update their strategies 'locally' using outdated strategies from the other players, and then perform a synchronization step to exchange all updated strategies. In strongly-convex-strongly-concave minmax games and in N-player strongly concave games that exhibits the weakly-coupled property, the paper shows that Decoupled SGDA demonstrates communication acceleration (i.e. reduced communication complexity) over baseline methods. Moreover, with a slightly stronger assumption on the coupling degree, Decoupled SGDA can be shown to exhibit communication acceleration over the optimal first order method FOAM in SCSC minmax games. Finally, the paper presents deeper analysis in several other regimes, including in quadratic minmax games with bilinear coupling between players, in a federated learning setup, and with a heuristic modification to the proposed algorithm called the Ghost Sequence. Moreover, the paper presents experiments that corroborate the theoretical results and also show interesting behavior in settings which are not theoretically studied.

**Strengths:**

- The paper studies an interesting problem setup with many real-world applications and possible extensions. The core research questions are well-formulated and motivated. Moreover, the proposed algorithm is intuitive and simple to implement.
- The paper presents convergence results for Decoupled SGDA, the proposed algorithm, in a number of game settings and shows a meaningful improvement in terms of communication complexity in these settings.
- The experiments presented are quite compelling and provide some interesting extensions to the theory, speaking to the efficacy of Decoupled SGDA.

**Weaknesses:**

- There seems to be a discussion which is missing comparing the convergence rates of the proposed algorithm with standard methods in the literature. Table 1 compares the communication complexity of decoupled SGDA in comparison with existing methods, but it would also be interesting to have a table with the convergence rates to the minmax equilibrium, which would depend on $R,K$.
- The related work section on Federated Learning feels slightly misplaced given that only the experiments in Sec 5.3 are related to Federated Learning. Would it make sense to move the related work for FL to the appendix and expand more on the gradient descent/minmax optimization specific related work in the main text?
- The notation for condition numbers $\kappa_u$, $\kappa_v$ are used before their definition (e.g. in the related work section, in the paragraph before Corollary 3). Meanwhile they are only defined at the bottom of page 6.

- There are also several typos I've found, listed below:
    - Line 157: do denote should be 'to' denote
    - Line 209-210: intiualized should be initialized, and in the definition of (local-SGDA) the $x^u_{t+1}$ in the first summation should be $x^u_{t+i}$.
    - Line 390: Should this be Decoupled SGDA instead of Decoupled GDA?
    - Line 402: Decouped-SGDA should be Decoupled SGDA.
    - Figure 4: The right figure is denoted 'Left', and the reference to 5 links to the experiment section. Should this instead link to Appendix F?

**Questions:**

- How does Decoupled SGDA behave if the stepsizes are decreasing? Many of the existing results in the literature crucially depend on carefully chosen decreasing stepsize, so having a method which uses constant stepsize might be preferable. But regardless, I am curious how the convergence results change if (for instance) the local updates are performed with a decreasing stepsize.
- The fact that only the noise on the self-gradients needs to be bounded for the Decoupled SGDA update seems unintuitive to me, can you comment on what the trade-off is? For instance, does this come at a cost of slower convergence rate if the gradients are too noisy, even if it does not affect the communication complexity?

---

> ### Author Response · Authors · 2024-11-22
>
> Thanks a lot for your time reviewing our paper and for your feedback.
>
> >There seems to be a discussion which is missing comparing the convergence rates of the proposed algorithm with standard methods in the literature. Table 1 compares the communication complexity of decoupled SGDA in comparison with existing methods, but it would also be interesting to have a table with the convergence rates to the minmax equilibrium, which would depend on R,K .
>
> We believe Table 1 addresses the point you raised. In Table 1, we compare the communication complexity of our method with that of other methods. Note that for other methods, the communication complexity is the same as the iteration complexity (i.e., the total number of steps required to reach $\epsilon$ accuracy) because they do not utilize local steps. To make this clearer,
>
> - we changed the caption of Table 1
> - we added citations for the remaining methods in the table
>
> Thanks for raising this.
>
> >The related work section on Federated Learning feels slightly misplaced given that only the experiments in Sec 5.3 are related to Federated Learning. Would it make sense to move the related work for FL to the appendix and expand more on the gradient descent/minmax optimization specific related work in the main text?
>
> Thanks a lot for your improvement suggestions.
> In the uploaded revised version, we moved the Federated Learning discussion to the appendix and expanded the discussion of the min-max related work.
>
>
> >The notation for condition numbers  $\kappa_u$, $\kappa_v$ are used before their definition (e.g. in the related work section, in the paragraph before Corollary 3). Meanwhile they are only defined at the bottom of page 6.
>
> The revised version resolves this; thanks for noticing.
>
>
> >There are also several typos I've found, listed below
>
> Thanks for pointing out the typos. We fixed them in the revised version.
>
> >How does Decoupled SGDA behave if the stepsizes are decreasing? Many of the existing results in the literature crucially depend on carefully chosen decreasing stepsize, so having a method which uses constant stepsize might be preferable. But regardless, I am curious how the convergence results change if (for instance) the local updates are performed with a decreasing stepsize.
>
> Thank you for your insightful question. In this paper, we focus on analyzing SGD with a constant step size. However, we believe it is possible to extend our analysis to variations of SGD that employ a decreasing step size (the consensus error Lemma 17 would need to be adapted accordingly). That said, we are uncertain whether this would further reduce communication complexity. We leave this exploration for future work.
>
> >The fact that only the noise on the self-gradients needs to be bounded for the Decoupled SGDA update seems unintuitive to me, can you comment on what the trade-off is? For instance, does this come at a cost of slower convergence rate if the gradients are too noisy, even if it does not affect the communication complexity?
>
> In our proposed method (Algorithm 1) there are two local update rules,
> $$u_{t+1}^r = u_t^r - \gamma\nabla_{u}
> f(u_t^r, v_0^r)$$
> $$v_{t+1}^r = v_t^r + \gamma\nabla_{v} f(u_0^r, v_t^r)$$
>
> Note that each of the above update rules uses a different oracle for computing gradients, as they are computed on different players (compute nodes) in parallel. We call the gradient oracle queried by player $u$, $G_u$, and the one queried by player $v$, $G_v$. Without loss of generality, let’s only consider the update rule for player $u$'s parameters. We can see that player $u$ only uses its self-gradient for performing local steps. In another word, player $u$ only uses $[G_{u}(x,\xi)]\_u$, although oracle $G_{u}(x,\xi)$ also returns the gradient with respect to parameter $v$ as well ($[G_{u}(x,\xi)]\_v$). Therefore, we can argue that the only necessary assumption on the boundedness of oracle $G_u$'s noise variance is
> $$E[\||[G_{u}(x,\xi)]\_u -[F(x)]\_u\||^2] \le \sigma^2_{u u}$$
>
> In another word, $E[\||[G_{u}(x,\xi)]\_v -[F(x)]\_v\||^2]$ can be arbitrarily large because $[G_{u}(x,\xi)]\_v$ is not used in Algorithm 1. Similarly, we can argue $E[\||[G_{v}(x,\xi)]\_u -[F(x)]\_u\||^2]$ can be arbitrarily large. This is, in fact, an advantage of our algorithm in comparison to other distributed minimax methods, as they require an upper bound on both $E[\||[G_{u}(x,\xi)]\_v -[F(x)]\_v\||^2]$ and $E[\||[G_{v}(x,\xi)]\_u -[F(x)]\_u\||^2]$. It means that a large value of $E[\||[G_{u}(x,\xi)]\_v -[F(x)]\_v\||^2]$ and $E[\||[G_{v}(x,\xi)]\_u -[F(x)]\_u\||^2]$ slows down the convergence of other methods while our method is not affected by that.
>
> We would like to thank the reviewer again for their comment and time.

---

> > ### Comment · Reviewer_FnWw · 2024-11-25
> > **Response to rebuttal**
> >
> > Thank you for the responses, my concerns have been reasonably addressed. Overall, given the improvements to notation and flow in the revised version, I am happy to maintain my score for the paper.

---

### Official Review · Reviewer_yRMF · 2024-11-04

**Soundness:** 3
**Presentation:** 3
**Contribution:** 3
**Rating:** 6
**Confidence:** 3

**Summary:**

This paper studies the problem of reducing the communication complexity of decentralized optimization algorithms in strongly monotone games. The main contribution is a strategy called Decoupled SGDA, where each player conducts $K$ local updates using outdated strategies of the other players, followed by strategy synchronization. They provide convergence and communication complexity analysis for Decoupled SGDA and conditions under which Decoupled SGDA outperforms existing methods. Numerical experiments supports the theoretical results.

**Strengths:**

Reducing the communication complexity of decentralized learning dynamics in games is an important and timely problem. This paper uses games with intermittent strategy communication as a model for limited communication and gives a simple algorithm for better communication complexity. The algorithm is simple and versatile for adaptation of other methods. They provide both theoretical results and experiment results showing the advantage of the algorithm over existing methods.

**Weaknesses:**

1. Many notations (i.e., the Lipschitzness constant and the strong convexity constants) make the theorems less intuitive and hard to interpret. It would be helpful if the algorithm's complexity and comparison with other methods were discussed in more detail.
2. Table 1 presents certain conditions under which the proposed algorithm is faster than existing methods, but the condition is less intuitive. It would be helpful to give concrete examples that satisfy these conditions.

Minor Comments:
1. Line 344-345: incomplete sentence.
2. Corollary 4: "weakly a coupled" -> "a weakly coupled"

**Questions:**

See weakness.
1. Could you give concrete examples that satisfy the conditions in Table 1?

---

> ### Author Response · Authors · 2024-11-22
>
> Thanks a lot for your time reviewing our paper and for your feedback.
>
> >Many notations (i.e., the Lipschitzness constant and the strong convexity constants) make the theorems less intuitive and hard to interpret. It would be helpful if the algorithm's complexity and comparison with other methods were discussed in more detail.
>
> Thanks for your improvement suggestions. We applied several changes to the revised version, which include:
>
>
> - In Equation 3, we replaced $L_{uv}$ and $L_{vu}$ with a single parameter $L_c$. Note that variables $L_{uv}$ and $L_{vu}$ are usually the same (in twice continuously differentiable functions). So we do not loose anything by using one variable $L_c$ instead.
> - We introduced a new parameter $\mu_0$, representing the strong monotonicity parameter for the operator $F_0$, instead of using $\mu_u$ and $\mu_v$. Our analysis only needs $\mu_0$ and we no longer need to keep two separate variables $\mu_u$ and $\mu_v$ (although we show that $\mu_0$ is indeed related to $\mu_u$ and $\mu_v$).
> - The definition of $\theta$ has been simplified to $\theta = \frac{L_c}{\mu_0}$.
> - We also made some minor corrections to fix typos.
>
> >Table 1 presents certain conditions under which the proposed algorithm is faster than existing methods, but the condition is less intuitive. It would be helpful to give concrete examples that satisfy these conditions.
>
> Consider the following quadratic function:
> $$
> f(u, v) = g(u) - h(v) +  \langle Cu, v \rangle,
> $$
> where $C$ is a fixed matrix, and $g(u)$ and $h(v)$ are both $\mu$-strongly convex and $L$-smooth.
>
> To achieve communication acceleration compared to GDA, our method needs $L_c := \||C\||$ to be sufficiently small compared to $\mu$:
> $$
> L_c \le \frac{\mu}{4}.
> $$
>
> Furthermore, for communication acceleration compared to the other algorithms in Table 1, a stronger assumption on the matrix $C$ is required:
> $$
> L_c \leq 2\mu\sqrt{1 - \frac{1}{\kappa}}.
> $$
>
> Here, $\kappa := \frac{L}{\mu}$ is the ``classical'' condition number. With this assumption, $f$ satisfies the conditions specified in the *Speed Up* column of Table 1.
>
> The right-hand side of the above inequality increases with $\kappa$. This implies that the more ill-conditioned the functions $g(u)$ and $h(v)$ in $f$ are, the greater the communication efficiency of our algorithm compared to others.
>
> >Minor Comments:
>
> Thanks for your comment. We fixed them in the revised version.
>
> We would like to thank the reviewer again for their comment and time. Please consider increasing your score if you believe your concerns have been resolved. Otherwise, please let us know so that we can provide more explanations.

---

### Official Review · Reviewer_Khdc · 2024-11-12

**Soundness:** 3
**Presentation:** 3
**Contribution:** 2
**Rating:** 6
**Confidence:** 2

**Summary:**

The paper studies minmax optimization, and in particular, consider the case that players only have access to their own (noisy) gradient information, so they need to communicate frequently with other player to get update information. The goal is to design algorithms that take few iterations to converge, and at the same time, minimize the communication cost.

The paper studies the strongly-convex and strong concave setting (and with some extra smoothness assumptions), and it introduces Decoupled SGDA, an extension of the Stochastic Gradient Descent Ascent (SGDA).
The idea is to let players perform independent updates using outdated strategies of opponents (i.e., the lastest communicated strategy), and only perform periodic synchronization.
The paper proves that Decoupled SGDA achieves near-optimal communication complexity comparable to standard GDA rates.
They introduce the concept of "weakly coupled games" - where the interaction between players is relatively minor compared to their individual objectives - and show that in this regime, their method can significantly reduce communication costs.

The paper also perform experiments on quadratic minmax optimization problem and toy GAN tasks to demonstrate the practical performance of the algorithm.

**Strengths:**

Overall, the paper makes solid contribution to minimax optimization. The idea of Decoupled SGDA seems to be fairly natural, but at the same time, it means the simplicity of the algorithm could make it practical.

**Weaknesses:**

The idea of the algorithm (Decoupled SGDA) is natural, and the analysis seems to be standard.



Minor issue:

I think the focus on the paper is on minmax optimization, instead of games (in particular, two-player zero-sum games are really really special case of games), so I suggest the author to properly changes the title to reflect this fact.

**Questions:**

.

**Details Of Ethics Concerns:**

.

---

> ### Author Response · Authors · 2024-11-22
>
> Thanks a lot for your time reviewing our paper and for your feedback.
>
> >The idea of the algorithm (Decoupled SGDA) is natural, and the analysis seems to be standard.
>
>  While the approach on a high level is standard, the difficulty is in identifying regimes and quantifying the acceleration—both of which are novel aspects of this work. Particularly, prior works rely on the assumption $\||\nabla f(x_1,y_1) - \nabla f(x_2,y_2) \||^2 \leq L^2[\||x_1-x_2\||^2 + \||y_1-y_2\||^2]$. We do not rely solely on this assumption, as it can be pessimistic when the interaction between players is low.
>   This necessitates a novel proof approach, particularly leveraging Equation 3 from the paper. Thus, we consider identifying this weakly coupled regime as the novelty of our work.
>
> >I think the focus on the paper is on minmax optimization, instead of games (in particular, two-player zero-sum games are really really special case of games), so I suggest the author to properly changes the title to reflect this fact.
>
> Note that this work also studies the more general case of $N$-player games, detailed in Appendix C. The main part focuses only on two-player games to simplify the notation. Therefore, we believe the title is adequate, as the broader class of $N$-player games is addressed. This point is highlighted under the contributions in Section 1. Thank you for bringing up this concern.
>
> We would like to thank the reviewer again for their comment and time. Please consider increasing your score if you believe your concerns have been resolved. Otherwise, please let us know so that we can provide more explanations.

---

> > ### Comment · Reviewer_Khdc · 2024-11-26
> > **Minor suggestions**
> >
> > Here is my suggestions for paper writing: If you want to claim the focus is on multi-player game, you should put it into the main paper (otherwise people would think the multi-player results are simple corollary of the main result). I understand that there are page constraints, but I think the bottom line is that you should put the most important staff in the main paper.
> > Also, I note even for multi-player games, you assume concavity, so please consider at least add concavity to your title.

---

> > > ### Author Response · Authors · 2024-11-29
> > >
> > > Thank you for your suggestions. Due to space limitations, we have only included the 2-player case in the main text, as we believe it effectively conveys the core message of our work. Additionally, it aligns well with the quadratic problem setting, which we have explored extensively in the appendix.

---

### Author Response · Authors · 2024-11-29

We sincerely thank all reviewers for their valuable feedback. Reviewer Khdc suggested including the strong monotonicity assumption in the paper’s title for enhanced clarity. We would like to invite the other reviewers to share their opinions on this suggestion, as their input will help us make a well-rounded decision.

---

### Meta-Review · Area_Chair_uJj8 · 2024-12-22

**Metareview:**

The majority of the reviewers are still concerned by the technical novelty of the analysis. I concur with the reviewers on this point, and believe that although the proposed method is faster under certain conditions, there is not a technically new idea in the paper that meets the high bar of must-publish. The authors should revise the paper with this point in mind before submitting it to the next venue.

**Additional Comments On Reviewer Discussion:**

NA

---

### Decision · Program_Chairs · 2025-01-22

Reject